# Reinforcement Learning with Segment Feedback

**Yihan Du**[1]  **Anna Winnicki**[2]  **Gal Dalal**[3]  **Shie Mannor**[4][3]  **R. Srikant**[1]

## Abstract

Standard reinforcement learning (RL) assumes that an agent can observe a reward for each state-action pair. However, in practical applications, it is often difficult and costly to collect a reward for each state-action pair. While there have been several works considering RL with trajectory feedback, it is unclear if trajectory feedback is inefficient for learning when trajectories are long. In this work, we consider a model named RL with segment feedback, which offers a general paradigm filling the gap between per-state-action feedback and trajectory feedback. In this model, we consider an episodic Markov decision process (MDP), where each episode is divided into $m$ segments, and the agent observes reward feedback only at the end of each segment. Under this model, we study two popular feedback settings: binary feedback and sum feedback, where the agent observes a binary outcome and a reward sum according to the underlying reward function, respectively. To investigate the impact of the number of segments $m$ on learning performance, we design efficient algorithms and establish regret upper and lower bounds for both feedback settings. Our theoretical and experimental results show that: under binary feedback, increasing the number of segments $m$ decreases the regret at an exponential rate; in contrast, surprisingly, under sum feedback, increasing $m$ does not reduce the regret significantly.

Reinforcement learning (RL) is a class of sequential decision-making algorithms, where an agent interacts with an unknown environment through time with the goal of maximizing the obtained reward. RL has variant applications such as robotics, autonomous driving and game playing.

In classic RL, when the agent takes an action in a state, the environment will provide a reward for this state-action pair. However, in real-world applications, it is often difficult and costly to collect a reward for each state-action pair. For example, in robotics, when we instruct a robot to scramble eggs, it is hard to specify a reward for each individual action. In autonomous driving, it is difficult and onerous to evaluate each action, considering multiple criteria including safety, comfort and speed.

Motivated by this fact, there have been several works that consider RL with trajectory feedback (Efroni et al., 2021; Chatterji et al., 2021). In these works, the agent observes a reward signal only at the end of each episode, instead of at each step, with the signal indicating the quality of the trajectory generated during the episode. While these works mitigate the issue of impractical per-step reward feedback in classic RL, the relationship between the frequency of feedback and the performance of RL algorithms is unknown. In particular, if for example we get feedback twice in each trajectory, does that significantly improve performance over once per trajectory feedback?

To answer this question, we study a general model called RL with segment feedback, which bridges the gap between per-state-action feedback in classic RL (Sutton & Barto, 2018) and trajectory feedback in recent works (Efroni et al., 2021; Chatterji et al., 2021). In this model, we consider an episodic Markov decision process (MDP), where an episode is equally divided into $m$ segments. In each episode, at each step, the agent first observes the current state, and takes an action, and then transitions to a next state according to the transition distribution. The agent *observes a reward signal at the end of each segment*. Under this model, we consider two reward feedback settings: binary feedback and sum feedback. In the binary feedback setting, the agent observes a binary outcome (e.g., thumbs up/down) generated by a sigmoid function of the reward on this segment. In the sum feedback setting, the agent observes the sum of the rewards over this segment. In our model, the agent needs to learn the underlying reward function (i.e., the expected reward as a function of states and actions) from binary or sum segment feedback, and maximize the expected reward achieved. While Tang et al. (2024) also studied this segment model before (they called it RL from bagged reward), their work is mostly empirical, and does not provide theoretical

[1]University of Illinois at Urbana-Champaign [2]Stanford University [3]NVIDIA Research [4]Technion. Correspondence to: Yihan Du <duyihan1996@gmail.com>, R. Srikant <rsrikant@illinois.edu>.

*Proceedings of the 42$^{nd}$ International Conference on Machine Learning*, Vancouver, Canada. PMLR 267, 2025. Copyright 2025 by the author(s).

guarantees for algorithms and rigorously reveal the influence of segments on learning.

This model is applicable to many scenarios involving human queries. For instance, in autonomous driving, a driving trajectory is often divided into several segments, and human annotators are asked to provide feedback for each segment, e.g., thumbs up/down. Compared to state-action pairs or whole trajectories, segments are easier and more efficient to evaluate, since human annotators can focus on and rate behaviors in each segment, e.g., passing through intersections, reversing the car and parking.

In this segment model, there is an interesting balance between the number of segments (queries to humans) and the collected observations, i.e., we desire more observations, but we also want to reduce the number of queries. Therefore, in this problem, it is critical to investigate the trade-off between the benefits brought by segments and the increase of queries, which essentially comes down to a question: *How does the number of segments $m$ impact learning performance?*

To answer this question, we design efficient algorithms for binary and sum feedback settings in both known and unknown transition cases. Regret upper and lower bounds are provided to rigorously show the influence of the number of segments on learning performance. We also present experiments to validate our theoretical results.

Note that studying RL with equal segments is an important starting point and serves as a foundation for further investigation on more general models and analysis for RL with unequal segments. Even under equal segments, this problem is already very challenging: (i) This problem cannot be solved by applying prior trajectory feedback works, e.g., (Efroni et al., 2021), since they use the martingale property of subsequent trajectories in analysis, while subsequent segments are not a martingale due to *dependency among segments* within a trajectory. (ii) In prior trajectory feedback works (Efroni et al., 2021; Chatterji et al., 2021), there exists a gap between upper and lower bounds for sum feedback, and there is no lower bound for binary feedback. This fact poses a significant challenge for us when trying to understand the influence of the number of segments $m$ on learning performance.

Our work overcomes the above challenges and makes contributions as follows.

1. We study a general model called RL with segment feedback, which bridges the gap between per-state-action feedback in classic RL and trajectory feedback seemlessly. Under this model, we consider two feedback settings: binary feedback and sum feedback.

2. For binary feedback, we design computationally-efficient and sample-efficient algorithms `SegBiTS` and `SegBiTS-Tran` for known and unknown transitions, respectively. We provide regret upper and lower bounds which depend on $\exp(\frac{Hr_{\max}}{2m})$, where $H$ is the length of each episode, and $r_{\max}$ is a universal upper bound of rewards. Our results exhibit that under binary feedback, increasing the number of segments $m$ significantly helps accelerate learning.

3. For sum feedback, we devise algorithms `E-LinUCB` and `LinUCB-Tran`, which achieve near-optimal regrets in terms of $H$ and $m$. We also establish lower bounds to validate the optimality, and show that optimal regrets do not depend on $m$. Our results reveal that surprisingly, under sum feedback, increasing the number of segments $m$ does not help expedite learning much.

4. We develop novel techniques which can be of independent interest, including the KL divergence analysis to derive an exponential lower bound under binary feedback, and the use of E-optimal experimental design in algorithm `E-LinUCB` to refine the eigenvalue of the covariance matrix and reduce the regret.

## 1. Related Work

In this section, we briefly review prior related works.

Algorithms and analysis for classic RL were well studied in the literature (Sutton & Barto, 2018; Jaksch et al., 2010; Azar et al., 2017; Jin et al., 2018; Zanette & Brunskill, 2019). Tang et al. (2024) proposed the RL with segment feedback problem (they called it RL from bagged rewards), and designed a transformer-based algorithm. However, their work is mostly empirical and does not provide theoretical guarantees. Gao et al. (2025) considers RL with bagged decision times, where the state transitions are non-Markovian within a bag, and a reward is observed at the end of the bag. But the focus of (Gao et al., 2025) is to handle the non-Markovian state transitions within a bag using a causal directed acyclic graph, instead of investigating how to infer the reward function of state-action pairs from bagged rewards like us. In addition, to the best of our knowledge, there is no existing work that rigorously quantifies the influence of segments on learning performance.

There are two prior works (Efroni et al., 2021; Chatterji et al., 2021) studying RL with trajectory feedback, which are most related to our work. Efroni et al. (2021) investigated RL with sum trajectory feedback, and designed upper confidence bound (UCB)-type and Thompson sampling (TS)-type algorithms with regret guarantees. Chatterji et al. (2021) studied RL with binary trajectory feedback, but considered a different formulation for binary feedback from ours. Specifically, in their formulation, the objective is to find the policy that maximizes the expected probability of generating feedback 1, and their optimal policy can be non-

Markovian due to the non-linearity of the sigmoid function; In our formulation, our objective is to find the optimal policy under the standard MDP definition by inferring rewards from binary feedback, and thus we consider Markovian policies. The algorithms in (Chatterji et al., 2021) are either computationally inefficient or have a suboptimal regret order due to the non-linearity of their objective and direct maximization over all non-Markovian policies. Our algorithms are computationally efficient by adopting the TS algorithmic style and efficient MDP planning under Markovian policies. Our regret results cannot be directly compared to those in (Chatterji et al., 2021) due to the difference in formulation.

Moreover, different from (Efroni et al., 2021; Chatterji et al., 2021), we study RL with segment feedback, which allows feedback from multiple segments within a trajectory, with per-state-action feedback and trajectory feedback as the two extremes. Under sum feedback, we improve the result in (Efroni et al., 2021) by a factor of $\sqrt{H}$ using experimental design, when the problem reduces to the trajectory feedback setting. Under binary feedback, we propose TS-style algorithms which are computationally efficient, and build a lower bound to reveal an inevitable exponential factor in the regret bound, which is novel to the RL literature.

Our work is also related to linear bandits (Abbasi-Yadkori et al., 2011) and logistic bandits (Filippi et al., 2010; Faury et al., 2020; Russac et al., 2021), and uses analytical techniques from that literature.

## 2. Formulation

In this section, we present the formulation of RL with binary and sum segment feedback.

We consider an episodic MDP denoted by $\mathcal{M}(\mathcal{S}, \mathcal{A}, H, r, p, \rho)$. Here $\mathcal{S}$ is the state space, and $\mathcal{A}$ is the action space. $H$ is the length of each episode. $r : \mathcal{S} \times \mathcal{A} \to [-r_{\max}, r_{\max}]$ is an unknown reward function, where $r_{\max} > 0$ is a universal constant. Define the reward parameter $\theta^* := [r(s,a)]_{(s,a) \in \mathcal{S} \times \mathcal{A}} \in \mathbb{R}^{|\mathcal{S}||\mathcal{A}|}$. $p : \mathcal{S} \times \mathcal{A} \to \triangle_{\mathcal{S}}$ is the transition distribution. For any $(s, a, s') \in \mathcal{S} \times \mathcal{A} \times \mathcal{S}$, $p(s'|s, a)$ is the probability of transitioning to $s'$ if action $a$ is taken in state $s$. $\rho \in \triangle_{\mathcal{S}}$ is an initial state distribution.

A policy $\pi : \mathcal{S} \times [H] \to \mathcal{A}$ is defined as a mapping from the state space and step indices to the action space, so that $\pi_h(s)$ specifies what action to take in state $s$ at step $h$. For any policy $\pi$, $h \in [H]$ and $(s, a) \in \mathcal{S} \times \mathcal{A}$, let $V_h^{\pi}(s)$ be the state value function, and $Q_h^{\pi}(s, a)$ be the state-action value function, which denote the cumulative expected reward obtained under policy $\pi$ till the end of an episode, starting from $s$ and $(s, a)$ at step $h$, respectively. Formally, $V_h^{\pi}(s) := \mathbb{E}[\sum_{t=h}^{H} r(s_t, a_t)|s_h = s, \pi]$, and $Q_h^{\pi}(s, a) := \mathbb{E}[\sum_{t=h}^{H} r(s_t, a_t)|s_h = s, a_h = a, \pi]$.

The optimal policy is defined as $\pi^* = \operatorname{argmax}_{\pi} V_h^{\pi}(s)$ for all $s \in \mathcal{S}$ and $h \in [H]$. For any $s \in \mathcal{S}$ and $h \in [H]$, denote $V_h^*(s) := V_h^{\pi^*}(s)$.

The process of RL with segment feedback is as follows. In each episode $k$, the agent chooses a policy $\pi^k$ at the beginning of this episode, and starts from $s_1^k \sim \rho$. At each step $h \in [H]$, the agent first observes the current state $s_h^k$, and takes an action $a_h^k = \pi_h^k(s_h^k)$ according to her policy, and then transitions to a next state $s_{h+1}^k \sim p(\cdot|s_h^k, a_h^k)$.

Each episode is equally divided into $m$ segments, and each segment is of length $\frac{H}{m}$. For convenience, assume that $H$ is divisible by $m$. For any $k > 0$ and $i \in [m]$, let $\tau^k = (s_1^k, a_1^k, \dots, s_h^k, a_h^k)$ denote the trajectory in episode $k$, and $\tau_i^k = (s_{\frac{H}{m} \cdot (i-1)+1}^k, a_{\frac{H}{m} \cdot (i-1)+1}^k, \dots, s_{\frac{H}{m} \cdot i}^k, a_{\frac{H}{m} \cdot i}^k)$ denote the $i$-th segment of the trajectory in episode $k$.

For any trajectory or trajectory segment $\tau$, $\phi^{\tau} \in \mathbb{R}^{|\mathcal{S}||\mathcal{A}|}$ denotes the vector where each entry $\phi^{\tau}(s, a)$ is the number of times $(s, a)$ is visited in $\tau$. For any policy $\pi$, $\phi^{\pi} \in \mathbb{R}^{|\mathcal{S}||\mathcal{A}|}$ denotes the vector where each entry $\phi^{\pi}(s, a)$ is the expected number of times $(s, a)$ is visited in an episode under policy $\pi$, i.e.,

$$\phi^{\pi}(s, a) := \mathbb{E}\left[\sum_{h=1}^{H} \mathbb{1}\{s_h = s, a_h = a\}\Big|\pi\right].$$

In our model, the agent observes reward feedback *only at the end of each segment*, instead of each step as in classic RL. We consider two reward feedback settings as follows.

**Binary Segment Feedback.** Denote the sigmoid function by $\mu(x) := \frac{1}{1+\exp(-x)}$ for any $x \in \mathbb{R}$. In the binary segment feedback setting, in each episode $k$, at the end of each segment $i \in [m]$, the agent observes a binary outcome

$$y_i^k = \begin{cases} 1, & \text{w.p. } \mu((\phi^{\tau_i^k})^{\top}\theta^*), \\ 0, & \text{w.p. } 1 - \mu((\phi^{\tau_i^k})^{\top}\theta^*). \end{cases}$$

Note that our formulation is different from that in prior work for binary feedback (Chatterji et al., 2021). Chatterji et al. (2021) aim to find the policy that maximizes the expected probability of generating feedback 1, i.e., $\max_{\pi} \mathbb{E}_{\tau \sim \pi, p}[\mu((\phi^{\tau})^{\top}\theta^*)]$, where the optimal policy can be non-Markovian due to the non-linearity of $\mu(\cdot)$. In contrast, we aim to find the optimal policy under the standard MDP definition, i.e., $\max_{\pi} \mathbb{E}_{\tau \sim \pi, p}[(\phi^{\tau})^{\top}\theta^*]$, by inferring reward $\theta^*$ from binary feedback, and thus we consider Markovian policies. Both formulations have value and are applicable in different contexts. In particular, our formulation is better suited to situations where we want to solve an MDP but only get binary segment feedback. Under our formulation, we design TS-type algorithms with confidence

bonuses added on $\theta^*$ element-wise to achieve computational efficiency, which cannot be done without sacrificing the regret order under the formulation of (Chatterji et al., 2021).

**Sum Segment Feedback.** In the sum segment feedback setting, in each episode $k$, at each step $h$, the environment generates an underlying random reward $R_h^k = r(s_h^k, s_h^k) + \varepsilon_h^k$, where $\varepsilon_h^k$ is a zero-mean and 1-sub-Gaussian noise, and independent of transition. At the end of each segment $i \in [m]$, the agent observes the sum of random rewards

$$R_i^k = \sum_{t=\frac{H}{m}(i-1)+1}^{\frac{H}{m} \cdot i} R(s_t^k, a_t^k) = (\phi^{\tau_i^k})^\top \theta^* + \sum_{t=\frac{H}{m}(i-1)+1}^{\frac{H}{m} \cdot i} \varepsilon_t^k.$$

Under sum feedback, when $m = H$, our model degenerates to classic RL (Azar et al., 2017; Sutton & Barto, 2018). When $m = 1$, the above two settings reduce to the problems of RL with binary (Chatterji et al., 2021) and sum trajectory feedback (Efroni et al., 2021), respectively.

In our model, the agent needs to infer the reward function from sparse and implicit reward feedback. Let $K$ denote the number of episodes played. The goal of the agent is to minimize the cumulative regret, which is defined as

$$\mathcal{R}(K) := \sum_{k=1}^K (V_1^*(s_1) - V_1^{\pi^k}(s_1)).$$

We note that to the best of our knowledge, the fact that one gets extremely coarse information about the sum reward in the binary feedback case makes it impossible to have a common analysis for both feedback models.

## 3. Reinforcement Learning with Binary Segment Feedback

In this section, we investigate RL with binary segment feedback. To isolate the effect of segment feedback from transition model learning, we first design a computationally-efficient and sample-efficient algorithm `SegBiTS` for the known transition case, and establish a novel lower bound to exhibit the indispensable exponential dependency in the result under binary feedback. Then, we further develop an algorithm `SegBiTS-Tran` with carefully-designed transition bonuses for the unknown transition case.

### 3.1. Algorithm `SegBiTS` for Known Transition

Building upon the Thompson sampling algorithm (Thompson, 1933), `SegBiTS` adopts the maximum likelihood estimator (MLE) to learn rewards from binary feedback, and performs posterior sampling to compute the optimal policy. Different from prior trajectory feedback algorithms (Chatterji et al., 2021) which are either computationally inefficient

---

**Algorithm 1** `SegBiTS`

1: **Input:** $\delta, \delta' := \frac{\delta}{3}, \alpha := \exp(\frac{Hr_{\max}}{m}) + \exp(-\frac{Hr_{\max}}{m}) + 2, \lambda$.
2: **for** $k = 1, \ldots, K$ **do**
3: $\quad \hat{\theta}_{k-1} \leftarrow \arg\min_\theta -(\sum_{k'=1}^{k-1} \sum_{i=1}^m (y_i^{k'} \cdot \log(\mu((\phi^{\tau_i^{k'}})^\top \theta)) + (1 - y_i^{k'}) \cdot \log(1 - \mu((\phi^{\tau_i^{k'}})^\top \theta))) - \frac{1}{2}\lambda\|\theta\|_2^2)$
4: $\quad \Sigma_{k-1} \leftarrow \sum_{k'=1}^{k-1} \sum_{i=1}^m \phi^{\tau_i^{k'}} (\phi^{\tau_i^{k'}})^\top + \alpha\lambda I$
5: $\quad$ Sample $\xi_k \sim \mathcal{N}(0, \alpha \cdot \nu(k-1)^2 \cdot \Sigma_{k-1}^{-1})$, where $\nu(k-1)$ is defined in Eq. (1)
6: $\quad \tilde{\theta}_k \leftarrow \hat{\theta}_{k-1} + \xi_k$
7: $\quad \pi^k \leftarrow \arg\max_\pi (\phi^\pi)^\top \tilde{\theta}_k$
8: $\quad$ Play episode $k$ with policy $\pi^k$. Observe trajectory $\tau^k$ and binary segment feedback $\{y_i^k\}_{i=1}^m$
9: **end for**

---

or have a $O(K^{\frac{2}{3}})$ regret bound, `SegBiTS` is both computationally efficient and has a $O(\sqrt{K})$ regret bound.

Algorithm 1 presents the procedure of `SegBiTS`. Specifically, in each episode $k$, `SegBiTS` first employs MLE with past binary reward observations to obtain the estimated reward parameter $\hat{\theta}_{k-1}$ (Line 3). Then, `SegBiTS` calculates the feature covariance matrix of past segments $\Sigma_{k-1}$ (Line 4). After that, `SegBiTS` samples a noise $\xi_k$ from Gaussian distribution $\mathcal{N}(0, \alpha \cdot \nu(k-1)^2 \cdot \Sigma_{k-1}^{-1})$ (Line 5). Here $\alpha$ is a universal upper bound of the inverse of the sigmoid function's derivative. For any $k > 0$, we define

$$\nu(k) := \frac{m\sqrt{\lambda}}{H}\left(1 + \frac{Hr_{\max}\sqrt{|\mathcal{S}||\mathcal{A}|}}{m} + \frac{H}{m\sqrt{\lambda}} \cdot \right.$$
$$\left. \sqrt{1 + \frac{Hr_{\max}\sqrt{|\mathcal{S}||\mathcal{A}|}}{m}\omega(k) + \frac{H^2}{m^2\lambda} \cdot \omega(k)^2}\right)^{\frac{3}{2}}, \quad (1)$$

and

$$\omega(k) := \sqrt{\lambda}\left(r_{\max}\sqrt{|\mathcal{S}||\mathcal{A}|} + \frac{1}{2}\right)$$
$$+ \frac{|\mathcal{S}||\mathcal{A}|}{\sqrt{\lambda}} \log\left(\frac{4}{\delta'}\left(1 + \frac{H^2 k}{4|\mathcal{S}||\mathcal{A}|\lambda m}\right)\right). \quad (2)$$

$\nu(k)$ is the confidence radius factor of the MLE estimate $\hat{\theta}_k$. With high probability, we have $|\phi^\top \theta^* - \phi^\top \hat{\theta}_k| \leq \sqrt{\alpha} \cdot \nu(k)\|\phi\|_{\Sigma_k^{-1}}$, where $\phi$ is the visitation indicator of any trajectory (Lemma C.7 in Appendix C.1).

Adding noise $\xi_k$ to $\hat{\theta}_{k-1}$, `SegBiTS` obtains a posterior reward estimate $\tilde{\theta}_k$ (Line 6). Then, it computes the optimal policy $\pi^k$ under reward $\tilde{\theta}_k$, i.e., $\arg\max_\pi (\phi^\pi)^\top \tilde{\theta}_k$ (Line 7). Note that this step is *computationally efficient*, which can be easily solved by any MDP planning algorithm, e.g., value iteration, by taking $\tilde{\theta}_k$ as the reward function. After obtaining $\pi^k$, `SegBiTS` plays episode $k$, and observes trajectory $\tau^k$ and binary feedback $\{y_i^k\}_{i=1}^m$ on each segment (Line 8).

Now we provide a regret upper bound for `SegBiTS`.

**Theorem 3.1.** *With probability at least* $1-\delta$, *for any* $K > 0$, *the regret of algorithm* `SegBiTS` *is bounded by*

$$\mathcal{R}(K) = \tilde{O}\left( \exp\left( \frac{Hr_{\max}}{2m} \right) \nu(K) \sqrt{|\mathcal{S}||\mathcal{A}|} \cdot \right.$$
$$\left. \left( \sqrt{Km|\mathcal{S}||\mathcal{A}| \max\left\{ \frac{H^2}{m\alpha\lambda}, 1 \right\}} + H\sqrt{\frac{K}{\alpha\lambda}} \right) \right).$$

In this result, the dependency on $|\mathcal{S}|$, $|\mathcal{A}|$ and $H$ are $|\mathcal{S}|^3$, $|\mathcal{A}|^3$ and $\exp(\frac{Hr_{\max}}{2m})H^2$, respectively. Our focus here is to reveal the exponential dependency on $\frac{Hr_{\max}}{m}$ in the regret bound under binary feedback, instead of pursuing absolute tightness of every polynomial factor. Since the exponential factor is usually the dominating factor, this result implies that as the number of segments $m$ increases, the regret decays rapidly. Thus, under binary feedback, increasing the number of segments significantly helps accelerate learning.

The intuition behind this exponential dependency is that when the reward scale $x = \frac{Hr_{\max}}{m}$ is large, the binary feedback is generated from the range where the sigmoid function $\mu(x) = \frac{1}{1+\exp(-x)}$ is flat, i.e., the derivative of the sigmoid function $\mu'(x)$ is small. Then, the generated binary feedback is likely always 0 or always 1, and it is hard to distinguish between a good action and a bad action, leading to a higher regret; On the contrary, when the reward scale $x = \frac{Hr_{\max}}{m}$ is small, the binary feedback is generated from the range where the sigmoid function $\mu(x)$ is steep, i.e., $\mu'(x)$ is large. Then, the generated binary feedback is more dispersed to be 0 or 1, and it is easier to distinguish between a good action and a bad action, leading to a lower regret. In other words, the regret bound depends on the inverse of the sigmoid function's derivative $\mu'(x) = \frac{1}{\exp(x)+\exp(-x)+2}$.

### 3.2. Regret Lower Bound for Known Transition

Below we provide a lower bound, which *firstly* demonstrates the inevitability of the exponential factor in the regret bound for RL with binary feedback.

**Theorem 3.2.** *Consider RL with binary segment feedback and known transition. There exists a distribution of instances where for any* $c_0 \in (0, \frac{1}{2})$, *when* $K \geq \exp(\frac{Hr_{\max}}{m})\frac{4|\mathcal{S}||\mathcal{A}|m}{H^2 r_{\max}^2 c_0^2}$, *the regret of any algorithm must be*

$$\Omega\left( \exp\left( \left(\frac{1}{2} - c_0\right)\frac{Hr_{\max}}{m} \right) \sqrt{|\mathcal{S}||\mathcal{A}|mK} \right).$$

Theorem 3.2 shows that under binary feedback, the exponential dependency on $\frac{Hr_{\max}}{m}$ in the result is indispensable, and the $\exp(\frac{Hr_{\max}}{2m})$ factor in Theorem 3.1 nearly matches the exponential factor in the lower bound up to an arbitrarily

small factor $c_0$ in $\exp(\cdot)$. Theorem 3.2 reveals that when the number of segments $m$ increases, the regret indeed decreases at an exponential rate. In addition, this lower bound also holds for the unknown transition case, by constructing the same problem instance as in its proof.

To the best of our knowledge, our lower bound for binary feedback and its analysis are novel in the RL literature. In the analysis, we calculate the KL divergence of Bernoulli distributions with the sigmoid function being in their parameters. Then, we employ Pinsker's inequality and the fact that $\mu'(x) = \mu(x)(1 - \mu(x))$ to build a connection between the calculated KL divergence and $\mu'(\frac{Hr_{\max}}{m})$. Since $\mu'(x) = \frac{1}{\exp(x)+\exp(-x)+2}$ contains an exponential factor, we can finally derive an exponential dependency in the lower bound. Below we give a proof sketch of Theorem 3.2, and defer a full proof to Appendix C.2.

*Proof Sketch.* Consider an instance as follows: there are $n$ bandit states $s_1, \ldots, s_n$ (i.e., there is an optimal action and multiple suboptimal actions), a good absorbing state $s_{n+1}$ and a bad absorbing state $s_{n+2}$. The agent starts from $s_1, \ldots, s_n$ with equal probability $\frac{1}{n}$. For any $i \in [n]$, in state $s_i$, under the optimal action $a_i^*$, the agent transitions to $s_{n+1}$ deterministically, and $r(s_i, a_i^*) = r_{\max}$; Under any suboptimal action $a_i^{\text{sub}}$, the agent transitions to $s_{n+2}$ deterministically, and $r(s_i, a_i^{\text{sub}}) = (1 - \varepsilon)r_{\max}$, where $\varepsilon \in (0, \frac{1}{2})$ is a parameter specified later. For all actions $a \in \mathcal{A}$, $r(s_{n+1}, a) = r_{\max}$ and $r(s_{n+2}, a) = (1 - \varepsilon)r_{\max}$.

*Figure 1.* Lower bound instance.

Then, the KL divergence of binary observations between the optimal action and suboptimal actions in an episode is

$$\sum_{i=1}^{m}\text{KL}\left( \mathcal{B}\left( \mu\left( \frac{(1-\varepsilon)Hr_{\max}}{m} \right) \right) \middle\| \mathcal{B}\left( \mu\left( \frac{Hr_{\max}}{m} \right) \right) \right)$$

$$\overset{(a)}{\leq} m \cdot \frac{\left( \mu\left( \frac{(1-\varepsilon)Hr_{\max}}{m} \right) - \mu\left( \frac{Hr_{\max}}{m} \right) \right)^2}{\mu'\left( \frac{Hr_{\max}}{m} \right)}$$

$$\overset{(b)}{\leq} m \cdot \frac{\mu'\left( \frac{(1-\varepsilon)Hr_{\max}}{m} \right)^2 \left( \varepsilon \cdot \frac{Hr_{\max}}{m} \right)^2}{\mu'\left( \frac{Hr_{\max}}{m} \right)}, \tag{3}$$

Here $\mathcal{B}(p)$ denotes the Bernoulli distribution with parameter $p$. Inequality (a) uses the facts that $\text{KL}(\mathcal{B}(p)\|\mathcal{B}(q)) \leq \frac{(p-q)^2}{q(1-q)}$ and $\mu'(x) = \mu(x)(1 - \mu(x))$. Inequality (b) is due to that $\mu'(x)$ is monotonically decreasing when $x > 0$.

Furthermore, we consider the reward scale $Hr_{\max}$ in each

episode, and the enumeration over each bandit state $s_i$ ($i \in [n]$) and each possible optimal action $a_i^* \in \mathcal{A}$ in the lower bound derivation. Then, following the analysis in (Auer et al., 2002), to learn the difference between the optimal action and suboptimal actions, the agent must suffer a regret

$$\Omega \left( H r_{\max} \sqrt{n|\mathcal{A}|} \cdot \frac{1}{\sqrt{Eq.\ (3)}} \right)$$
$$= \Omega \left( \frac{\sqrt{|\mathcal{A}|nm}}{\varepsilon} \sqrt{\frac{\mu'\left(\frac{Hr_{\max}}{m}\right)}{\mu'\left((1-\varepsilon)\frac{Hr_{\max}}{m}\right)^2}} \right).$$

Recall that $\mu'(x) = \frac{1}{\exp(x)+\exp(-x)+2}$. Let $\varepsilon = \Theta(\frac{1}{\sqrt{K}})$. For any constant $c_0 \in (0, \frac{1}{2})$, letting $K$ large enough ($\varepsilon$ small enough) to satisfy $\varepsilon \leq c_0$, then the regret is

$$\Omega \left( \sqrt{K|\mathcal{A}|nm} \sqrt{\exp\left((1-2\varepsilon)\frac{Hr_{\max}}{m}\right)} \right)$$
$$= \Omega \left( \sqrt{K|\mathcal{S}||\mathcal{A}|m} \exp\left(\left(\frac{1}{2}-c_0\right)\frac{Hr_{\max}}{m}\right) \right). \quad \square$$

### 3.3. Algorithm `SegBiTS-Tran` for Unknown Transition

Now we extend our results to the unknown transition case.

We develop an efficient algorithm `SegBiTS-Tran` for binary segment feedback and unknown transition. `SegBiTS-Tran` includes a transition bonus $p_{k-1}^{pv}$ in posterior reward estimate $\tilde{\theta}_k^b$, and replaces visitation indicator $\phi^\pi$ by its estimate $\hat{\phi}_{k-1}^\pi$. For any $(s, a)$, $\hat{\phi}_{k-1}^\pi(s, a)$ is the expected number of times $(s, a)$ is visited in an episode under policy $\pi$ on empirical MDP $\hat{p}_{k-1}$, where $\hat{p}_{k-1}$ is the empirical estimate of transition distribution $p$. Then, `SegBiTS-Tran` computes the optimal policy via $\arg\max_\pi (\hat{\phi}_{k-1}^\pi)^\top \tilde{\theta}_k^b$, which can be efficiently solved by any MDP planning algorithm with transition distribution $\hat{p}_{k-1}$ and reward $\tilde{\theta}_k^b$. We defer the details of `SegBiTS-Tran` to Appendix C.3, and present its regret performance as follows.

**Theorem 3.3.** *With probability at least $1-\delta$, for any $K > 0$, the regret of algorithm `SegBiTS-Tran` is bounded by*

$$\tilde{O}\left( \exp\left(\frac{Hr_{\max}}{2m}\right) \nu(K) \sqrt{|\mathcal{S}||\mathcal{A}|} \cdot \right.$$
$$\left( \sqrt{Km|\mathcal{S}||\mathcal{A}| \max\left\{\frac{H^2}{m\alpha\lambda}, 1\right\}} + H\sqrt{\frac{K}{\alpha\lambda}} \right)$$
$$\left. + \left(\nu(K)\sqrt{\frac{|\mathcal{S}||\mathcal{A}|}{\lambda}} + Hr_{\max}\right)|\mathcal{S}|^2|\mathcal{A}|^{\frac{3}{2}}H^{\frac{3}{2}}\sqrt{K} \right).$$

Similar to algorithm `SegBiTS` (Theorem 3.1), the regret bound of algorithm `SegBiTS-Tran` also has a factor $\exp(\frac{Hr_{\max}}{2m})$. When the number of segments $m$ increases,

the regret of `SegBiTS-Tran` significantly decreases. Compared to `SegBiTS`, the regret of `SegBiTS-Tran` has an additional polynomial term in $|\mathcal{S}|$, $|\mathcal{A}|$, $H$ and $\sqrt{K}$, which is incurred due to learning the unknown transition distribution.

## 4. Reinforcement Learning with Sum Segment Feedback

In this section, we turn to RL with sum segment feedback. Different from prior sum trajectory feedback algorithm (Efroni et al., 2021), which directly uses the least squares estimator and has a suboptimal regret bound, we develop an algorithm `E-LinUCB` for the known transition case, which adopts experimental design to perform an initial exploration and achieves a near-optimal regret with respect to $H$ and $m$. To validate the optimality, we further establish a regret lower bound. Moreover, we design an algorithm `LinUCB-Tran` equipped with a variance-aware transition bonus to handle the unknown transition case.

### 4.1. Algorithm `E-LinUCB` for Known Transition

If we regard visitation indicators $\phi^{\pi_i^k}$ as feature vectors and $\theta^*$ as the reward parameter, RL with sum segment feedback and known transition is similar to linear bandits.

Building upon the classic linear bandit algorithm `LinUCB` (Abbasi-Yadkori et al., 2011), our algorithm `E-LinUCB` performs the E-optimal design (Pukelsheim, 2006) to conduct an initial exploration. This scheme ensures sufficient coverage of the covariance matrix and further sharpens the norm under the inverse of the covariance matrix, which enables an improved regret bound over prior trajectory feedback algorithm (Efroni et al., 2021).

Algorithm 2 shows the procedure of `E-LinUCB`. Specifically, `E-LinUCB` first performs the E-optimal design to compute a distribution on policies $w^*$, which maximizes the minimum eigenvalue of the feature covariance matrix $\sum_{\pi \in \Pi} w(\pi)(\sum_{i=1}^m \mathbb{E}_{\tau_i \sim \pi}[\phi^{\tau_i}(\phi^{\tau_i})^\top])$ (Line 2). We assume that there exists a policy distribution $w$ under which this matrix is invertible. Then, `E-LinUCB` calculates the number of samples $K_0$ for initial exploration according to the optimal value of the E-optimal design (Line 3).

Then, in Line 4, `E-LinUCB` calls a rounding procedure `ROUND` (Allen-Zhu et al., 2021) to transform sampling distribution $w^*$ into discrete sampling sequence $(\pi^1, \ldots, \pi^{K_0})$, which satisfies (see Appendix B for more details of `ROUND`)

$$\left\| \left( \sum_{k=1}^{K_0} \left( \sum_{i=1}^m \mathbb{E}_{\tau_i \sim \pi_k} \left[ \phi^{\tau_i}(\phi^{\tau_i})^\top \right] \right) \right)^{-1} \right\|$$
$$\leq (1+\gamma) \left\| \left( K_0 \sum_{\pi \in \Pi} w^*(\pi) \left( \sum_{i=1}^m \mathbb{E}_{\tau_i \sim \pi} \left[ \phi^{\tau_i}(\phi^{\tau_i})^\top \right] \right) \right)^{-1} \right\|.$$

---

**Algorithm 2** E-LinUCB

---

1: **Input:** $\delta, \delta' := \frac{\delta}{3}, \lambda := \frac{H}{r_{\max}^2 m}$, rounding procedure ROUND, rounding approximation parameter $\gamma := \frac{1}{10}$. $\beta(k) :=$
$\sqrt{\frac{H|\mathcal{S}||\mathcal{A}|}{m} \log(1 + \frac{kH^2}{\lambda|\mathcal{S}||\mathcal{A}|m})} + 2\log(\frac{1}{\delta'}) + r_{\max}\sqrt{\lambda|\mathcal{S}||\mathcal{A}|}, \forall k > 0.$
2: Let $w^* \in \triangle_\Pi$ and $z^*$ be the optimal solution and optimal value of the optimization:

$$\min_{w \in \triangle_\Pi} \left\| \left( \sum_{\pi \in \Pi} w(\pi) \left( \sum_{i=1}^{m} \mathbb{E}_{\tau_i \sim \pi} \left[ \phi^{\tau_i}(\phi^{\tau_i})^\top \right] \right) \right)^{-1} \right\| \tag{4}$$

3: $K_0 \leftarrow \lceil \max\{26(1+\gamma)^2(z^*)^2 H^4 \log(\frac{2|\mathcal{S}||\mathcal{A}|}{\delta'}), \frac{|\mathcal{S}||\mathcal{A}|}{\gamma^2}\} \rceil$
4: $(\pi^1, \ldots, \pi^{K_0}) \leftarrow \text{ROUND}(\{\sum_{i=1}^{m} \mathbb{E}_{\tau_i \sim \pi} [\phi^{\tau_i}(\phi^{\tau_i})^\top]\}_{\pi \in \Pi}, w^*, \gamma, K_0)$
5: Play $K_0$ episodes with policies $\pi^1, \ldots, \pi^{K_0}$. Observe trajectories $\tau^1, \ldots, \tau^{K_0}$ and rewards $\{R_i^1\}_{i=1}^{m}, \ldots, \{R_i^{K_0}\}_{i=1}^{m}$
6: **for** $k = K_0 + 1, \ldots, K$ **do**
7: $\quad \hat{\theta}_{k-1} \leftarrow (\lambda I + \sum_{k'=1}^{k-1} \sum_{i=1}^{m} \phi^{\tau_i^{k'}}(\phi^{\tau_i^{k'}})^\top)^{-1} \sum_{k'=1}^{k-1} \sum_{i=1}^{m} \phi^{\tau_i^{k'}} R_i^{k'}$
8: $\quad \Sigma_{k-1} \leftarrow \lambda I + \sum_{k'=1}^{k-1} \sum_{i=1}^{m} \phi^{\tau_i^{k'}}(\phi^{\tau_i^{k'}})^\top$
9: $\quad \pi^k \leftarrow \text{argmax}_{\pi \in \Pi}((\phi^\pi)^\top \hat{\theta}_{k-1} + \beta(k-1) \cdot \|\phi^\pi\|_{(\Sigma_{k-1})^{-1}})$
10: $\quad$ Play episode $k$ with policy $\pi^k$. Observe trajectory $\tau^k$ and sum segment feedback $\{R_i^k\}_{i=1}^{m}$
11: **end for**

---

After that, E-LinUCB plays $K_0$ episodes with $(\pi^1, \ldots, \pi^{K_0})$ to perform initial exploration (Line 5). Owing to the E-optimal design, the covariance matrix of initial exploration $\Sigma_{K_0}$ has an optimized minimum eigenvalue, and then $\|\phi^\pi\|_{(\Sigma_{k-1})^{-1}}$ has a sharp upper bound for any $k > K_0$. This is the key to the optimality of E-LinUCB.

In each episode $k > K_0$, E-LinUCB first calculates the least squares reward estimate $\hat{\theta}_{k-1}$ using past reward observations and covariance matrix $\Sigma_{k-1}$ (Lines 7-8). Then, it computes the optimal policy with reward estimate $\hat{\theta}_{k-1}$ and reward confidence bonus $\|\phi^\pi\|_{(\Sigma_{k-1})^{-1}}$ (Line 9). E-LinUCB plays episode $k$ with the computed optimal policy $\pi^k$, and collects trajectory $\tau^k$ and reward observations on each segment $\{R_i^k\}_{i=1}^{m}$ (Line 10). Below we present a regret upper bound for algorithm E-LinUCB.

**Theorem 4.1.** *With probability at least $1-\delta$, for any $K > 0$, the regret of algorithm* E-LinUCB *is bounded by*

$$O\left( |\mathcal{S}||\mathcal{A}|\sqrt{HK} \log \left( \left(1 + \frac{KHr_{\max}}{|\mathcal{S}||\mathcal{A}|m}\right) \frac{1}{\delta} \right) \right.$$
$$\left. + (z^*)^2 H^5 \log \left( \frac{|\mathcal{S}||\mathcal{A}|}{\delta} \right) + |\mathcal{S}||\mathcal{A}|H \right).$$

Surprisingly, under sum feedback, when the number of segments $m$ increases, the regret bound does not decrease significantly, e.g., at a rate of $\frac{1}{\sqrt{m}}$ or $\frac{1}{m}$. While this looks surprising at the first glance, we discover an *intuition* through analysis: The performance in RL is measured by the expected reward sum of an episode, namely, we only need to accurately estimate the expected reward sum of an episode.

When the number of segments $m$ increases, while we obtain more observations, the segment features $\phi^{\tau_i^{k'}}$ contributed to covariance matrix $\Sigma_k$ shrink, which makes the reward estimation uncertainty $\|\phi^\pi\|_{(\Sigma_k)^{-1}}$ inflate. When we focus on the estimation performance of the expected reward sum of an episode, these two effects cancel out with each other, and the regret result is not influenced by $m$ distinctly.

When $m = 1$, our problem reduces to RL with sum trajectory feedback (Efroni et al., 2021), and our result *improves* theirs by a factor of $\sqrt{H}$ and achieves the optimality with respect to $H$. This improvement comes from the fact that we conduct the E-optimal design and perform an initial exploration to guarantee that $\|\phi^\pi\|_{(\Sigma_{k-1})^{-1}} \leq 1$, instead of $\|\phi^\pi\|_{(\Sigma_{k-1})^{-1}} \leq \frac{H}{\sqrt{\lambda}}$ as used in (Efroni et al., 2021).

Next, we study the lower bound to see if the number of segments $m$ really does not influence the regret bound much.

### 4.2. Regret Lower Bound for Known Transition

We establish a lower bound for RL with sum segment feedback and known transition as follows.

**Theorem 4.2.** *Consider RL with sum segment feedback and known transition. There exists a distribution of instances where the regret of any algorithm must be*

$$\Omega\left( \sqrt{|\mathcal{S}||\mathcal{A}|HK} \right).$$

Theorem 4.2 demonstrates that our regret upper bound for algorithm E-LinUCB (Theorem 4.1) is optimal with respect to $H$ and $m$ when ignoring logarithmic factors. In addition, this lower bound corroborates that the number of segments

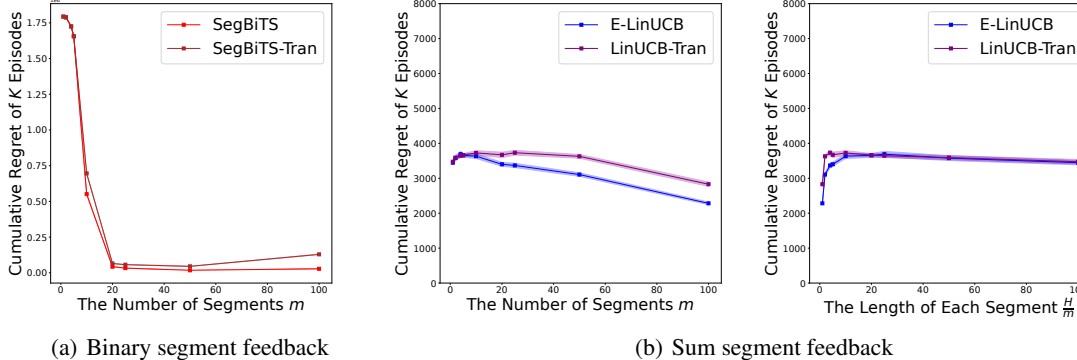

*Figure 2.* Experimental results for RL with binary or sum segment feedback.

$m$ does not impact the regret result in essence.

### 4.3. Algorithm `LinUCB-Tran` for Unknown Transition

Now we investigate RL with sum segment feedback in the unknown transition case.

For unknown transition, we design an algorithm `LinUCB-Tran`, which establishes a variance-aware uncertainty bound for the estimated visitation indicator $\hat{\phi}_k^\pi$, and incorporates this uncertainty bound into exploration bonuses. In analysis, we handle the estimation error of visitation indicators $\|\hat{\phi}_k^\pi - \phi^\pi\|_1$ by this variance-aware uncertainty bound, which enables us to achieve a near-optimal regret in terms of $H$. The details of `LinUCB-Tran` are deferred to Appendix D.3, and we state the regret performance of algorithm `LinUCB-Tran` below.

**Theorem 4.3.** *With probability at least $1-\delta$, for any $K > 0$, the regret of algorithm `LinUCB-Tran` is bounded by*

$$\tilde{O}\left((1 + r_{\max})|\mathcal{S}|^{\frac{5}{2}}|\mathcal{A}|^2 H\sqrt{K}\right).$$

Theorem 4.3 shows that similar to algorithm `E-LinUCB`, the regret of `LinUCB-Tran` does not depend on the number of segments $m$ when ignoring logarithmic factors. The heavier dependency on $|\mathcal{S}|$, $|\mathcal{A}|$ and $H$ is due to the estimation of the unknown transition distribution. We also provide a lower bound for the unknown transition case, which demonstrates that the optimal regret indeed does not depend on $m$ and our upper bound is near-optimal with respect to $H$ (see Appendix D.5).

## 5. Experiments

Below we present experiments for RL with segment feedback to validate our theoretical results.

For the binary feedback setting, we evaluate our algorithms

SegBiTS and SegBiTS-Tran in known and unknown transition cases, respectively, and we set $|\mathcal{S}| = 9$, $|\mathcal{A}| = 5$ and $K = 30000$. For the sum feedback setting, similarly, we run our algorithms E-LinUCB and LinUCB-Tran in known and unknown transition cases, respectively. Since E-LinUCB and LinUCB-Tran are computationally inefficient (mainly designed to reveal the optimal dependency on $m$), we use a small MDP with $|\mathcal{S}| = 3$ and $|\mathcal{A}| = 5$, and set $K = 1000$. The details of the instances used in our experiments are described in Appendix A. In both settings, we set $r_{\max} = 0.5$, $\delta = 0.005$, $H = 100$ and $m \in \{1, 2, 4, 5, 10, 20, 25, 50, 100\}$. For each algorithm, we perform 20 independent runs, and plot the average cumulative regret up to episode $K$ across runs with a 95% confidence interval.

Figure 2(a) reports the regrets of algorithms SegBiTS and SegBiTS-Tran under binary feedback. One sees that as the number of segments $m$ increases, the regret decreases rapidly. Specifically, when $m$ decreases from 20 to 1, i.e., $\frac{H}{2m}$ increases from $\exp(2.5)$ to $\exp(50)$, the regret grows explosively. This matches our theoretical results, i.e., Theorems 3.1 and 3.3, which show a dependency on $\exp(\frac{Hr_{\max}}{2m})$.

Figure 2(b) plots the regrets of algorithms E-LinUCB and LinUCB-Tran under sum feedback. To see the impact of segments on regrets clearly, here we show the regrets with respect to the number of segments $m$ and the length of each segment $\frac{H}{m}$ in the left and right subfigures, respectively. In the left subfigure, when $m$ increases, the regrets almost keep the same for small $m$ and slightly decrease for large $m$. To see the dependency on $m$ more clearly, we turn to the right subfigure: When the length of each segment $\frac{H}{m}$ increases, the regrets slightly increase in a logarithmic trend. This also matches our theoretical bounds in Theorems 4.1 and 4.3, which do not depend on $m$ except for the $\log(\frac{H}{m})$ factor.

# 6. Conclusion

In this work, we formulate a model named RL with segment feedback, which offers a general paradigm for feedback, bridging the gap between per-state-action feedback in classic RL and trajectory feedback. In the binary feedback setting, we deign efficient algorithms `SegBiTS` and `SegBiTS-Tran`, and provide regret upper and lower bounds which show a dependency on $\exp(\frac{Hr_{\max}}{2m})$. These results reveal that under binary feedback, increasing the number of segments $m$ greatly helps expedite learning. In the sum feedback setting, we develop near-optimal algorithms `E-LinUCB` and `LinUCB-Tran` in terms of $H$ and $m$, where the regret results do not depend on $m$ when ignoring logarithmic factors. These results exhibit that under sum feedback, increasing $m$ does not help accelerate learning much.

There are several interesting directions worth further investigation. One direction is to consider segments of unequal lengths and study how to divide segments to optimize learning. The variable segment length will affect the noise variance of reward feedback, and the sum analysis of segment visitation indicators. More advanced techniques are needed to handle these challenges. Another direction is to generalize the results to the function approximation setting. Since our analysis is based on the fact that segment reward feedback is generated linearly with respect to visitation indicators $\phi(s,a)$, we believe that generalizing $\phi(s,a)$ from a visitation indicator in the tabular setting to a feature vector in the linear function approximation setting is viable.

## Acknowledgements

The work of Yihan Du and R. Srikant is supported by NSF grants CNS 23-12714, CCF 22-07547, CNS 21-06801 and AFOSR grant FA9550-24-1-0002.

## Impact Statement

This paper presents work whose goal is to advance the field of Machine Learning. There are many potential societal consequences of our work, none which we feel must be specifically highlighted here.

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

# Appendix

## A. Details of the Experimental Setup

In this section, we detail the instances used in our experiments.

For the binary segment feedback setting, we consider an MDP as in Figure 2(a): There are 9 states and 5 actions. For any $a \in \mathcal{A}$, we have $r(s_0, a) = 0$, $r(s_i, a) = r_{\max}$ for any $i \in \{1, 3, 5, 7\}$ (called good states), and $r(s_i, a) = -r_{\max}$ for any $i \in \{2, 4, 6, 8\}$ (called bad states). There is an optimal action $a^*$ and four suboptimal actions $a^{\text{sub}}$ for all states. The agent starts from an initial state $s_0$. For any $0 \leq i \leq 6$, in state $s_i$, under the optimal action $a^*$, the agent transitions to the good state and bad state at the next horizon with probabilities $0.9$ and $0.1$, respectively; Under the suboptimal action $a^{\text{sub}}$, the agent transitions to the good state and bad state at the next horizon with probabilities $0.1$ and $0.9$, respectively. In $s_7$ or $s_8$, under the optimal action $a^*$, the agent transitions to $s_1$ and $s_2$ with probabilities $0.9$ and $0.1$, respectively; Under the suboptimal action $a^{\text{sub}}$, the agent transitions to $s_1$ and $s_2$ with probabilities $0.1$ and $0.9$, respectively.

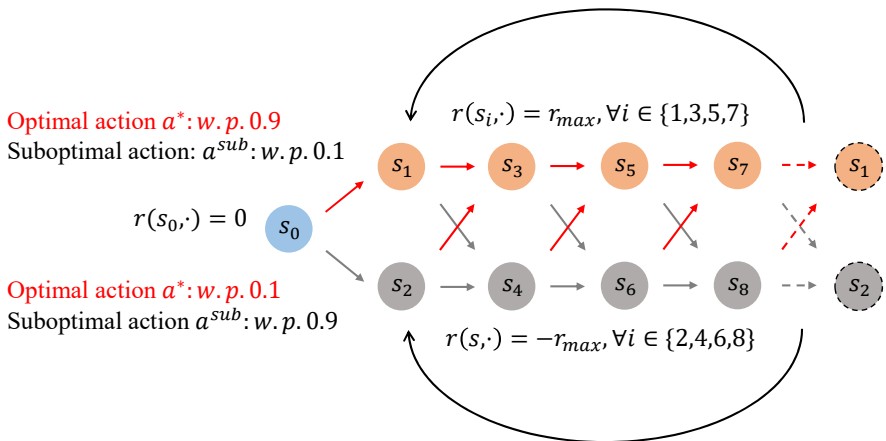

*Figure 3.* Instance used in the experiment for RL with binary segment feedback.

For the sum segment feedback setting, since algorithms E-LinUCB and LinUCB-Tran are computationally inefficient (which are mainly designed for revealing the optimal dependency on $H$ and $m$), we consider a smaller MDP as in Figure 2(b): There are 3 states and 5 actions. For any $a \in \mathcal{A}$, we have $r(s_0, a) = 0$, $r(s_1, a) = r_{\max}$ (called a good state), and $r(s_2, a) = -r_{\max}$ (called a bad state). There is an optimal action $a^*$ and four suboptimal actions $a^{\text{sub}}$ for all states. The agent starts from an initial state $s_0$. In any state $s \in \mathcal{S}$, under the optimal action $a^*$, the agent transitions to $s_1$ and $s_2$ with probabilities $0.9$ and $0.1$, respectively; Under the suboptimal action $a^{\text{sub}}$, the agent transitions to $s_1$ and $s_2$ with probabilities $0.1$ and $0.9$, respectively.

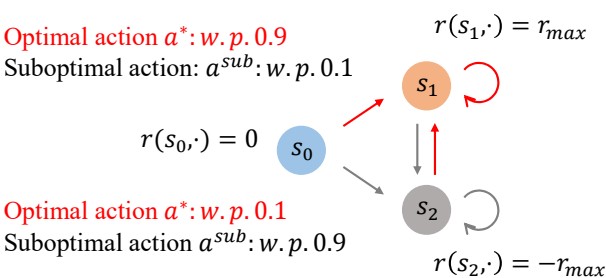

*Figure 4.* Instance used in the experiment for RL with sum segment feedback.

# B. Rounding Procedure ROUND

Algorithm E-LinUCB calls a rounding procedure ROUND (Allen-Zhu et al., 2021) in the experimental design literature. Taking $X_1, \ldots, X_n \in \mathbb{S}_+^d$, distribution $w \in \triangle_{\{X_1, \ldots, X_n\}}$, rounding approximation error $\gamma > 0$ and the number of samples $T \geq \frac{d}{\gamma^2}$ as inputs, ROUND rounds sampling distribution $w$ into a discrete sampling sequence $(Y_1, \ldots, Y_T) \in \{X_1, \ldots, X_n\}^T$ that satisfies

$$\left\| \left( \sum_{t=1}^{T} Y_t \right)^{-1} \right\| \leq (1 + \gamma) \left\| \left( T \sum_{i \in [n]} w(X_i) X_i \right)^{-1} \right\|.$$

In implementation, we can regard $xx^\top$ in (Allen-Zhu et al., 2021) as $\sum_{i=1}^{m} \mathbb{E}_{\tau_i \sim \pi}[\phi^{\tau_i}(\phi^{\tau_i})^\top]$, and regard sampling weight on $x$ as the sampling weight on $\pi$ in our work.

# C. Proofs for RL with Binary Segment Feedback

In this section, we present the proofs for RL with binary segment feedback.

## C.1. Proof for the Regret Upper Bound with Known Transition

First, we prove the regret upper bound (Theorem 3.1) of algorithm SegBiTS for known transition.

For any $k > 0$ and $\theta \in \Theta$, define

$$Z_k := \sum_{k'=1}^{k} \sum_{i=1}^{m} \varepsilon_{k',i} \cdot \phi^{\tau_i^{k'}},$$

$$g_k(\theta) := \sum_{k'=1}^{k} \sum_{i=1}^{m} \mu((\phi^{\tau_i^{k'}})^\top \theta) \cdot \phi^{\tau_i^{k'}} + \lambda\theta, \tag{5}$$

$$\Lambda_k(\theta) := \sum_{k'=1}^{k} \sum_{i=1}^{m} \mu'((\phi^{\tau_i^{k'}})^\top \theta) \cdot \phi^{\tau_i^{k'}}(\phi^{\tau_i^{k'}})^\top + \lambda I. \tag{6}$$

**Lemma C.1.** *For any $k > 0$ and $\theta \in \Theta$, we have*

$$\det(\Lambda_k(\theta)) \leq \left( \frac{H^2 \mu'_{\max} k}{|\mathcal{S}||\mathcal{A}|m} + \lambda \right)^{|\mathcal{S}||\mathcal{A}|}.$$

*Proof.* For any $k > 0$, we have

$$\det(\Lambda_k(\theta)) \leq \left( \frac{\text{tr}(\Lambda_k(\theta))}{|\mathcal{S}||\mathcal{A}|} \right)^{|\mathcal{S}||\mathcal{A}|}$$

$$\leq \left( \frac{1}{|\mathcal{S}||\mathcal{A}|} \cdot \left( km \cdot \mu'_{\max} \cdot \left( \frac{H}{m} \right)^2 + \lambda|\mathcal{S}||\mathcal{A}| \right) \right)^{|\mathcal{S}||\mathcal{A}|}$$

$$= \left( \frac{H^2 \mu'_{\max} k}{|\mathcal{S}||\mathcal{A}|m} + \lambda \right)^{|\mathcal{S}||\mathcal{A}|}.$$

$\square$

For any $k > 0$, let $F_k$ denote the filtration that includes all events up to the end of episode $k$, and $\tilde{F}_k$ denote the filtration that includes all events before playing $\pi^k$ in episode $k$. Then, $\pi^k$ is $\tilde{F}_k$-measurable.

For any $k > 0$ and $i \in [m]$, let $\varepsilon_{k,i} := y_i^k - (\phi^{\tau_i^k})^\top \theta^*$ denote the noise of binary feedback, and $v_{k,i}^2 := \mathbb{E}[\varepsilon_{k,i}^2 | \tilde{F}_k] = (\phi^{\tau_i^k})^\top \theta^* \cdot (1 - (\phi^{\tau_i^k})^\top \theta^*) = \mu'((\phi^{\tau_i^k})^\top \theta^*)$ denote the variance of $\varepsilon_{k,i}$ conditioning on $\tilde{F}_k$.

Then, we have

$$\Lambda_k(\theta^*) := \sum_{k'=1}^{k} \sum_{i=1}^{m} \mu'((\phi_i^{\tau_i^{k'}})^\top \theta^*) \cdot \phi_i^{\tau_i^{k'}} (\phi_i^{\tau_i^{k'}})^\top + \lambda I$$

$$= \sum_{k'=1}^{k} \sum_{i=1}^{m} v_{k',i}^2 \cdot \phi_i^{\tau_i^{k'}} (\phi_i^{\tau_i^{k'}})^\top + \lambda I.$$

**Lemma C.2** (Concentration of Noises under Binary Feedback). *With probability at least $1 - \delta'$, for any $k > 0$,*

$$\left\| \sum_{k'=1}^{k} \sum_{i=1}^{m} \varepsilon_{k',i} \cdot \phi_i^{\tau_i^{k'}} \right\|_{\Lambda_k^{-1}(\theta^*)} \le \frac{\sqrt{\lambda}}{2} + \frac{|\mathcal{S}||\mathcal{A}|}{\sqrt{\lambda}} \log \left( \frac{4}{\delta'} \cdot \left( 1 + \frac{H^2 \mu'_{\max} k}{|\mathcal{S}||\mathcal{A}| m \lambda} \right) \right).$$

*Proof.* According to Theorem 1 in (Faury et al., 2020), we have that with probability at least $1 - \delta'$, for any $k > 0$,

$$\left\| \sum_{k'=1}^{k} \sum_{i=1}^{m} \varepsilon_{k',i} \cdot \phi_i^{\tau_i^{k'}} \right\|_{\Lambda_k^{-1}(\theta^*)} \le \frac{\sqrt{\lambda}}{2} + \frac{2}{\sqrt{\lambda}} \log \left( \frac{\det(\Lambda_k(\theta^*))^{\frac{1}{2}} \cdot \lambda^{-\frac{|\mathcal{S}||\mathcal{A}|}{2}}}{\delta'} \right) + \frac{2}{\sqrt{\lambda}} |\mathcal{S}||\mathcal{A}| \log(2)$$

$$\overset{(a)}{\le} \frac{\sqrt{\lambda}}{2} + \frac{2}{\sqrt{\lambda}} \log \left( \frac{1}{\delta'} \left( 1 + \frac{H^2 \mu'_{\max} k}{|\mathcal{S}||\mathcal{A}| m \lambda} \right)^{\frac{|\mathcal{S}||\mathcal{A}|}{2}} \right) + \frac{2}{\sqrt{\lambda}} |\mathcal{S}||\mathcal{A}| \log(2)$$

$$\le \frac{\sqrt{\lambda}}{2} + \frac{|\mathcal{S}||\mathcal{A}|}{\sqrt{\lambda}} \log \left( \frac{1}{\delta'} \left( 1 + \frac{H^2 \mu'_{\max} k}{|\mathcal{S}||\mathcal{A}| m \lambda} \right) \right) + \frac{2}{\sqrt{\lambda}} |\mathcal{S}||\mathcal{A}| \log(2)$$

$$\le \frac{\sqrt{\lambda}}{2} + \frac{|\mathcal{S}||\mathcal{A}|}{\sqrt{\lambda}} \log \left( \frac{4}{\delta'} \left( 1 + \frac{H^2 \mu'_{\max} k}{|\mathcal{S}||\mathcal{A}| m \lambda} \right) \right),$$

where (a) uses Lemma C.1. $\qquad \square$

Define event

$$\mathcal{E} := \left\{ \left\| g_k(\hat{\theta}_k) - g_k(\theta^*) \right\|_{\Lambda_k^{-1}(\theta^*)} \le \omega(k), \ \forall k > 0 \right\}.$$

**Lemma C.3.** *It holds that*

$$\Pr[\mathcal{E}] \ge 1 - \delta'.$$

*Proof.* This proof is similar to that for Lemma 8 in (Faury et al., 2020).

Define

$$\mathcal{L}_k(\theta) := - \left( \sum_{k'=1}^{k} \sum_{i=1}^{m} \left( y_i^{k'} \cdot \log \left( \mu((\phi_i^{\tau_i^{k'}})^\top \theta) \right) + (1 - y_i^{k'}) \cdot \log \left( 1 - \mu((\phi_i^{\tau_i^{k'}})^\top \theta) \right) \right) - \frac{1}{2} \lambda \|\theta\|_2^2 \right).$$

Recall that $\hat{\theta}_k = \operatorname{argmin}_\theta \mathcal{L}_k(\theta)$. Using the facts that $\nabla \mathcal{L}_k(\hat{\theta}_k) = 0$ and $\mu'(x) = \mu(x)(1 - \mu(x))$, we have

$$\underbrace{\sum_{k'=1}^{k} \sum_{i=1}^{m} \mu((\phi_i^{\tau_i^{k'}})^\top \hat{\theta}_k) \cdot \phi_i^{\tau_i^{k'}} + \lambda \hat{\theta}_k}_{g_k(\hat{\theta}_k)} = \sum_{k'=1}^{k} \sum_{i=1}^{m} y_i^{k'} \cdot \phi_i^{\tau_i^{k'}}.$$

Hence, we have

$$g_k(\hat{\theta}_k) - g_k(\theta^*) = \sum_{k'=1}^{k} \sum_{i=1}^{m} y_i^{k'} \cdot \phi_i^{\tau_i^{k'}} - \left( \sum_{k'=1}^{k} \sum_{i=1}^{m} \mu((\phi_i^{\tau_i^{k'}})^\top \theta^*) \cdot \phi_i^{\tau_i^{k'}} + \lambda \theta^* \right)$$

$$= \sum_{k'=1}^{k} \sum_{i=1}^{m} \varepsilon_{k',i} \cdot \phi^{\tau_i^{k'}} - \lambda \theta^*. \tag{7}$$

Then, using Lemma C.2, we have that with probability at least $1 - \delta'$, for any $k > 0$,

$$\left\| g_k(\hat{\theta}_k) - g_k(\theta^*) \right\|_{\Lambda_k^{-1}(\theta^*)} \leq \left\| \sum_{k'=1}^{k} \sum_{i=1}^{m} \varepsilon_{k',i} \cdot \phi^{\tau_i^{k'}} \right\|_{\Lambda_k^{-1}(\theta^*)} + r_{\max} \sqrt{\lambda |\mathcal{S}||\mathcal{A}|}$$

$$\leq \frac{\sqrt{\lambda}}{2} + \frac{|\mathcal{S}||\mathcal{A}|}{\sqrt{\lambda}} \log \left( \frac{4}{\delta'} \cdot \left( 1 + \frac{H^2 \mu'_{\max} k}{|\mathcal{S}||\mathcal{A}|m\lambda} \right) \right) + r_{\max} \sqrt{\lambda |\mathcal{S}||\mathcal{A}|}$$

$$= \omega(k).$$

$\square$

For any $\phi \in \mathbb{R}^{|\mathcal{S}||\mathcal{A}|}$ and $\theta_1, \theta_2 \in \Theta$, define

$$b(\phi, \theta_1, \theta_2) := \int_{z=0}^{1} \mu'((1-z) \cdot \phi^\top \theta_1 + z \cdot \phi^\top \theta_2) dz.$$

For any $k > 0$ and $\theta_1, \theta_2 \in \Theta$, define

$$\Gamma_k(\theta_1, \theta_2) := \sum_{k'=1}^{k} \sum_{i=1}^{m} b(\phi, \theta_1, \theta_2) \cdot \phi^{\tau_i^{k'}} (\phi^{\tau_i^{k'}})^\top + \lambda I.$$

In the definitions of $b(\phi, \theta_1, \theta_2)$ and $\Gamma_k(\theta_1, \theta_2)$, $\theta_1$ and $\theta_2$ have the same roles and can be interchanged.

Recall that

$$\alpha := \exp(\frac{H r_{\max}}{m}) + \exp(-\frac{H r_{\max}}{m}) + 2.$$

Then, we have

$$\sup_{\tau^{\mathrm{seg}}, \theta} \frac{1}{\mu'((\phi^{\tau^{\mathrm{seg}}})^\top \theta)} \leq \alpha,$$

where $\tau^{\mathrm{seg}}$ denotes the visitation indicator of any possible trajectory segment.

**Lemma C.4.** *For any $k \geq 1$ and $\theta \in \Theta$, we have*

$$\Sigma_k \preceq \alpha \Lambda_k(\theta).$$

*Proof.* We have

$$\frac{1}{\alpha} = \inf_{\tau^{\mathrm{seg}}, \theta} \mu'((\phi^{\tau^{\mathrm{seg}}})^\top \theta).$$

Then, it holds that

$$\Sigma_k = \sum_{k'=1}^{k} \sum_{i=1}^{m} \phi^{\tau_i^{k'}} (\phi^{\tau_i^{k'}})^\top + \alpha \lambda I$$

$$= \alpha \left( \sum_{k'=1}^{k} \sum_{i=1}^{m} \frac{1}{\alpha} \cdot \phi^{\tau_i^{k'}} (\phi^{\tau_i^{k'}})^\top + \lambda I \right)$$

$$\preceq \alpha \left( \sum_{k'=1}^{k} \sum_{i=1}^{m} \mu'((\phi^{\tau_i^{k'}})^\top \theta) \cdot \phi^{\tau_i^{k'}} (\phi^{\tau_i^{k'}})^\top + \lambda I \right)$$

$$= \alpha \Lambda_k(\theta).$$

$\square$

**Lemma C.5.** *For any $\phi \in \mathbb{R}^{|\mathcal{S}||\mathcal{A}|}$ and $\theta_1, \theta_2 \in \Theta$, we have*

$$\mu(\phi^\top \theta_1) - \mu(\phi^\top \theta_2) = b(\phi, \theta_2, \theta_1) \cdot \phi^\top (\theta_1 - \theta_2).$$

*In addition, for any $k > 0$ and $\theta_1, \theta_2 \in \Theta$, we have*

$$\|\theta_1 - \theta_2\|_{\Gamma_k(\theta_2, \theta_1)} = \|g_k(\theta_1) - g_k(\theta_2)\|_{\Gamma_k^{-1}(\theta_2, \theta_1)}.$$

*Proof.* The first statement follows from the mean-value theorem.

Then, using the first statement, we have that for any $k > 0$,

$$g_k(\theta_1) - g_k(\theta_2) = \sum_{k'=1}^{k} \sum_{i=1}^{m} \left( \mu((\phi_i^{\tau^{k'}})^\top \theta_1) - \mu((\phi_i^{\tau^{k'}})^\top \theta_2) \right) \cdot \phi_i^{\tau^{k'}} + \lambda (\theta_1 - \theta_2)$$

$$= \sum_{k'=1}^{k} \sum_{i=1}^{m} b(\phi_i^{\tau^{k'}}, \theta_2, \theta_1) \cdot \phi_i^{\tau^{k'}} (\phi_i^{\tau^{k'}})^\top (\theta_1 - \theta_2) + \lambda (\theta_1 - \theta_2)$$

$$= \Gamma_k(\theta_2, \theta_1) \cdot (\theta_1 - \theta_2),$$

and thus

$$\|\theta_1 - \theta_2\|_{\Gamma_k(\theta_2, \theta_1)} = \sqrt{(\theta_1 - \theta_2)^\top \cdot \Gamma_k(\theta_2, \theta_1) \cdot (\theta_1 - \theta_2)}$$

$$= \sqrt{(\theta_1 - \theta_2)^\top \cdot \Gamma_k(\theta_2, \theta_1) \cdot \Gamma_k^{-1}(\theta_2, \theta_1) \cdot \Gamma_k(\theta_2, \theta_1) \cdot (\theta_1 - \theta_2)}$$

$$= \|g_k(\theta_1) - g_k(\theta_2)\|_{\Gamma_k^{-1}(\theta_2, \theta_1)},$$

which gives the second statement. $\qquad\square$

Recall that for any $k > 0$, $Z_k := \sum_{k'=1}^{k} \sum_{i=1}^{m} \varepsilon_{k', i} \cdot \phi_i^{\tau^{k'}}$.

**Lemma C.6.** *For any $k > 0$, we have*

$$\Gamma_k(\theta^*, \hat{\theta}_k) \succeq \left( 1 + \frac{H r_{\max} \sqrt{|\mathcal{S}||\mathcal{A}|}}{m} + \frac{H}{m\sqrt{\lambda}} \|Z_k\|_{\Gamma_k^{-1}(\theta^*, \hat{\theta}_k)} \right)^{-1} \Lambda_k(\theta^*),$$

$$\|Z_k\|_{\Gamma_k^{-1}(\theta^*, \hat{\theta}_k)} \leq \sqrt{1 + \frac{H r_{\max} \sqrt{|\mathcal{S}||\mathcal{A}|}}{m}} \|Z_k\|_{\Lambda_k^{-1}(\theta^*)} + \frac{H}{m\sqrt{\lambda}} \|Z_k\|_{\Lambda_k^{-1}(\theta^*)}^2.$$

*Furthermore, assuming that event $\mathcal{E}$ holds, we have*

$$\|Z_k\|_{\Gamma_k^{-1}(\theta^*, \hat{\theta}_k)} \leq \sqrt{1 + \frac{H r_{\max} \sqrt{|\mathcal{S}||\mathcal{A}|}}{m}} \cdot \omega(k) + \frac{H}{m\sqrt{\lambda}} \cdot \omega(k)^2.$$

*Proof.* This proof follows the analysis of Proposition 6 and Corollary 5 in (Russac et al., 2021).

From Eq. (7), we have that for any $k > 0$,

$$g_k(\hat{\theta}_k) - g_k(\theta^*) = Z_k - \lambda \theta^*.$$

Using Lemma E.1, we have that for any $\phi \in \mathbb{R}^{|\mathcal{S}||\mathcal{A}|}$ such that $\|\phi\|_2 \leq L_\phi$,

$$b(\phi, \theta^*, \hat{\theta}_k) \geq \left( 1 + \left| \phi^\top (\theta^* - \hat{\theta}_k) \right| \right)^{-1} \mu'(\phi^\top \theta^*)$$

$$= \left( 1 + \left| \phi^\top \Gamma_k^{-1}(\theta^*, \hat{\theta}_k) \cdot (g_k(\theta^*) - g_k(\hat{\theta}_k)) \right| \right)^{-1} \mu'(\phi^\top \theta^*)$$

$$\geq \left( 1 + \|\phi\|_{\Gamma_k^{-1}(\theta^*, \hat{\theta}_k)} \left\| g_k(\theta^*) - g_k(\hat{\theta}_k) \right\|_{\Gamma_k^{-1}(\theta^*, \hat{\theta}_k)} \right)^{-1} \mu'(\phi^\top \theta^*)$$

$$\geq \left( 1 + \frac{L_\phi}{\sqrt{\lambda}} \left\| g_k(\theta^*) - g_k(\hat{\theta}_k) \right\|_{\Gamma_k^{-1}(\theta^*, \hat{\theta}_k)} \right)^{-1} \mu'(\phi^\top \theta^*)$$

$$= \left( 1 + \frac{L_\phi}{\sqrt{\lambda}} \left\| Z_k - \lambda\theta^* \right\|_{\Gamma_k^{-1}(\theta^*, \hat{\theta}_k)} \right)^{-1} \mu'(\phi^\top \theta^*)$$

$$\geq \left( 1 + L_\phi r_{\max} \sqrt{|\mathcal{S}||\mathcal{A}|} + \frac{L_\phi}{\sqrt{\lambda}} \left\| Z_k \right\|_{\Gamma_k^{-1}(\theta^*, \hat{\theta}_k)} \right)^{-1} \mu'(\phi^\top \theta^*).$$

Using the above equation with $\phi = \phi^{\tau_i^{k'}}$ and $L_\phi = \frac{H}{m}$, we have

$$\Gamma_k(\theta^*, \hat{\theta}_k) := \sum_{k'=1}^{k} \sum_{i=1}^{m} b(\phi^{\tau_i^{k'}}, \theta^*, \hat{\theta}_k) \cdot \phi^{\tau_i^{k'}} (\phi^{\tau_i^{k'}})^\top + \lambda I$$

$$\succeq \sum_{k'=1}^{k} \sum_{i=1}^{m} \left( 1 + \frac{H r_{\max} \sqrt{|\mathcal{S}||\mathcal{A}|}}{m} + \frac{H}{m\sqrt{\lambda}} \left\| Z_k \right\|_{\Gamma_k^{-1}(\theta^*, \hat{\theta}_k)} \right)^{-1} \mu'(\phi^\top \theta^*) \cdot \phi^{\tau_i^{k'}} (\phi^{\tau_i^{k'}})^\top + \lambda I$$

$$= \left( 1 + \frac{H r_{\max} \sqrt{|\mathcal{S}||\mathcal{A}|}}{m} + \frac{H}{m\sqrt{\lambda}} \left\| Z_k \right\|_{\Gamma_k^{-1}(\theta^*, \hat{\theta}_k)} \right)^{-1} \Lambda_k(\theta^*).$$

This implies

$$\|Z_k\|_{\Gamma_k^{-1}(\theta^*, \hat{\theta}_k)}^2 \leq \left( 1 + \frac{H r_{\max} \sqrt{|\mathcal{S}||\mathcal{A}|}}{m} + \frac{H}{m\sqrt{\lambda}} \left\| Z_k \right\|_{\Gamma_k^{-1}(\theta^*, \hat{\theta}_k)} \right) \|Z_k\|_{\Lambda_k^{-1}(\theta^*)}^2,$$

which is equivalent to

$$\|Z_k\|_{\Gamma_k^{-1}(\theta^*, \hat{\theta}_k)}^2 - \frac{H}{m\sqrt{\lambda}} \|Z_k\|_{\Lambda_k^{-1}(\theta^*)}^2 \|Z_k\|_{\Gamma_k^{-1}(\theta^*, \hat{\theta}_k)} - \left( 1 + \frac{H r_{\max} \sqrt{|\mathcal{S}||\mathcal{A}|}}{m} \right) \|Z_k\|_{\Lambda_k^{-1}(\theta^*)}^2 \leq 0.$$

By analysis of quadratic functions, we have

$$\|Z_k\|_{\Gamma_k^{-1}(\theta^*, \hat{\theta}_k)} \leq \sqrt{1 + \frac{H r_{\max} \sqrt{|\mathcal{S}||\mathcal{A}|}}{m}} \|Z_k\|_{\Lambda_k^{-1}(\theta^*)} + \frac{H}{m\sqrt{\lambda}} \|Z_k\|_{\Lambda_k^{-1}(\theta^*)}^2$$

$$\leq \sqrt{1 + \frac{H r_{\max} \sqrt{|\mathcal{S}||\mathcal{A}|}}{m}} \cdot \omega(k) + \frac{H}{m\sqrt{\lambda}} \cdot \omega(k)^2.$$

$\square$

**Lemma C.7** (Concentration of $\phi^\top \hat{\theta}_k$ under Binary Feedback). *Assume that event $\mathcal{E}$ holds. Then, for any $k > 0$ and $\phi \in \mathbb{R}^{|\mathcal{S}||\mathcal{A}|}$,*

$$|\phi^\top \theta^* - \phi^\top \hat{\theta}_k| \leq \sqrt{\alpha} \cdot \nu(k) \|\phi\|_{\Sigma_k^{-1}}.$$

*Proof.* We have

$$|\phi^\top \theta^* - \phi^\top \hat{\theta}_k|$$

$$= \|\phi\|_{\Gamma_k^{-1}(\theta^*, \hat{\theta}_k)} \left\| \theta^* - \hat{\theta}_k \right\|_{\Gamma_k(\theta^*, \hat{\theta}_k)}$$

$$\overset{(a)}{\leq} \sqrt{1 + \frac{Hr_{\max}\sqrt{|\mathcal{S}||\mathcal{A}|}}{m} + \frac{H}{m\sqrt{\lambda}} \|Z_k\|_{\Gamma_k^{-1}(\theta^*,\hat{\theta}_k)} \|\phi\|_{\Lambda_k^{-1}(\theta^*)} \left\| g_k(\theta^*) - g_k(\hat{\theta}_k) \right\|_{\Gamma_k^{-1}(\theta^*,\hat{\theta}_k)}}$$

$$= \sqrt{1 + \frac{Hr_{\max}\sqrt{|\mathcal{S}||\mathcal{A}|}}{m} + \frac{H}{m\sqrt{\lambda}} \|Z_k\|_{\Gamma_k^{-1}(\theta^*,\hat{\theta}_k)} \|\phi\|_{\Lambda_k^{-1}(\theta^*)} \|Z_k - \lambda\theta^*\|_{\Gamma_k^{-1}(\theta^*,\hat{\theta}_k)}}$$

$$\leq \sqrt{1 + \frac{Hr_{\max}\sqrt{|\mathcal{S}||\mathcal{A}|}}{m} + \frac{H}{m\sqrt{\lambda}} \|Z_k\|_{\Gamma_k^{-1}(\theta^*,\hat{\theta}_k)} \|\phi\|_{\Lambda_k^{-1}(\theta^*)} \left( \|Z_k\|_{\Gamma_k^{-1}(\theta^*,\hat{\theta}_k)} + r_{\max}\sqrt{\lambda|\mathcal{S}||\mathcal{A}|} \right)}$$

$$= \frac{m\sqrt{\lambda}}{H} \sqrt{1 + \frac{Hr_{\max}\sqrt{|\mathcal{S}||\mathcal{A}|}}{m} + \frac{H}{m\sqrt{\lambda}} \|Z_k\|_{\Gamma_k^{-1}(\theta^*,\hat{\theta}_k)} \|\phi\|_{\Lambda_k^{-1}(\theta^*)}} \cdot$$
$$\left( \frac{H}{m\sqrt{\lambda}} \|Z_k\|_{\Gamma_k^{-1}(\theta^*,\hat{\theta}_k)} + \frac{Hr_{\max}\sqrt{|\mathcal{S}||\mathcal{A}|}}{m} \right)$$

$$\leq \frac{m\sqrt{\lambda}}{H} \left( 1 + \frac{Hr_{\max}\sqrt{|\mathcal{S}||\mathcal{A}|}}{m} + \frac{H}{m\sqrt{\lambda}} \|Z_k\|_{\Gamma_k^{-1}(\theta^*,\hat{\theta}_k)} \right)^{\frac{3}{2}} \|\phi\|_{\Lambda_k^{-1}(\theta^*)}$$

$$\overset{(b)}{\leq} \frac{m\sqrt{\alpha\lambda}}{H} \left( 1 + \frac{Hr_{\max}\sqrt{|\mathcal{S}||\mathcal{A}|}}{m} + \frac{H}{m\sqrt{\lambda}} \left( \sqrt{1 + \frac{Hr_{\max}\sqrt{|\mathcal{S}||\mathcal{A}|}}{m}}\omega(k) + \frac{H}{m\sqrt{\lambda}}\omega(k)^2 \right) \right)^{\frac{3}{2}} \|\phi\|_{\Sigma_k^{-1}}$$

$$= \sqrt{\alpha} \cdot \nu(k) \|\phi\|_{\Sigma_k^{-1}},$$

where inequality (a) is due to Lemmas C.5 and C.6, and inequality (b) follows from Lemmas C.4 and C.6. $\qquad \square$

**Lemma C.8** (Gaussian Anti-Concentration). *Assume that event $\mathcal{E}$ holds. Then, for any $k > 0$ and $F_{k-1}$-measurable random variable $X \in \mathbb{R}^{|\mathcal{S}||\mathcal{A}|}$, we have*

$$\Pr\left[ X^\top \tilde{\theta}_k > X^\top \theta^* \mid F_{k-1} \right] \geq \frac{1}{2\sqrt{2\pi e}}.$$

*Proof.* This proof is originated from the analysis of Lemma 11 in (Efroni et al., 2021).

Using Lemma C.7, we have that for any $k > 0$,

$$|X^\top \theta^* - X^\top \hat{\theta}_{k-1}| \leq \sqrt{\alpha} \cdot \nu(k-1) \|X\|_{\Sigma_{k-1}^{-1}}.$$

It holds that

$$\Pr\left[ X^\top \tilde{\theta}_k > X^\top \theta^* \mid F_{k-1} \right]$$
$$= \Pr\left[ \frac{X^\top \tilde{\theta}_k - X^\top \hat{\theta}_{k-1}}{\sqrt{\alpha} \cdot \nu(k-1)\|X\|_{\Sigma_{k-1}^{-1}}} > \frac{X^\top \theta^* - X^\top \hat{\theta}_{k-1}}{\sqrt{\alpha} \cdot \nu(k-1)\|X\|_{\Sigma_{k-1}^{-1}}} \mid F_{k-1} \right].$$

Here given $F_{k-1}$, $X^\top \tilde{\theta}_k - X^\top \hat{\theta}_{k-1} = X^\top \xi_k$ is a Gaussian random variable with mean 0 and standard deviation $\sqrt{\alpha} \cdot \nu(k-1)\|X\|_{\Sigma_{k-1}^{-1}}$.

Since when event $\mathcal{E}$ holds,

$$\frac{X^\top \theta^* - X^\top \hat{\theta}_{k-1}}{\sqrt{\alpha} \cdot \nu(k-1)\|X\|_{\Sigma_{k-1}^{-1}}} \leq \frac{\sqrt{\alpha} \cdot \nu(k-1)\|X\|_{\Sigma_{k-1}^{-1}}}{\sqrt{\alpha} \cdot \nu(k-1)\|X\|_{\Sigma_{k-1}^{-1}}} = 1,$$

we have

$$\Pr\left[ X^\top \tilde{\theta}_k > X^\top \theta^* \mid F_{k-1} \right] \geq \Pr\left[ \frac{X^\top \tilde{\theta}_k - X^\top \hat{\theta}_{k-1}}{\sqrt{\alpha} \cdot \nu(k-1)\|X\|_{\Sigma_{k-1}^{-1}}} > 1 \mid F_{k-1} \right]$$

$$= \Pr\left[\frac{X^\top \xi_k}{\sqrt{\alpha} \cdot \nu(k-1)\|X\|_{\Sigma_{k-1}^{-1}}} > 1 \mid F_{k-1}\right]$$

$$\overset{(a)}{\geq} \frac{1}{2\sqrt{2\pi e}},$$

where inequality (a) comes from that if $Z \sim \mathcal{F}^{\mathrm{B}}_{\mathrm{UTran}}(0,1)$, $\Pr[Z > z] \geq \frac{1}{\sqrt{2\pi}} \cdot \frac{z}{1+z^2} e^{-\frac{z^2}{2}}$ (Borjesson & Sundberg, 1979). $\square$

**Lemma C.9.** *Let $\xi_k, \xi'_k \in \mathbb{R}^{|\mathcal{S}||\mathcal{A}|}$ be i.i.d. random variables given $F_{k-1}$. Let $\tilde{p}$ be a $F_{k-1}$-measurable transition model, and $x_{k-1} \in \mathbb{R}^{|\mathcal{S}||\mathcal{A}|}$ be a $F_{k-1}$-measurable random variable. For any policy $\pi$, denote the visitation indicator under policy $\pi$ on MDP $\tilde{p}$ by $\tilde{\phi}^\pi$. Let $\tilde{\pi}^k := \mathrm{argmax}_\pi (\tilde{\phi}^\pi)^\top (x_{k-1} + \xi_k)$. Then, we have*

$$\mathbb{E}\left[\left((\tilde{\phi}^{\tilde{\pi}^k})^\top (x_{k-1} + \xi_k) - \mathbb{E}\left[(\tilde{\phi}^{\tilde{\pi}^k})^\top (x_{k-1} + \xi_k) \mid F_{k-1}\right]\right)^+ \mid F_{k-1}\right]$$

$$\leq \mathbb{E}\left[|(\tilde{\phi}^{\tilde{\pi}^k})^\top \xi_k| + |(\tilde{\phi}^{\tilde{\pi}^k})^\top \xi'_k| \mid F_{k-1}\right].$$

*Proof.* This proof is originated from Lemma 12 in (Efroni et al., 2021).

First, using the definition of $\tilde{\pi}^k$ and the fact that $\xi_k$ and $\xi'_k$ follow the same distribution, we have

$$\mathbb{E}\left[(\tilde{\phi}^{\tilde{\pi}^k})^\top (x_{k-1} + \xi_k) \mid F_{k-1}\right] = \mathbb{E}\left[\max_\pi (\tilde{\phi}^\pi)^\top (x_{k-1} + \xi'_k) \mid F_{k-1}\right]. \tag{8}$$

Then, since given $F_{k-1}$, $\xi_k$ and $\xi'_k$ are independent, we have

$$\mathbb{E}\left[\max_\pi (\tilde{\phi}^\pi)^\top (x_{k-1} + \xi'_k) \mid F_{k-1}\right] = \mathbb{E}\left[\max_\pi (\tilde{\phi}^\pi)^\top (x_{k-1} + \xi'_k) \mid F_{k-1}, \xi_k, \tilde{\pi}_k\right]$$

$$\geq \mathbb{E}\left[(\phi^{\tilde{\pi}_k})^\top (x_{k-1} + \xi'_k) \mid F_{k-1}, \xi_k, \tilde{\pi}_k\right]. \tag{9}$$

Hence, combining Eqs. (8) and (9), we have

$$\mathbb{E}\left[\left((\phi^{\tilde{\pi}_k})^\top (x_{k-1} + \xi_k) - \mathbb{E}\left[(\phi^{\tilde{\pi}_k})^\top (x_{k-1} + \xi_k) \mid F_{k-1}\right]\right)^+ \mid F_{k-1}\right]$$

$$\leq \mathbb{E}\left[\left((\phi^{\tilde{\pi}_k})^\top (x_{k-1} + \xi_k) - \mathbb{E}\left[(\phi^{\tilde{\pi}_k})^\top (x_{k-1} + \xi'_k) \mid F_{k-1}, \xi_k, \tilde{\pi}_k\right]\right)^+ \mid F_{k-1}\right]$$

$$= \mathbb{E}\left[\left(\mathbb{E}\left[(\phi^{\tilde{\pi}_k})^\top (x_{k-1} + \xi_k) - (\phi^{\tilde{\pi}_k})^\top (x_{k-1} + \xi'_k) \mid F_{k-1}, \xi_k, \tilde{\pi}_k\right]\right)^+ \mid F_{k-1}\right]$$

$$= \mathbb{E}\left[\left(\mathbb{E}\left[(\phi^{\tilde{\pi}_k})^\top \xi_k - (\phi^{\tilde{\pi}_k})^\top \xi'_k \mid F_{k-1}, \xi_k, \tilde{\pi}_k\right]\right)^+ \mid F_{k-1}\right]$$

$$\leq \mathbb{E}\left[\left|\mathbb{E}\left[(\phi^{\tilde{\pi}_k})^\top \xi_k - (\phi^{\tilde{\pi}_k})^\top \xi'_k \mid F_{k-1}, \xi_k, \tilde{\pi}_k\right]\right| \mid F_{k-1}\right]$$

$$\leq \mathbb{E}\left[\mathbb{E}\left[|(\phi^{\tilde{\pi}_k})^\top \xi_k - (\phi^{\tilde{\pi}_k})^\top \xi'_k| \mid F_{k-1}, \xi_k, \tilde{\pi}_k\right] \mid F_{k-1}\right]$$

$$= \mathbb{E}\left[|(\phi^{\tilde{\pi}_k})^\top \xi_k - (\phi^{\tilde{\pi}_k})^\top \xi'_k| \mid F_{k-1}\right]$$

$$\leq \mathbb{E}\left[|(\phi^{\tilde{\pi}_k})^\top \xi_k| \mid F_{k-1}\right] + \mathbb{E}\left[|(\phi^{\tilde{\pi}_k})^\top \xi'_k| \mid F_{k-1}\right].$$

$\square$

For any $k > 0$ and $\delta_k \in (0,1)$, define event

$$\mathcal{M}_k(\delta_k) := \left\{\forall \phi \in \mathbb{R}^{|\mathcal{S}||\mathcal{A}|} : |\phi^\top \xi_k| \leq \sqrt{\alpha} \cdot \nu(k-1)\left(\sqrt{|\mathcal{S}||\mathcal{A}|} + 2\sqrt{\log\left(\frac{1}{\delta_k}\right)}\right)\|\phi\|_{\Sigma_{k-1}^{-1}}\right\}.$$

**Lemma C.10.** *For any $k > 0$ and $\delta_k \in (0, 1)$, we have*

$$\Pr\left[\mathcal{M}_k(\delta_k) \mid F_{k-1}\right] \geq 1 - \delta_k.$$

*In addition, for a random variable $X \in \mathbb{R}^{|\mathcal{S}||\mathcal{A}|}$ such that $\|X\|_{\Sigma_{k-1}^{-1}} \leq L_X$, we have*

$$\mathbb{E}\left[|X^\top \xi_k| \,|F_{k-1}\right] \leq \sqrt{\alpha} \cdot \nu(k-1) \left( \sqrt{|\mathcal{S}||\mathcal{A}|} + 2\sqrt{\log\left(\frac{1}{\delta_k}\right)} \right) \mathbb{E}\left[\|X\|_{\Sigma_{k-1}^{-1}} \,|F_{k-1}\right]$$
$$+ \sqrt{\alpha} \cdot \nu(k-1) \cdot L_X \sqrt{|\mathcal{S}||\mathcal{A}|\delta_k}.$$

*Proof.* This proof is similar to the analysis of Lemma 13 in (Efroni et al., 2021).

First, we prove the first statement.

For any $\phi \in \mathbb{R}^{|\mathcal{S}||\mathcal{A}|}$, we have

$$\begin{aligned}
|\phi^\top \xi_k| &= |\phi^\top \Sigma_{k-1}^{-\frac{1}{2}} \Sigma_{k-1}^{\frac{1}{2}} \xi_k| \\
&\leq \left\|\Sigma_{k-1}^{-\frac{1}{2}} \phi\right\|_2 \left\|\Sigma_{k-1}^{\frac{1}{2}} \xi_k\right\|_2 \\
&= \sqrt{\alpha} \cdot \nu(k-1) \|\phi\|_{\Sigma_{k-1}^{-1}} \left\|\frac{1}{\sqrt{\alpha} \cdot \nu(k-1)} \Sigma_{k-1}^{\frac{1}{2}} \xi_k\right\|_2.
\end{aligned} \tag{10}$$

Since given $F_{k-1}$, $\frac{1}{\sqrt{\alpha}\cdot\nu(k-1)} \Sigma_{k-1}^{\frac{1}{2}} \xi_k \in \mathbb{R}^{|\mathcal{S}||\mathcal{A}|}$ is a vector with each entry being a standard Gaussian random variable, we have that $\|\frac{1}{\sqrt{\alpha}\cdot\nu(k-1)} \Sigma_{k-1}^{\frac{1}{2}} \xi_k\|_2$ is chi-distributed with parameter $|\mathcal{S}||\mathcal{A}|$.

Then, using Lemma 1 in (Laurent & Massart, 2000), we have that with probability at least $1 - \delta_k$,

$$\begin{aligned}
\left\|\frac{1}{\sqrt{\alpha} \cdot \nu(k-1)} \Sigma_{k-1}^{\frac{1}{2}} \xi_k\right\|_2 &\leq \sqrt{|\mathcal{S}||\mathcal{A}| + 2\sqrt{|\mathcal{S}||\mathcal{A}| \log\left(\frac{1}{\delta_k}\right)} + 2\log\left(\frac{1}{\delta_k}\right)} \\
&= \sqrt{\left(\sqrt{|\mathcal{S}||\mathcal{A}|} + \sqrt{\log\left(\frac{1}{\delta_k}\right)}\right)^2 + \log\left(\frac{1}{\delta_k}\right)} \\
&\leq \sqrt{|\mathcal{S}||\mathcal{A}|} + 2\sqrt{\log\left(\frac{1}{\delta_k}\right)}.
\end{aligned}$$

Next, we prove the second statement.

For a random variable $X \in \mathbb{R}^{|\mathcal{S}||\mathcal{A}|}$, we have

$$\begin{aligned}
\mathbb{E}\left[|X^\top \xi_k| \,|F_{k-1}\right] &= \Pr\left[\mathcal{M}_k(\delta_k)\right] \cdot \mathbb{E}\left[|X^\top \xi_k| \,|F_{k-1}, \mathcal{M}_k(\delta_k)\right] \\
&\quad + \Pr\left[\bar{\mathcal{M}}_k(\delta_k)\right] \cdot \mathbb{E}\left[|X^\top \xi_k| \,|F_{k-1}, \bar{\mathcal{M}}_k(\delta_k)\right] \\
&\leq \sqrt{\alpha} \cdot \nu(k-1) \left( \sqrt{|\mathcal{S}||\mathcal{A}|} + 2\sqrt{\log\left(\frac{1}{\delta_k}\right)} \right) \mathbb{E}\left[\|X\|_{\Sigma_{k-1}^{-1}} \,|F_{k-1}\right] \\
&\quad + \sqrt{\Pr\left[\bar{\mathcal{M}}_k(\delta_k)\right] \cdot \mathbb{E}\left[|X^\top \xi_k|^2 \,|F_{k-1}, \bar{\mathcal{M}}_k(\delta_k)\right]} \\
&\overset{(a)}{\leq} \sqrt{\alpha} \cdot \nu(k-1) \left( \sqrt{|\mathcal{S}||\mathcal{A}|} + 2\sqrt{\log\left(\frac{1}{\delta_k}\right)} \right) \mathbb{E}\left[\|X\|_{\Sigma_{k-1}^{-1}} \,|F_{k-1}\right] \\
&\quad + \sqrt{\alpha} \cdot \nu(k-1) \sqrt{\delta_k \mathbb{E}\left[\|X\|_{\Sigma_{k-1}^{-1}}^2 \cdot \left\|\frac{1}{\sqrt{\alpha} \cdot \nu(k-1)} \Sigma_{k-1}^{\frac{1}{2}} \xi_k\right\|_2^2 \,|F_{k-1}\right]}
\end{aligned}$$

$$\leq \sqrt{\alpha} \cdot \nu(k-1) \left( \sqrt{|\mathcal{S}||\mathcal{A}|} + 2\sqrt{\log\left(\frac{1}{\delta_k}\right)} \right) \mathbb{E}\left[ \|X\|_{\Sigma_{k-1}^{-1}} \,|F_{k-1} \right]$$

$$+ \sqrt{\alpha} \cdot \nu(k-1) \sqrt{\delta_k L_X^2 \mathbb{E}\left[ \left\| \frac{1}{\sqrt{\alpha} \cdot \nu(k-1)} \Sigma_{k-1}^{\frac{1}{2}} \xi_k \right\|_2^2 \,|F_{k-1} \right]}$$

$$\overset{(b)}{\leq} \sqrt{\alpha} \cdot \nu(k-1) \left( \sqrt{|\mathcal{S}||\mathcal{A}|} + 2\sqrt{\log\left(\frac{1}{\delta_k}\right)} \right) \mathbb{E}\left[ \|X\|_{\Sigma_{k-1}^{-1}} \,|F_{k-1} \right]$$

$$+ \sqrt{\alpha} \cdot \nu(k-1) \cdot L_X \sqrt{|\mathcal{S}||\mathcal{A}|\delta_k}.$$

Here inequality (a) follows from the Cauchy-Schwarz inequality. Inequality (b) is due to the fact that given $F_{k-1}$, $\|\frac{1}{\sqrt{\alpha}\cdot\nu(k-1)} \Sigma_{k-1}^{\frac{1}{2}} \xi_k\|_2$ is chi-distributed with parameter $|\mathcal{S}||\mathcal{A}|$, and then $\mathbb{E}[\|\frac{1}{\sqrt{\alpha}\cdot\nu(k-1)} \Sigma_{k-1}^{\frac{1}{2}} \xi_k\|_2^2 \,|F_{k-1}] = |\mathcal{S}||\mathcal{A}|$. $\qquad \square$

Define event

$$\mathcal{F}_{\mathrm{KTran}}^{\mathrm{B}} := \left\{ \left| \sum_{k'=1}^{k} \left( \mathbb{E}_{\tau \sim \pi^{k'}} \left[ \|\phi^\tau\|_{(\Sigma_{k'-1})^{-1}} \,|F_{k'-1} \right] - \|\phi^\tau\|_{(\Sigma_{k'-1})^{-1}} \right) \right| \leq 4H\sqrt{\frac{k}{\alpha\lambda} \log\left(\frac{4k}{\delta'}\right)}, \right.$$

$$\left. \left| \sum_{k'=1}^{k} \left( \mathbb{E}\left[ (\phi^{\pi^{k'}})^\top \theta^* | F_{k'-1} \right] - (\phi^{\pi^{k'}})^\top \theta^* \right) \right| \leq 4H r_{\max} \sqrt{k \log\left(\frac{4k}{\delta'}\right)}, \; \forall k > 0 \right\}. \qquad (11)$$

**Lemma C.11.** *It holds that*

$$\Pr\left[ \mathcal{F}_{\mathrm{KTran}}^{\mathrm{B}} \right] \geq 1 - 2\delta'.$$

*Proof.* We prove the first inequality as follows.

For any $k' \geq 1$, we have that $\|\phi^\tau\|_{(\Sigma_{k'-1})^{-1}} \leq \frac{H}{\sqrt{\alpha\lambda}}$, and then $|\mathbb{E}_{\tau \sim \pi^{k'}}[\|\phi^\tau\|_{(\Sigma_{k'-1})^{-1}} \,|F_{k'-1}] - \|\phi^\tau\|_{(\Sigma_{k'-1})^{-1}}| \leq \frac{2H}{\sqrt{\alpha\lambda}}$.

Using the Azuma-Hoeffding inequality, we have that for any fixed $k > 0$, with probability at least $1 - \frac{\delta'}{2k^2}$,

$$\left| \sum_{k'=1}^{k} \left( \mathbb{E}_{\tau \sim \pi^{k'}} \left[ \|\phi^\tau\|_{(\Sigma_{k'-1})^{-1}} \,|F_{k'-1} \right] - \|\phi^\tau\|_{(\Sigma_{k'-1})^{-1}} \right) \right| \leq \sqrt{2 \cdot \frac{4H^2}{\alpha\lambda} \cdot k \log\left(\frac{4k^2}{\delta'}\right)}.$$

Since $\sum_{k=1}^{\infty} \frac{\delta'}{2k^2} \leq \delta'$, by a union bound over $k$, we have that with probability at least $\delta'$, for any $k \geq 1$,

$$\left| \sum_{k'=1}^{k} \left( \mathbb{E}_{\tau \sim \pi^{k'}} \left[ \|\phi^\tau\|_{(\Sigma_{k'-1})^{-1}} \,|F_{k'-1} \right] - \|\phi^\tau\|_{(\Sigma_{k'-1})^{-1}} \right) \right| \leq \sqrt{2 \cdot \frac{4H^2}{\alpha\lambda} \cdot k \log\left(\frac{4k^2}{\delta'}\right)}$$

$$\leq 4H\sqrt{\frac{k}{\alpha\lambda} \log\left(\frac{4k}{\delta'}\right)}.$$

The second inequality can be obtained by a similar argument and the fact that $|(\phi^{\pi^k})^\top \theta^*| \leq H r_{\max}$ for any $k > 0$. $\qquad \square$

**Lemma C.12.** *For any $K \geq 1$, we have*

$$\sum_{k=1}^{K} \sum_{i=1}^{m} \left\| \phi^{\tau_i^k} \right\|_{(\Sigma_{k-1})^{-1}} \leq \sqrt{2Km|\mathcal{S}||\mathcal{A}| \cdot \max\left\{ \frac{H^2}{m\alpha\lambda}, 1 \right\} \cdot \log\left( 1 + \frac{KH^2}{\alpha\lambda|\mathcal{S}||\mathcal{A}|m} \right)}.$$

*Proof.* We have

$$\sum_{k=1}^{K} \sum_{i=1}^{m} \left\| \phi^{\tau_i^k} \right\|_{(\Sigma_{k-1})^{-1}} \leq \sqrt{Km \sum_{k=1}^{K} \sum_{i=1}^{m} \left\| \phi^{\tau_i^k} \right\|_{(\Sigma_{k-1})^{-1}}^2}$$

$$\overset{(a)}{\leq} \sqrt{2Km \cdot \max\left\{\frac{H^2}{m\alpha\lambda}, 1\right\} \cdot \sum_{k=1}^{K} \log\left(1 + \sum_{i=1}^{m} \left\|\phi^{\tau_i^k}\right\|^2_{(\Sigma_{k-1})^{-1}}\right)}$$

$$= \sqrt{2Km \cdot \max\left\{\frac{H^2}{m\alpha\lambda}, 1\right\} \cdot \log\left(\frac{\det(\Sigma_K)}{\det(\alpha\lambda I)}\right)}$$

$$\leq \sqrt{2Km|\mathcal{S}||\mathcal{A}| \cdot \max\left\{\frac{H^2}{m\alpha\lambda}, 1\right\} \cdot \log\left(1 + \frac{KH^2}{\alpha\lambda|\mathcal{S}||\mathcal{A}|m}\right)}, \tag{12}$$

where inequality (a) is due to that for any $x \in [0, c]$ with constant $c \geq 0$, it holds that $x \leq 2\max\{c, 1\} \cdot \log(1 + x)$. $\qquad\square$

*Proof of Theorem 3.1.* Letting $\delta' = \frac{\delta}{3}$, we have $\Pr[\mathcal{E} \cap \mathcal{F}^B_{\text{KTran}}] \leq 1 - \delta$. Then, to prove this theorem, it suffices to prove the regret bound when event $\mathcal{E} \cap \mathcal{F}^B_{\text{KTran}}$ holds.

Assume that event $\mathcal{E} \cap \mathcal{F}^B_{\text{KTran}}$ holds. Then, we have

$$\mathcal{R}(K) = \sum_{k=1}^{K} \left((\phi^{\pi^*})^\top \theta^* - (\phi^{\pi^k})^\top \theta^*\right)$$

$$= \sum_{k=1}^{K} \left(\mathbb{E}\left[(\phi^{\pi^*})^\top \theta^* - (\phi^{\pi^k})^\top \theta^* | F_{k-1}\right] + \mathbb{E}\left[(\phi^{\pi^k})^\top \theta^* | F_{k-1}\right] - (\phi^{\pi^k})^\top \theta^*\right)$$

$$\leq \sum_{k=1}^{K} \left(\mathbb{E}\left[(\phi^{\pi^*})^\top \theta^* - (\phi^{\pi^k})^\top \theta^* | F_{k-1}\right]\right) + 4Hr_{\max}\sqrt{K\log\left(\frac{4K}{\delta'}\right)}. \tag{13}$$

For the first term, we have

$$\sum_{k=1}^{K} \mathbb{E}\left[(\phi^{\pi^*})^\top \theta^* - (\phi^{\pi^k})^\top \theta^* | F_{k-1}\right]$$

$$= \sum_{k=1}^{K} \left(\mathbb{E}\left[(\phi^{\pi^*})^\top \theta^* - (\phi^{\pi^k})^\top \tilde{\theta}_k | F_{k-1}\right] + \mathbb{E}\left[(\phi^{\pi^k})^\top \tilde{\theta}_k - (\phi^{\pi^k})^\top \theta^* | F_{k-1}\right]\right). \tag{14}$$

In the following, we prove

$$\mathbb{E}\left[(\phi^{\pi^*})^\top \theta^* - (\phi^{\pi^k})^\top \tilde{\theta}_k | F_{k-1}\right] \leq 2\sqrt{2\pi e} \cdot \mathbb{E}\left[\left((\phi^{\pi^k})^\top \tilde{\theta}_k - \mathbb{E}\left[(\phi^{\pi^k})^\top \tilde{\theta}_k | F_{k-1}\right]\right)^+ | F_{k-1}\right]. \tag{15}$$

If $\mathbb{E}[(\phi^{\pi^*})^\top \theta^* - (\phi^{\pi^k})^\top \tilde{\theta}_k | F_{k-1}] < 0$, then Eq. (15) trivially holds.

Otherwise, letting $z := \mathbb{E}[(\phi^{\pi^*})^\top \theta^* - (\phi^{\pi^k})^\top \tilde{\theta}_k | F_{k-1}]$, we have

$$\mathbb{E}\left[\left((\phi^{\pi^k})^\top \tilde{\theta}_k - \mathbb{E}\left[(\phi^{\pi^k})^\top \tilde{\theta}_k | F_{k-1}\right]\right)^+ | F_{k-1}\right]$$

$$\geq z \Pr\left[(\phi^{\pi^k})^\top \tilde{\theta}_k - \mathbb{E}\left[(\phi^{\pi^k})^\top \tilde{\theta}_k | F_{k-1}\right] \geq z | F_{k-1}\right]$$

$$\geq \left(\mathbb{E}\left[(\phi^{\pi^*})^\top \theta^* - (\phi^{\pi^k})^\top \tilde{\theta}_k | F_{k-1}\right]\right) \cdot \Pr\left[(\phi^{\pi^k})^\top \tilde{\theta}_k \geq (\phi^{\pi^*})^\top \theta^* | F_{k-1}\right]$$

$$\overset{(a)}{\geq} \left(\mathbb{E}\left[(\phi^{\pi^*})^\top \theta^* - (\phi^{\pi^k})^\top \tilde{\theta}_k | F_{k-1}\right]\right) \cdot \Pr\left[(\phi^{\pi^*})^\top \tilde{\theta}_k \geq (\phi^{\pi^*})^\top \theta^* | F_{k-1}\right]$$

$$\overset{(b)}{\geq} \left(\mathbb{E}\left[(\phi^{\pi^*})^\top \theta^* - (\phi^{\pi^k})^\top \tilde{\theta}_k | F_{k-1}\right]\right) \cdot \frac{1}{2\sqrt{2\pi e}},$$

where inequality (a) uses the definition of $\pi^k$, and inequality (b) follows from Lemma C.8. Thus, we complete the proof of Eq. (15).

Let $\xi'_k \in \mathbb{R}^{|\mathcal{S}||\mathcal{A}|}$ be a random variable that is i.i.d. with $\xi$ given $F_{k-1}$. Then, using Lemma C.9 with $p' = p$, $x_{k-1} = \hat{\theta}_{k-1}$ and $\tilde{\pi}^k = \pi^k$, we have

$$\mathbb{E}\left[(\phi^{\pi^*})^\top \theta^* - (\phi^{\pi^k})^\top \tilde{\theta}_k | F_{k-1}\right] \leq 2\sqrt{2\pi e} \cdot \mathbb{E}\left[\left((\phi^{\pi^k})^\top \tilde{\theta}_k - \mathbb{E}\left[(\phi^{\pi^k})^\top \tilde{\theta}_k | F_{k-1}\right]\right)^+ | F_{k-1}\right]$$

$$\leq 2\sqrt{2\pi e} \cdot \mathbb{E}\left[|\phi(\pi^k)^\top \xi_k| + |\phi(\pi^k)^\top \xi'_k| \,| F_{k-1}\right].$$

Plugging the above inequality into Eq. (14) and using Lemma C.10 with $\delta_k = \frac{1}{k^4}$ and $L_X = \frac{H}{\sqrt{\alpha\lambda}}$, we have

$$\sum_{k=1}^{K} \mathbb{E}\left[(\phi^{\pi^*})^\top \theta^* - (\phi^{\pi^k})^\top \theta^* | F_{k-1}\right]$$

$$= \sum_{k=1}^{K}\left(2\sqrt{2\pi e}\, \mathbb{E}\left[|(\phi^{\pi^k})^\top \xi_k| + |(\phi^{\pi^k})^\top \xi'_k| \,| F_{k-1}\right] + \mathbb{E}\left[(\phi^{\pi^k})^\top \left(\hat{\theta}_{k-1} + \xi_k\right) - (\phi^{\pi^k})^\top \theta^* | F_{k-1}\right]\right)$$

$$= \sum_{k=1}^{K}\left(\left(2\sqrt{2\pi e} + 1\right) \cdot \mathbb{E}\left[|(\phi^{\pi^k})^\top \xi_k| \,| F_{k-1}\right] + 2\sqrt{2\pi e} \cdot \mathbb{E}\left[|(\phi^{\pi^k})^\top \xi'_k| \,| F_{k-1}\right]\right.$$

$$\left. + \mathbb{E}\left[(\phi^{\pi^k})^\top \hat{\theta}_{k-1} - (\phi^{\pi^k})^\top \theta^* | F_{k-1}\right]\right)$$

$$\overset{(a)}{\leq} \sum_{k=1}^{K}\left(\left(4\sqrt{2\pi e} + 2\right)\sqrt{\alpha} \cdot \nu(k-1)\left(\sqrt{|\mathcal{S}||\mathcal{A}|} + 4\sqrt{\log(k)}\right) \cdot \mathbb{E}\left[\left\|\phi^{\pi^k}\right\|_{\Sigma_{k-1}^{-1}} | F_{k-1}\right]\right.$$

$$\left. + \left(4\sqrt{2\pi e} + 1\right)\sqrt{\alpha} \cdot \nu(k-1)\frac{\sqrt{|\mathcal{S}||\mathcal{A}|}}{k^2} \cdot \frac{H}{\sqrt{\alpha\lambda}}\right), \tag{16}$$

where inequality (a) uses Lemmas C.7 and C.10.

Here according to the definition of event $\mathcal{F}_{\mathrm{KTran}}^{\mathrm{B}}$ and Lemma C.12, we have

$$\sum_{k=1}^{K} \mathbb{E}\left[\left\|\phi^{\pi^k}\right\|_{\Sigma_{k-1}^{-1}} | F_{k-1}\right] = \sum_{k=1}^{K}\left(\mathbb{E}\left[\left\|\phi^{\pi^k}\right\|_{\Sigma_{k-1}^{-1}} | F_{k-1}\right] - \left\|\phi^{\pi^k}\right\|_{\Sigma_{k-1}^{-1}}\right) + \sum_{k=1}^{K}\left\|\phi^{\pi^k}\right\|_{\Sigma_{k-1}^{-1}}$$

$$\leq 4H\sqrt{\frac{K}{\alpha\lambda}\log\left(\frac{4K}{\delta'}\right)}$$

$$+ \sqrt{2Km|\mathcal{S}||\mathcal{A}| \cdot \max\left\{\frac{H^2}{m\alpha\lambda}, 1\right\} \cdot \log\left(1 + \frac{KH^2}{\alpha\lambda|\mathcal{S}||\mathcal{A}|m}\right)}. \tag{17}$$

Therefore, plugging the above two equations into Eq. (13), we have

$$\mathcal{R}(K) \leq \left(4\sqrt{2\pi e} + 2\right)\sqrt{\alpha} \cdot \nu(K)\left(\sqrt{|\mathcal{S}||\mathcal{A}|} + 4\sqrt{\log(K)}\right) \cdot$$

$$\left(4H\sqrt{\frac{K}{\alpha\lambda}\log\left(\frac{4K}{\delta'}\right)} + \sqrt{2Km|\mathcal{S}||\mathcal{A}|\max\left\{\frac{H^2}{m\alpha\lambda}, 1\right\}\log\left(1 + \frac{KH^2}{\alpha\lambda|\mathcal{S}||\mathcal{A}|m}\right)}\right)$$

$$+ 2\left(4\sqrt{2\pi e} + 1\right)H \cdot \nu(K)\sqrt{\frac{|\mathcal{S}||\mathcal{A}|}{\lambda}} + 4Hr_{\max}\sqrt{K\log\left(\frac{4K}{\delta'}\right)}$$

$$\overset{(a)}{=} \tilde{O}\left(\exp(\frac{Hr_{\max}}{2m}) \cdot \nu(K)\sqrt{|\mathcal{S}||\mathcal{A}|}\left(\sqrt{Km|\mathcal{S}||\mathcal{A}| \cdot \max\left\{\frac{H^2}{m\alpha\lambda}, 1\right\}} + H\sqrt{\frac{K}{\alpha\lambda}}\right)\right),$$

where in equality (a), the last two terms are absorbed into $\tilde{O}(\cdot)$. $\qquad\square$

Optimal action: $r(s_i, a_i^*) = r_{max}$
Suboptimal action: $r(s_i, a_i^{sub}) = (1 - \varepsilon)r_{max}$

$a_i^*$: transition to $s_{n+1}$
$a_i^{sub}$: transition to $s_{n+2}$

w.p. $\frac{1}{n}$   $s_1$

w.p. $\frac{1}{n}$   $s_2$

$s_{n+1}$   $r(s_{n+1}, \cdot) = r_{max}$

$\vdots$

$s_{n+2}$   $r(s_{n+2}, \cdot) = (1 - \varepsilon)r_{max}$

w.p. $\frac{1}{n}$   $s_n$

*Figure 5.* Instance for the lower bound under binary segment feedback and known transition.

## C.2. Proof for the Regret Lower Bound with Known Transition

In the following, we prove the regret lower bound (Theorem 3.2) for RL with binary segment feedback and known transition.

*Proof of Theorem 3.2.* We construct a random instance $\mathcal{I}$ as follows. As shown in Figure 5, there are $n$ bandit states $s_1, \ldots, s_n$ (i.e., there is an optimal action and multiple suboptimal actions), a good absorbing state $s_{n+1}$ and a bad absorbing state $s_{n+2}$. The agent starts from $s_1, \ldots, s_n$ with equal probability $\frac{1}{n}$. For any $i \in [n]$, in state $s_i$, one action $a_J$ is uniformly chosen from $\mathcal{A}$ as the optimal action. In state $s_i$, under the optimal action $a_J$, the agent transitions to $s_{n+1}$ deterministically, and $r(s_i, a_J) = r_{\max}$; Under any suboptimal action $a \in \mathcal{A} \setminus \{s_J\}$, the agent transitions to $s_{n+2}$ deterministically, and $r(s_i, a) = (1 - \varepsilon)r_{\max}$, where $\varepsilon \in (0, \frac{1}{2})$ is a parameter specified later. For all actions $a \in \mathcal{A}$, $r(s_{n+1}, a) = r_{\max}$ and $r(s_{n+2}, a) = (1 - \varepsilon)r_{\max}$.

In this proof, we will also use an alternative uniform instance $\mathcal{I}_{\text{unif}}$. The only difference between $\mathcal{I}_{\text{unif}}$ and $\mathcal{I}$ is that for any $i \in [n]$, in state $s_i$, under all actions $a \in \mathcal{A}$, the agent transitions to $s_{n+2}$ deterministically, and $r(s_i, a) = (1 - \varepsilon)r_{\max}$.

Fix an algorithm $\mathbb{A}$. Let $\mathbb{E}_{\text{unif}}[\cdot]$ denote the expectation with respect to $\mathcal{I}_{\text{unif}}$. Let $\mathbb{E}_*[\cdot]$ denote the expectation with respect to $\mathcal{I}$. For any $i \in [n]$ and $j \in [|\mathcal{A}|]$, let $\mathbb{E}_{i,j}[\cdot]$ denote the expectation with respect to the case where $a_j$ is the optimal action in state $s_i$, and $N_{i,j}$ denote the number of episodes where algorithm $\mathbb{A}$ chooses $a_j$ in state $s_i$, i.e., $N_{i,j} = \sum_{k=1}^{K} \mathbb{1}\{\pi_1^k(s_i) = a_j\}$.

The KL divergence of binary observations if taking $a_J$ in $s_i$ in each episode between $\mathcal{I}_{\text{unif}}$ and $\mathcal{I}$ is

$$
\sum_{i=1}^{m} \text{KL}\left( \mathcal{B}\left( \mu\left( (1 - \varepsilon)r_{\max} \cdot \frac{H}{m} \right) \right) \,\middle\|\, \mathcal{B}\left( \mu\left( r_{\max} \cdot \frac{H}{m} \right) \right) \right)
$$

$$
\overset{(a)}{\leq} m \cdot \frac{\left( \mu\left( (1 - \varepsilon)r_{\max} \cdot \frac{H}{m} \right) - \mu\left( r_{\max} \cdot \frac{H}{m} \right) \right)^2}{\mu'\left( r_{\max} \cdot \frac{H}{m} \right)}
$$

$$
\overset{(b)}{\leq} m \cdot \frac{\mu'\left( (1 - \varepsilon)\frac{Hr_{\max}}{m} \right)^2 \left( \varepsilon \cdot \frac{Hr_{\max}}{m} \right)^2}{\mu'\left( \frac{Hr_{\max}}{m} \right)},
$$

where inequality (a) uses the fact that $\text{KL}(\mathcal{B}(p) \| \mathcal{B}(q)) \leq \frac{(p-q)^2}{q(1-q)}$, and inequality (b) is due to that $\mu'(x)$ is monotonically decreasing when $x > 0$.

In addition, the agent has probability only $\frac{1}{n}$ to arrive at (observe) state $s_i$.

Thus, using Lemma A.1 in (Auer et al., 2002), we have that for any $i \in [n]$, in state $s_i$,

$$
\mathbb{E}_{i,j}[N_{i,j}] \leq \mathbb{E}_{\text{unif}}[N_{i,j}] + \frac{K}{2} \sqrt{ \frac{1}{n} \cdot \mathbb{E}_{\text{unif}}[N_{i,j}] \cdot m \cdot \frac{\mu'\left( (1 - \varepsilon)\frac{Hr_{\max}}{m} \right)^2 \left( \varepsilon \cdot \frac{Hr_{\max}}{m} \right)^2}{\mu'\left( \frac{Hr_{\max}}{m} \right)} }
$$

$$= \mathbb{E}_{\text{unif}}[N_{i,j}] + \frac{K}{2} \cdot \varepsilon \cdot \frac{Hr_{\max}}{m} \sqrt{\frac{m}{n} \cdot \mathbb{E}_{\text{unif}}[N_{i,j}] \cdot \frac{\mu'\left((1-\varepsilon)\frac{Hr_{\max}}{m}\right)^2}{\mu'\left(\frac{Hr_{\max}}{m}\right)}}.$$

Summing over $j \in [|\mathcal{A}|]$, using the Cauchy-Schwarz inequality and the fact that $\sum_{j=1}^{|\mathcal{A}|} \mathbb{E}_{\text{unif}}[N_{i,j}] = K$, we have

$$\sum_{j=1}^{|\mathcal{A}|} \mathbb{E}_{i,j}[N_{i,j}] \le K + \frac{KHr_{\max}\varepsilon}{2} \sqrt{\frac{|\mathcal{A}|K}{mn} \cdot \frac{\mu'\left((1-\varepsilon)\frac{Hr_{\max}}{m}\right)^2}{\mu'\left(\frac{Hr_{\max}}{m}\right)}}$$

$$\le K + \frac{KHr_{\max}\varepsilon}{2} \sqrt{\frac{|\mathcal{A}|K}{mn} \cdot \frac{\mu'\left((1-c_0)\frac{Hr_{\max}}{m}\right)^2}{\mu'\left(\frac{Hr_{\max}}{m}\right)}},$$

where $c_0 \in (0, \frac{1}{2})$ is a constant which satisfies $c_0 \ge \varepsilon$. We will specify how to make $c_0 \ge \varepsilon$ to satisfy this condition later. Then, we have

$$\mathcal{R}(K) = \sum_{k=1}^{K} \mathbb{E}_* \left[ V^* - V^{\pi^k} \right]$$

$$= r_{\max}HK - \frac{1}{n} \sum_{i=1}^{n} \left( (1-\varepsilon)r_{\max}HK + \varepsilon r_{\max}H \cdot \frac{1}{|\mathcal{A}|} \sum_{j=1}^{|\mathcal{A}|} \mathbb{E}_{i,j}[N_{i,j}] \right)$$

$$\ge \varepsilon r_{\max}H \left( K - \frac{K}{|\mathcal{A}|} - \frac{KHr_{\max}\varepsilon}{2} \sqrt{\frac{K}{|\mathcal{A}|mn} \cdot \frac{\mu'\left((1-c_0)\frac{Hr_{\max}}{m}\right)^2}{\mu'\left(\frac{Hr_{\max}}{m}\right)}} \right).$$

Let

$$\varepsilon = \frac{1}{2Hr_{\max}} \sqrt{\frac{|\mathcal{A}|mn}{K} \cdot \frac{\mu'\left(\frac{Hr_{\max}}{m}\right)}{\mu'\left((1-c_0)\frac{Hr_{\max}}{m}\right)^2}}.$$

Then, the constant $c_0$ should satisfy

$$\varepsilon = \frac{1}{2Hr_{\max}} \sqrt{\frac{|\mathcal{A}|mn}{K} \cdot \frac{\mu'\left(\frac{Hr_{\max}}{m}\right)}{\mu'\left((1-c_0)\frac{Hr_{\max}}{m}\right)^2}} \le c_0.$$

Since

$$\frac{\mu'\left(\frac{Hr_{\max}}{m}\right)}{\mu'\left((1-c_0)\frac{Hr_{\max}}{m}\right)^2} = \frac{\left(\exp\left((1-c_0)\frac{Hr_{\max}}{m}\right) + \exp\left(-(1-c_0)\frac{Hr_{\max}}{m}\right) + 2\right)^2}{\exp\left(\frac{Hr_{\max}}{m}\right) + \exp\left(-\frac{Hr_{\max}}{m}\right) + 2}$$

$$\le \frac{\left(4\exp\left((1-c_0)\frac{Hr_{\max}}{m}\right)\right)^2}{\exp\left(\frac{Hr_{\max}}{m}\right)}$$

$$= 16\exp\left((1-2c_0)\frac{Hr_{\max}}{m}\right),$$

it suffices to let $c_0$ satisfy

$$\frac{1}{2Hr_{\max}} \sqrt{\frac{|\mathcal{A}|mn}{K} \cdot 16\exp\left((1-2c_0)\frac{Hr_{\max}}{m}\right)} \le c_0,$$

---

**Algorithm 3** SegBiTS-Tran

---

1: **Input:** $\delta, \delta' := \frac{\delta}{8}, \lambda$.
2: **for** $k = 1, \ldots, K$ **do**
3:      $\hat{\theta}_{k-1} \leftarrow \arg\min_\theta -(\sum_{k'=1}^{k-1} \sum_{i=1}^m (y_i^{k'} \cdot \log(\mu((\phi^{\tau_i^{k'}})^\top \theta)) + (1 - y_i^{k'}) \cdot \log(1 - \mu((\phi^{\tau_i^{k'}})^\top \theta))) - \frac{1}{2}\lambda\|\theta\|_2^2)$
4:      $\Sigma_{k-1} \leftarrow \sum_{k'=1}^{k-1} \sum_{i=1}^m \phi^{\tau_i^{k'}} (\phi^{\tau_i^{k'}})^\top + \alpha\lambda I$
5:      Draw a noise $\xi_k \sim \mathcal{N}(0, \alpha \cdot \nu(k-1)^2 \cdot \Sigma_{k-1}^{-1})$, where $\nu(k-1)$ is defined in Eq. (1)
6:      $b_{k-1}^{pv}(s,a) \leftarrow \min\{2Hr_{\max}\sqrt{\frac{\log\left(\frac{KH|\mathcal{S}||\mathcal{A}|}{\delta'}\right)}{n_{k-1}(s,a)}}, Hr_{\max}\}$ for any $(s,a) \in \mathcal{S} \times \mathcal{A}$
7:      $\tilde{\theta}_k^b \leftarrow \hat{\theta}_{k-1} + \xi_k + b_{k-1}^{pv}$
8:      $\pi^k \leftarrow \arg\max_\pi (\hat{\phi}_{k-1}^\pi)^\top \tilde{\theta}_k^b$, where $\hat{\phi}_{k-1}^\pi$ is defined in Eq. (18)
9:      Play episode $k$ with policy $\pi^k$. Observe $\tau^k$ and binary segment feedback $\{y_i^k\}_{i=1}^m$
10: **end for**

---

which is equivalent to $K \geq \frac{4|\mathcal{A}|mn}{H^2 r_{\max}^2 c_0^2} \exp((1 - 2c_0)\frac{Hr_{\max}}{m})$.

It suffices to let

$$K \geq \frac{4|\mathcal{A}|mn}{H^2 r_{\max}^2 c_0^2} \exp\left(\frac{Hr_{\max}}{m}\right),$$

and then $c_0$ can be any constant in $(0, \frac{1}{2})$.

Let $|\mathcal{S}| \geq 3, |\mathcal{A}| \geq 2, c_0 \in (0, \frac{1}{2})$ and $K \geq \frac{4|\mathcal{A}|mn}{H^2 r_{\max}^2 c_0^2} \exp(\frac{Hr_{\max}}{m})$. Since

$$
\frac{\mu'\left(\frac{Hr_{\max}}{m}\right)}{\mu'\left((1 - c_0)\frac{Hr_{\max}}{m}\right)^2} = \frac{\left(\exp\left((1 - c_0)\frac{Hr_{\max}}{m}\right) + \exp\left(-(1 - c_0)\frac{Hr_{\max}}{m}\right) + 2\right)^2}{\exp\left(\frac{Hr_{\max}}{m}\right) + \exp\left(-\frac{Hr_{\max}}{m}\right) + 2}
$$

$$
\geq \frac{\left(\exp\left((1 - c_0)\frac{Hr_{\max}}{m}\right)\right)^2}{4\exp\left(\frac{Hr_{\max}}{m}\right)}
$$

$$
= \frac{1}{4}\exp\left((1 - 2c_0)\frac{Hr_{\max}}{m}\right),
$$

we have

$$
\mathcal{R}(K) \geq \frac{1}{2Hr_{\max}}\sqrt{\frac{|\mathcal{A}|mn}{K} \cdot \frac{\mu'\left(\frac{Hr_{\max}}{m}\right)}{\mu'\left((1 - c_0)\frac{Hr_{\max}}{m}\right)^2}} \cdot r_{\max}H\left(K - \frac{K}{|\mathcal{A}|} - \frac{K}{4}\right)
$$

$$
= \Omega\left(\sqrt{\exp\left((1 - 2c_0)\frac{Hr_{\max}}{m}\right)|\mathcal{S}||\mathcal{A}|mK}\right)
$$

$$
= \Omega\left(\exp\left(\left(\frac{1}{2} - c_0\right)\frac{Hr_{\max}}{m}\right)\sqrt{|\mathcal{S}||\mathcal{A}|mK}\right).
$$

$\square$

## C.3. Pseudo-code and Detailed Description of Algorithm SegBiTS-Tran

Algorithm 3 illustrates the procedure of SegBiTS-Tran. In episode $k$, similar to SegBiTS, SegBiTS-Tran first uses MLE with past binary segment observations to obtain a reward estimate $\hat{\theta}_{k-1}$, and calculates the covariance matrix of past observations $\Sigma_{k-1}$ (Lines 3-4). After that, SegBiTS-Tran samples a Gaussian noise $\xi_k$ using $\Sigma_{k-1}$ (Line 4).

For any $k > 0$ and $(s,a) \in \mathcal{S} \times \mathcal{A}$, let $\hat{p}_k(\cdot|s,a)$ denote the empirical estimate of $p(\cdot|s,a)$, and $n_k(s,a)$ denote the number of times $(s,a)$ was visited at the end of episode $k$. Then, SegBiTS-Tran constructs a transition bonus $b_{k-1}^{pv}(s,a)$, which represents the uncertainty on transition estimation. Incorporating the MLE estimate $\hat{\theta}_{k-1}$, noise $\xi_k$ and transition bonus $b_{k-1}^{pv}(s,a)$, SegBiTS-Tran constitutes a posterior estimate of the reward parameter $\tilde{\theta}_k$ (Line 7).

For any policy $\pi$, $k > 0$ and $(s, a) \in \mathcal{S} \times \mathcal{A}$, we define

$$\hat{\phi}_k^\pi(s, a) := \mathbb{E}_{\hat{p}_k}\left[\sum_{h=1}^{H} \mathbb{1}\{s_h = s, a_h = a\}|\pi\right], \tag{18}$$

which denotes the expected number of times $(s, a)$ is visited in an episode under policy $\pi$ on the empirical MDP $\hat{p}_k$. In addition, let $\hat{\phi}_k^\pi := [\hat{\phi}_k^\pi(s, a)]_{(s,a) \in \mathcal{S} \times \mathcal{A}} \in \mathbb{R}^{|\mathcal{S}||\mathcal{A}|}$.

Then, `SegBiTS-Tran` finds the optimal policy via $\operatorname{argmax}_\pi (\hat{\phi}_{k-1}^\pi)^\top \tilde{\theta}_k^b$, which can be efficiently solved by any MDP planning algorithm with transition $\hat{p}_{k-1}$ and reward $\tilde{\theta}_k^b$ (Line 8). With the computed optimal policy $\pi^k$, `SegBiTS-Tran` plays episode $k$, and observes a trajectory and binary feedback on each segment (Line 9).

### C.4. Proof for the Regret Upper Bound with Unknown Transition

In the following, we prove the regret upper bound (Theorem 3.3) of algorithm `SegBiTS-Tran` for unknown transition.

Define event

$$\mathcal{G}_{\text{Hoeff}} := \left\{ \left|\hat{p}_{k-1}(\cdot|s, a)^\top V_{h+1}^* - p(\cdot|s, a)^\top V_{h+1}^*\right| \leq \left(2Hr_{\max}\sqrt{\frac{\log\left(\frac{KH|\mathcal{S}||\mathcal{A}|}{\delta'}\right)}{n_{k-1}(s, a)}} \wedge Hr_{\max}\right), \right.$$

$$\left. \forall(s, a) \in \mathcal{S} \times \mathcal{A}, \forall k > 0 \right\}.$$

**Lemma C.13.** *It holds that*

$$\Pr\left[\mathcal{G}_{\text{Hoeff}}\right] \geq 1 - 2\delta'.$$

*Proof.* This lemma follows from the Hoeffding inequality and a union bound over $n_{k-1}(s, a) \in [KH]$ and $(s, a) \in \mathcal{S} \times \mathcal{A}$. $\qquad \square$

**Lemma C.14** (Optimism of Thompson Sampling with Unknown Transition). *Assume that event $\mathcal{E}$ and $\mathcal{G}_{\text{Hoeff}}$ holds. Then, for any $k > 0$, we have*

$$\Pr\left[\hat{\phi}_{k-1}(\pi^k)^\top \tilde{\theta}_k^b > (\phi^{\pi^*})^\top \theta^* \mid F_{k-1}\right] \geq \frac{1}{2\sqrt{2\pi e}}.$$

*Proof.* This proof follows the analysis of Lemma 17 in (Efroni et al., 2021).

Using the value difference lemma (see Lemma E.2), we have

$$\hat{\phi}_{k-1}(\pi^*)^\top \tilde{\theta}_k^b - (\phi^{\pi^*})^\top \theta^*$$

$$= \mathbb{E}_{\hat{p}_{k-1},\pi^*}\left[\sum_{h=1}^{H}\left(\tilde{\theta}_k^b(s_h, a_h) - \theta^*(s_h, a_h) + (\hat{p}_{k-1}(\cdot|s_h, a_h) - p(\cdot|s_h, a_h))^\top V_{h+1}^*\right)\right]$$

$$= \mathbb{E}_{\hat{p}_{k-1},\pi^*}\left[\sum_{h=1}^{H}\left(\tilde{\theta}_k(s_h, a_h) - \theta^*(s_h, a_h) + b_{k-1}^{pv}(s_h, a_h) + (\hat{p}_{k-1}(\cdot|s_h, a_h) - p(\cdot|s_h, a_h))^\top V_{h+1}^*\right)\right]$$

$$\stackrel{(a)}{\geq} \mathbb{E}_{\hat{p}_{k-1},\pi^*}\left[\sum_{h=1}^{H}\left(\tilde{\theta}_k(s_h, a_h) - \theta^*(s_h, a_h) + b_{k-1}^{pv}(s_h, a_h) - b_{k-1}^{pv}(s_h, a_h)\right)\right]$$

$$= \mathbb{E}_{\hat{p}_{k-1},\pi^*}\left[\sum_{h=1}^{H}\left(\tilde{\theta}_k(s_h, a_h) - \theta^*(s_h, a_h)\right)\right]$$

$$= \hat{\phi}_{k-1}(\pi^*)^\top \tilde{\theta}_k - \hat{\phi}_{k-1}(\pi^*)^\top \theta^*,$$

where inequality (a) uses the definition of event $\mathcal{G}_{\text{Hoeff}}$.

Thus, by the definition of $\pi^k$, we have

$$
\begin{aligned}
\Pr\left[\hat{\phi}_{k-1}(\pi^k)^\top \tilde{\theta}_k^b > (\phi^{\pi^*})^\top \theta^* \mid F_{k-1}\right] &\overset{(a)}{\geq} \Pr\left[\hat{\phi}_{k-1}(\pi^*)^\top \tilde{\theta}_k^b > (\phi^{\pi^*})^\top \theta^* \mid F_{k-1}\right] \\
&= \Pr\left[\hat{\phi}_{k-1}(\pi^*)^\top \tilde{\theta}_k^b - (\phi^{\pi^*})^\top \theta^* > 0 \mid F_{k-1}\right] \\
&\geq \Pr\left[\hat{\phi}_{k-1}(\pi^*)^\top \tilde{\theta}_k - \hat{\phi}_{k-1}(\pi^*)^\top \theta^* > 0 \mid F_{k-1}\right] \\
&\overset{(b)}{\geq} \frac{1}{2\sqrt{2\pi e}},
\end{aligned}
$$

where inequality (a) is due to the definition of $\pi^k$, and inequality (b) follows from Lemma C.8. $\qquad\square$

Define event

$$
\mathcal{G}_{\text{KL}} := \left\{ \text{KL}(\hat{p}_{k-1}(\cdot|s,a), p(\cdot|s,a)) \leq \frac{L}{n_{k-1}(s,a)}, \; \forall k > 0, \forall (s,a) \in \mathcal{S} \times \mathcal{A} \right\}. \tag{19}
$$

**Lemma C.15** (Concentration of Transition). *It holds that*

$$
\Pr[\mathcal{G}_{\text{KL}}] \geq 1 - \delta'.
$$

*Proof.* This lemma can be obtained by Theorem 3 and Lemma 3 in (Ménard et al., 2021). $\qquad\square$

Recall that for any $k > 0$ and $(s,a) \in \mathcal{S} \times \mathcal{A}$, $n_k(s,a)$ denotes the cumulative number of times that $(s,a)$ is visited at the end of episode $k$. For any $k > 0$, $h \in [H]$ and $(s,a) \in \mathcal{S} \times \mathcal{A}$, let $w_{k,h}(s,a)$ denote the probability that $(s,a)$ is visited at step $h$ in episode $k$, and let $w_k(s,a) := \sum_{h=1}^H w_{k,h}(s,a)$.

Define event

$$
\mathcal{H} := \left\{ n_k(s,a) \geq \frac{1}{2} \sum_{k'=1}^k w_{k'}(s,a) - H \log\left( \frac{|\mathcal{S}||\mathcal{A}|H}{\delta'} \right), \; \forall k > 0, \forall (s,a) \in \mathcal{S} \times \mathcal{A} \right\}. \tag{20}
$$

**Lemma C.16** (Concentration of the Number of Visitations). *It holds that*

$$
\Pr[\mathcal{H}] \geq 1 - \delta'.
$$

*Proof.* This lemma can be obtained from Lemma F.4 in (Dann et al., 2017) and summing over $h \in [H]$. $\qquad\square$

Define event

$$
\begin{aligned}
\mathcal{F}_{\text{UTran}}^{\text{B}} := \Bigg\{ &\left| \sum_{k'=1}^k \left( \mathbb{E}\left[ (\phi^{\pi^{k'}})^\top b_{k'-1}^{pv} | F_{k'-1} \right] - (\phi^{\pi^{k'}})^\top b_{k'-1}^{pv} \right) \right| \leq 4H^2 r_{\max} \sqrt{k \log\left( \frac{4k}{\delta'} \right)}, \\
&\left| \sum_{k'=1}^k \left( \mathbb{E}\left[ \left\| \hat{\phi}_{k'-1}(\pi^{k'}) - \phi(\pi^{k'}) \right\|_1 | F_{k'-1} \right] - \left\| \hat{\phi}_{k'-1}(\pi^{k'}) - \phi(\pi^{k'}) \right\|_1 \right) \right| \\
&\leq 8H \sqrt{k \log\left( \frac{4k}{\delta'} \right)}, \; \forall k > 0 \Bigg\}.
\end{aligned}
$$

**Lemma C.17.** *It holds that*

$$
\Pr\left[ \mathcal{F}_{\text{UTran}}^{\text{B}} \right] \geq 1 - 2\delta'.
$$

*Proof.* This lemma can be obtained by a similar analysis as Lemma C.11, and the facts that $|(\phi^{\pi^k})^\top b_{k-1}^{pv}| \le H^2 r_{\max}$ and $\|\hat{\phi}_{k-1}(\pi^k) - \phi^{\pi^k}\|_1 \le 2H$ for any $k \ge 1$. $\qquad\square$

**Lemma C.18.** *Assume that event $\mathcal{F}_{\mathrm{UTran}}^{\mathrm{B}} \cap \mathcal{G}_{\mathrm{KL}} \cap \mathcal{H}$ holds. Then, we have*

$$\sum_{k=1}^{K} \mathbb{E}\left[\left\|\hat{\phi}_{k-1}(\pi^k) - \phi^{\pi^k}\right\|_1 \Big| F_{k-1}\right] \le 24 e^{12} |\mathcal{S}|^{\frac{3}{2}} |\mathcal{A}|^{\frac{3}{2}} H^{\frac{3}{2}} \sqrt{KL \log(2KH)}$$
$$+ 192 e^{12} |\mathcal{S}|^2 |\mathcal{A}|^2 H^2 L \log\left(\frac{2KH|\mathcal{S}||\mathcal{A}|}{\delta'}\right).$$

*Proof.* First, from Lemmas D.10 and D.11, we have

$$\sum_{k=1}^{K} \left\|\hat{\phi}_{k-1}(\pi) - \phi(\pi)\right\|_1$$

$$\le e^{12} |\mathcal{S}||\mathcal{A}| \sum_{k=1}^{K} \sum_{h=1}^{H} \sum_{(s,a)\in D_k} w_h^{\pi^k}(s,a) \left(8H\sqrt{\frac{L}{n_{k-1}(s,a)}} + \frac{46H^2 L}{n_{k-1}(s,a)}\right)$$

$$+ e^{12} |\mathcal{S}||\mathcal{A}| H \sum_{k=1}^{K} \sum_{h=1}^{H} \sum_{(s,a)\notin D_k} w_h^{\pi^k}(s,a)$$

$$\le 8 e^{12} |\mathcal{S}||\mathcal{A}| H \sqrt{L} \sqrt{\sum_{k=1}^{K} \sum_{h=1}^{H} \sum_{(s,a)\in D_k} w_h^{\pi^k}(s,a)} \sqrt{\sum_{k=1}^{K} \sum_{h=1}^{H} \sum_{(s,a)\in D_k} \frac{w_h^{\pi^k}(s,a)}{n_{k-1}(s,a)}}$$

$$+ 46 e^{12} |\mathcal{S}||\mathcal{A}| H^2 L \sum_{k=1}^{K} \sum_{h=1}^{H} \sum_{(s,a)\in D_k} \frac{w_h^{\pi^k}(s,a)}{n_{k-1}(s,a)} + 8 e^{12} |\mathcal{S}|^2 |\mathcal{A}|^2 H^2 \log\left(\frac{|\mathcal{S}||\mathcal{A}|H}{\delta'}\right)$$

$$\le 16 e^{12} |\mathcal{S}|^{\frac{3}{2}} |\mathcal{A}|^{\frac{3}{2}} H^{\frac{3}{2}} \sqrt{KL \log(2KH)} + 184 e^{12} |\mathcal{S}|^2 |\mathcal{A}|^2 H^2 L \log(2KH)$$

$$+ 8 e^{12} |\mathcal{S}|^2 |\mathcal{A}|^2 H^2 \log\left(\frac{|\mathcal{S}||\mathcal{A}|H}{\delta'}\right)$$

$$\le 16 e^{12} |\mathcal{S}|^{\frac{3}{2}} |\mathcal{A}|^{\frac{3}{2}} H^{\frac{3}{2}} \sqrt{KL \log(2KH)} + 192 e^{12} |\mathcal{S}|^2 |\mathcal{A}|^2 H^2 L \log\left(\frac{2KH|\mathcal{S}||\mathcal{A}|}{\delta'}\right).$$

Next, we have

$$\sum_{k=1}^{K} \mathbb{E}\left[\left\|\hat{\phi}_{k-1}(\pi^k) - \phi^{\pi^k}\right\|_1 \Big| F_{k-1}\right]$$

$$\le \sum_{k=1}^{K} \left\|\hat{\phi}_{k-1}(\pi^k) - \phi^{\pi^k}\right\|_1 + \sum_{k=1}^{K} \left(\mathbb{E}\left[\left\|\hat{\phi}_{k-1}(\pi^k) - \phi^{\pi^k}\right\|_1 \Big| F_{k-1}\right] - \left\|\hat{\phi}_{k-1}(\pi^k) - \phi^{\pi^k}\right\|_1\right)$$

$$\le 16 e^{12} |\mathcal{S}|^{\frac{3}{2}} |\mathcal{A}|^{\frac{3}{2}} H^{\frac{3}{2}} \sqrt{KL \log(2KH)} + 192 e^{12} |\mathcal{S}|^2 |\mathcal{A}|^2 H^2 L \log\left(\frac{2KH|\mathcal{S}||\mathcal{A}|}{\delta'}\right)$$

$$+ 8H\sqrt{K \log\left(\frac{4K}{\delta'}\right)}$$

$$\le 24 e^{12} |\mathcal{S}|^{\frac{3}{2}} |\mathcal{A}|^{\frac{3}{2}} H^{\frac{3}{2}} \sqrt{KL \log(2KH)} + 192 e^{12} |\mathcal{S}|^2 |\mathcal{A}|^2 H^2 L \log\left(\frac{2KH|\mathcal{S}||\mathcal{A}|}{\delta'}\right).$$

$\qquad\square$

**Lemma C.19.** *Assume that event $\mathcal{F}_{\mathrm{UTran}}^{\mathrm{B}}$ holds. Then, we have*

$$\sum_{k=1}^{K} \mathbb{E}\left[(\phi^{\pi^k})^\top b_{k-1}^{pv} \Big| F_{k-1}\right] \le 20 |\mathcal{S}||\mathcal{A}| H^2 r_{\max} \sqrt{K} \log\left(\frac{4KH|\mathcal{S}||\mathcal{A}|}{\delta'}\right).$$

*Proof.* It holds that

$$\sum_{k=1}^{K} \mathbb{E}\left[(\phi^{\pi^k})^\top b_{k-1}^{pv}|F_{k-1}\right]$$

$$= \sum_{k=1}^{K} (\phi^{\pi^k})^\top b_{k-1}^{pv} + \sum_{k=1}^{K}\left(\mathbb{E}\left[(\phi^{\pi^k})^\top b_{k-1}^{pv}|F_{k-1}\right] - (\phi^{\pi^k})^\top b_{k-1}^{pv}\right)$$

$$\leq \sum_{k=1}^{K}\sum_{h=1}^{H}\sum_{s,a} w_h^{\pi^k}(s,a)\left(2Hr_{\max}\sqrt{\frac{\log\left(\frac{KH|\mathcal{S}||\mathcal{A}|}{\delta'}\right)}{n_{k-1}(s,a)}} \wedge Hr_{\max}\right) + 4H^2r_{\max}\sqrt{K\log\left(\frac{4K}{\delta'}\right)}$$

$$\leq 2Hr_{\max}\sqrt{\log\left(\frac{KH|\mathcal{S}||\mathcal{A}|}{\delta'}\right)}\sum_{k=1}^{K}\sum_{h=1}^{H}\sum_{(s,a)\in D_k}\frac{w_h^{\pi^k}(s,a)}{\sqrt{n_{k-1}(s,a)}}$$

$$+ Hr_{\max}\sum_{k=1}^{K}\sum_{h=1}^{H}\sum_{(s,a)\notin D_k} w_h^{\pi^k}(s,a) + 4H^2r_{\max}\sqrt{K\log\left(\frac{4K}{\delta'}\right)}$$

$$\leq 2Hr_{\max}\sqrt{\log\left(\frac{KH|\mathcal{S}||\mathcal{A}|}{\delta'}\right)} \cdot \sqrt{KH} \cdot \sqrt{\sum_{k=1}^{K}\sum_{h=1}^{H}\sum_{(s,a)\in D_k}\frac{w_h^{\pi^k}(s,a)}{n_{k-1}(s,a)}}$$

$$+ 8|\mathcal{S}||\mathcal{A}|H^2r_{\max}\log\left(\frac{|\mathcal{S}||\mathcal{A}|H}{\delta'}\right) + 4H^2r_{\max}\sqrt{K\log\left(\frac{4K}{\delta'}\right)}$$

$$\leq 2Hr_{\max}\sqrt{\log\left(\frac{KH|\mathcal{S}||\mathcal{A}|}{\delta'}\right)} \cdot \sqrt{KH} \cdot \sqrt{4|\mathcal{S}||\mathcal{A}|\log(2KH)}$$

$$+ 8|\mathcal{S}||\mathcal{A}|H^2r_{\max}\log\left(\frac{|\mathcal{S}||\mathcal{A}|H}{\delta'}\right) + 4H^2r_{\max}\sqrt{K\log\left(\frac{4K}{\delta'}\right)}$$

$$\leq 16|\mathcal{S}||\mathcal{A}|H^2r_{\max}\sqrt{K}\log\left(\frac{4KH|\mathcal{S}||\mathcal{A}|}{\delta'}\right).$$

$\square$

*Proof of Theorem 3.3.* Letting $\delta' = \frac{\delta}{8}$, we have $\Pr[\mathcal{E} \cap \mathcal{F}_{\text{KTran}}^{\text{B}} \cap \mathcal{G}_{\text{Hoeff}} \cap \mathcal{G}_{\text{KL}} \cap \mathcal{H} \cap \mathcal{F}_{\text{UTran}}^{\text{B}}] \leq 1 - \delta$. Then, to prove this theorem, it suffices to prove the regret bound when event $\mathcal{E} \cap \mathcal{F}_{\text{KTran}}^{\text{B}} \cap \mathcal{G}_{\text{Hoeff}} \cap \mathcal{G}_{\text{KL}} \cap \mathcal{H} \cap \mathcal{F}_{\text{UTran}}^{\text{B}}$ holds.

Assume that event $\mathcal{E} \cap \mathcal{F}_{\text{KTran}}^{\text{B}} \cap \mathcal{G}_{\text{Hoeff}} \cap \mathcal{G}_{\text{KL}} \cap \mathcal{H} \cap \mathcal{F}_{\text{UTran}}^{\text{B}}$ holds. Then, we have

$$\mathcal{R}(K) = \sum_{k=1}^{K}\left((\phi^{\pi^*})^\top\theta^* - (\phi^{\pi^k})^\top\theta^*\right)$$

$$= \sum_{k=1}^{K}\left(\mathbb{E}\left[(\phi^{\pi^*})^\top\theta^* - (\phi^{\pi^k})^\top\theta^*|F_{k-1}\right] + \mathbb{E}\left[(\phi^{\pi^k})^\top\theta^*|F_{k-1}\right] - (\phi^{\pi^k})^\top\theta^*\right)$$

$$= \sum_{k=1}^{K}\left(\mathbb{E}\left[(\phi^{\pi^*})^\top\theta^* - (\phi^{\pi^k})^\top\theta^*|F_{k-1}\right]\right) + 4Hr_{\max}\sqrt{K\log\left(\frac{4K}{\delta'}\right)}. \tag{21}$$

For the first term, we have

$$\sum_{k=1}^{K}\mathbb{E}\left[(\phi^{\pi^*})^\top\theta^* - (\phi^{\pi^k})^\top\theta^*|F_{k-1}\right]$$

$$= \sum_{k=1}^{K} \left( \mathbb{E}\left[ (\phi^{\pi^*})^\top \theta^* - \hat{\phi}_{k-1}(\pi^k)^\top \tilde{\theta}_k^b | F_{k-1} \right] + \mathbb{E}\left[ \hat{\phi}_{k-1}(\pi^k)^\top \tilde{\theta}_k^b - (\phi^{\pi^k})^\top \theta^* | F_{k-1} \right] \right). \tag{22}$$

In the following, we prove

$$\mathbb{E}\left[ (\phi^{\pi^*})^\top \theta^* - \hat{\phi}_{k-1}(\pi^k)^\top \tilde{\theta}_k^b | F_{k-1} \right]$$
$$\leq 2\sqrt{2\pi e} \cdot \mathbb{E}\left[ \left( \hat{\phi}_{k-1}(\pi^k)^\top \tilde{\theta}_k^b - \mathbb{E}\left[ \hat{\phi}_{k-1}(\pi^k)^\top \tilde{\theta}_k^b | F_{k-1} \right] \right)^+ | F_{k-1} \right]. \tag{23}$$

If $\mathbb{E}[(\phi^{\pi^*})^\top \theta^* - \hat{\phi}_{k-1}(\pi^k)^\top \tilde{\theta}_k^b | F_{k-1}] < 0$, then Eq. (23) trivially holds.

Otherwise, letting $z := \mathbb{E}[(\phi^{\pi^*})^\top \theta^* - \hat{\phi}_{k-1}(\pi^k)^\top \tilde{\theta}_k^b | F_{k-1}]$, we have

$$\mathbb{E}\left[ \left( \hat{\phi}_{k-1}(\pi^k)^\top \tilde{\theta}_k^b - \mathbb{E}\left[ \hat{\phi}_{k-1}(\pi^k)^\top \tilde{\theta}_k^b | F_{k-1} \right] \right)^+ | F_{k-1} \right]$$
$$\geq z \Pr\left[ \hat{\phi}_{k-1}(\pi^k)^\top \tilde{\theta}_k^b - \mathbb{E}\left[ \hat{\phi}_{k-1}(\pi^k)^\top \tilde{\theta}_k^b | F_{k-1} \right] \geq z | F_{k-1} \right]$$
$$\geq \left( \mathbb{E}\left[ (\phi^{\pi^*})^\top \theta^* - \hat{\phi}_{k-1}(\pi^k)^\top \tilde{\theta}_k^b | F_{k-1} \right] \right) \cdot \Pr\left[ \hat{\phi}_{k-1}(\pi^k)^\top \tilde{\theta}_k^b \geq (\phi^{\pi^*})^\top \theta^* | F_{k-1} \right]$$
$$\overset{(a)}{\geq} \left( \mathbb{E}\left[ (\phi^{\pi^*})^\top \theta^* - \hat{\phi}_{k-1}(\pi^k)^\top \tilde{\theta}_k^b | F_{k-1} \right] \right) \cdot \frac{1}{2\sqrt{2\pi e}},$$

where inequality (a) uses Lemma C.14. Thus, we complete the proof of Eq. (23).

Let $\xi_k' \in \mathbb{R}^{|\mathcal{S}||\mathcal{A}|}$ be an i.i.d. random variable with $\xi$ given $F_{k-1}$. Then, using Lemma C.9 with $p' = \hat{p}_{k-1}$, $x_{k-1} = \hat{\theta}_{k-1} + b_{k-1}^{pv}$ and $\tilde{\pi}^k = \pi^k$, we have

$$\mathbb{E}\left[ (\phi^{\pi^*})^\top \theta^* - \hat{\phi}_{k-1}(\pi^k)^\top \tilde{\theta}_k^b | F_{k-1} \right]$$
$$\leq 2\sqrt{2\pi e} \cdot \mathbb{E}\left[ \left( \hat{\phi}_{k-1}(\pi^k)^\top \tilde{\theta}_k^b - \mathbb{E}\left[ \hat{\phi}_{k-1}(\pi^k)^\top \tilde{\theta}_k^b | F_{k-1} \right] \right)^+ | F_{k-1} \right]$$
$$\leq 2\sqrt{2\pi e} \cdot \mathbb{E}\left[ |\hat{\phi}_{k-1}(\pi^k)^\top \xi_k| + |\hat{\phi}_{k-1}(\pi^k)^\top \xi_k'| \; | F_{k-1} \right].$$

Plugging the above inequality into Eq. (22) and using Lemma C.10 with $\delta_k = \frac{1}{k^4}$ and $L_X = \frac{H}{\sqrt{\alpha\lambda}}$, we have

$$\sum_{k=1}^{K} \mathbb{E}\left[ (\phi^{\pi^*})^\top \theta^* - (\phi^{\pi^k})^\top \theta^* | F_{k-1} \right]$$
$$= \sum_{k=1}^{K} \left( 2\sqrt{2\pi e} \cdot \mathbb{E}\left[ |\hat{\phi}_{k-1}(\pi^k)^\top \xi_k| + |\hat{\phi}_{k-1}(\pi^k)^\top \xi_k'| \; | F_{k-1} \right] \right.$$
$$\left. + \mathbb{E}\left[ \hat{\phi}_{k-1}(\pi^k)^\top \left( \hat{\theta}_{k-1} + b_{k-1}^{pv} + \xi_k \right) - (\phi^{\pi^k})^\top \theta^* | F_{k-1} \right] \right)$$
$$= \sum_{k=1}^{K} \left( \left( 2\sqrt{2\pi e} + 1 \right) \cdot \mathbb{E}\left[ |\hat{\phi}_{k-1}(\pi^k)^\top \xi_k| \; | F_{k-1} \right] + 2\sqrt{2\pi e} \cdot \mathbb{E}\left[ |\hat{\phi}_{k-1}(\pi^k)^\top \xi_k'| \; | F_{k-1} \right] \right.$$
$$\left. + \mathbb{E}\left[ \hat{\phi}_{k-1}(\pi^k)^\top \left( \hat{\theta}_{k-1} + b_{k-1}^{pv} \right) - (\phi^{\pi^k})^\top \theta^* | F_{k-1} \right] \right)$$
$$\leq \sum_{k=1}^{K} \left( \left( 4\sqrt{2\pi e} + 1 \right) \sqrt{\alpha} \cdot \nu(k-1) \left( \sqrt{|\mathcal{S}||\mathcal{A}|} + 4\sqrt{\log(k)} \right) \mathbb{E}\left[ \left\| \hat{\phi}_{k-1}(\pi^k) \right\|_{\Sigma_{k-1}^{-1}} | F_{k-1} \right] \right.$$
$$\left. + \left( 4\sqrt{2\pi e} + 1 \right) \sqrt{\alpha} \cdot \nu(k-1) \frac{\sqrt{|\mathcal{S}||\mathcal{A}|}}{k^2} \cdot \frac{H}{\sqrt{\alpha\lambda}} \right.$$

$$+ \mathbb{E}\left[\hat{\phi}_{k-1}(\pi^k)^\top \left(\hat{\theta}_{k-1} + b_{k-1}^{pv}\right) - (\phi^{\pi^k})^\top \theta^* | F_{k-1}\right]\right). \tag{24}$$

We have

$$\mathbb{E}\left[\hat{\phi}_{k-1}(\pi^k)^\top \left(\hat{\theta}_{k-1} + b_{k-1}^{pv}\right) - (\phi^{\pi^k})^\top \theta^* | F_{k-1}\right]$$

$$= \mathbb{E}\left[\hat{\phi}_{k-1}(\pi^k)^\top \left(\hat{\theta}_{k-1} - \theta^*\right) | F_{k-1}\right] + \mathbb{E}\left[\left(\hat{\phi}_{k-1}(\pi^k) - \phi^{\pi^k}\right)^\top \theta^* | F_{k-1}\right]$$

$$+ \mathbb{E}\left[\hat{\phi}_{k-1}(\pi^k)^\top b_{k-1}^{pv} | F_{k-1}\right]$$

$$\leq \sqrt{\alpha} \cdot \nu(k-1) \mathbb{E}\left[\left\|\hat{\phi}_{k-1}(\pi^k)\right\|_{\Sigma_{k-1}^{-1}} | F_{k-1}\right] + r_{\max}\mathbb{E}\left[\left\|\hat{\phi}_{k-1}(\pi^k) - \phi^{\pi^k}\right\|_1 | F_{k-1}\right]$$

$$+ \mathbb{E}\left[\hat{\phi}_{k-1}(\pi^k)^\top b_{k-1}^{pv} | F_{k-1}\right].$$

Hence, plugging the above inequality into Eq. (24), we have

$$\sum_{k=1}^K \mathbb{E}\left[(\phi^{\pi^*})^\top \theta^* - (\phi^{\pi^k})^\top \theta^* | F_{k-1}\right]$$

$$\leq \sum_{k=1}^K \left( \left(4\sqrt{2\pi e} + 2\right)\sqrt{\alpha} \cdot \nu(k-1)\left(\sqrt{|\mathcal{S}||\mathcal{A}|} + 4\sqrt{\log(k)}\right) \cdot \mathbb{E}\left[\left\|\hat{\phi}_{k-1}(\pi^k)\right\|_{\Sigma_{k-1}^{-1}} | F_{k-1}\right]\right.$$

$$+ \left(4\sqrt{2\pi e} + 1\right) \cdot \nu(k-1)\frac{H}{k^2}\sqrt{\frac{|\mathcal{S}||\mathcal{A}|}{\lambda}} + r_{\max}\mathbb{E}\left[\left\|\hat{\phi}_{k-1}(\pi^k) - \phi^{\pi^k}\right\|_1 | F_{k-1}\right]$$

$$+ \mathbb{E}\left[\hat{\phi}_{k-1}(\pi^k)^\top b_{k-1}^{pv} | F_{k-1}\right]\right).$$

Here we have

$$\sum_{k=1}^K \mathbb{E}\left[\left\|\hat{\phi}_{k-1}(\pi^k)\right\|_{\Sigma_{k-1}^{-1}} | F_{k-1}\right]$$

$$\leq \sum_{k=1}^K \mathbb{E}\left[\left\|\phi^{\pi^k}\right\|_{\Sigma_{k-1}^{-1}} | F_{k-1}\right] + \sum_{k=1}^K \mathbb{E}\left[\left\|\hat{\phi}_{k-1}(\pi^k) - \phi^{\pi^k}\right\|_{\Sigma_{k-1}^{-1}} | F_{k-1}\right]$$

$$\leq \sum_{k=1}^K \mathbb{E}\left[\left\|\phi^{\pi^k}\right\|_{\Sigma_{k-1}^{-1}} | F_{k-1}\right] + \frac{1}{\sqrt{\alpha\lambda}}\sum_{k=1}^K \mathbb{E}\left[\left\|\hat{\phi}_{k-1}(\pi^k) - \phi^{\pi^k}\right\|_1 | F_{k-1}\right]$$

$$\overset{(a)}{\leq} 4H\sqrt{\frac{K}{\alpha\lambda}\log\left(\frac{4K}{\delta'}\right)} + \sqrt{2Km|\mathcal{S}||\mathcal{A}| \cdot \max\left\{\frac{H^2}{m\alpha\lambda}, 1\right\} \cdot \log\left(1 + \frac{KH^2}{\alpha\lambda|\mathcal{S}||\mathcal{A}|m}\right)}$$

$$+ \frac{1}{\sqrt{\alpha\lambda}}\sum_{k=1}^K \mathbb{E}\left[\left\|\hat{\phi}_{k-1}(\pi^k) - \phi^{\pi^k}\right\|_1 | F_{k-1}\right],$$

where inequality (a) uses Eq. (17).

In addition, we have

$$\sum_{k=1}^K \mathbb{E}\left[\hat{\phi}_{k-1}(\pi^k)^\top b_{k-1}^{pv} | F_{k-1}\right]$$

$$\leq \sum_{k=1}^K \mathbb{E}\left[(\phi^{\pi^k})^\top b_{k-1}^{pv} | F_{k-1}\right] + \sum_{k=1}^K \mathbb{E}\left[\left\|\hat{\phi}_{k-1}(\pi^k) - \phi^{\pi^k}\right\|_1 \left\|b_{k-1}^{pv}\right\|_\infty | F_{k-1}\right]$$

$$\leq \sum_{k=1}^{K} \mathbb{E}\left[ (\phi^{\pi^k})^\top b_{k-1}^{pv} | F_{k-1} \right] + H r_{\max} \sum_{k=1}^{K} \mathbb{E}\left[ \left\| \hat{\phi}_{k-1}(\pi^k) - \phi^{\pi^k} \right\|_1 | F_{k-1} \right].$$

Therefore, plugging the above three equations into Eq. (21), we have

$$
\begin{aligned}
\mathcal{R}(K) \leq{}& \left( 4\sqrt{2\pi e} + 2 \right) \sqrt{\alpha} \cdot \nu(K) \left( \sqrt{|\mathcal{S}||\mathcal{A}|} + 4\sqrt{\log(K)} \right) \cdot \\
& \left( 4H \sqrt{\frac{K}{\alpha \lambda} \log\left( \frac{4K}{\delta'} \right)} + \sqrt{2Km|\mathcal{S}||\mathcal{A}| \max\left\{ \frac{H^2}{m\alpha\lambda}, 1 \right\} \log\left( 1 + \frac{KH^2}{\alpha\lambda|\mathcal{S}||\mathcal{A}|m} \right)} \right) \\
& + \left( \left( 4\sqrt{2\pi e} + 2 \right) \frac{\nu(K)}{\sqrt{\lambda}} \left( \sqrt{|\mathcal{S}||\mathcal{A}|} + 4\sqrt{\log(K)} \right) + 2H r_{\max} \right) \sum_{k=1}^{K} \mathbb{E}\left[ \left\| \hat{\phi}_{k-1}(\pi^k) - \phi^{\pi^k} \right\|_1 | F_{k-1} \right] \\
& + \sum_{k=1}^{K} \mathbb{E}\left[ (\phi^{\pi^k})^\top b_{k-1}^{pv} | F_{k-1} \right] + 2 \left( 4\sqrt{2\pi e} + 1 \right) H \cdot \nu(K) \sqrt{\frac{|\mathcal{S}||\mathcal{A}|}{\lambda}} + 4H r_{\max} \sqrt{K \log\left( \frac{4K}{\delta'} \right)} \\
\overset{(a)}{=}{}& \tilde{O}\left( \exp\left( \frac{H r_{\max}}{m} \right) \nu(K) \sqrt{|\mathcal{S}||\mathcal{A}|} \left( \sqrt{Km|\mathcal{S}||\mathcal{A}| \max\left\{ \frac{H^2}{m\alpha\lambda}, 1 \right\}} + H\sqrt{\frac{K}{\alpha\lambda}} \right) \right. \\
& \left. + \left( \nu(K) \sqrt{\frac{|\mathcal{S}||\mathcal{A}|}{\lambda}} + H r_{\max} \right) |\mathcal{S}|^2 |\mathcal{A}|^{\frac{3}{2}} H^{\frac{3}{2}} \sqrt{K} \right),
\end{aligned}
$$

where in equality (a), we use Lemmas C.18 and C.19, and the last three terms are absorbed into $\tilde{O}(\cdot)$. $\qquad\square$

# D. Proofs for RL with Sum Segment Feedback

In this section, we provide the proofs for RL with sum segment feedback.

## D.1. Proof for the Regret Upper Bound with Known Transition

We first prove the regret upper bound (Theorem 4.1) of algorithm E-LinUCB for known transition.

Define event

$$
\begin{aligned}
\mathcal{J} := \Bigg\{ & \left\| \sum_{k=1}^{K_0} \left( \sum_{i=1}^{m} \phi^{\tau_i^k}(\phi^{\tau_i^k})^\top - \mathbb{E}_{\tau_i \sim \pi^k}\left[ \sum_{i=1}^{m} \phi(\tau_i)\phi(\tau_i)^\top \right] \right) \right\| \\
& \leq \frac{4H^2}{m} \sqrt{K_0 \log\left( \frac{2|\mathcal{S}||\mathcal{A}|}{\delta'} \right)} + \frac{4H^2}{m} \log\left( \frac{2|\mathcal{S}||\mathcal{A}|}{\delta'} \right) \Bigg\}. 
\end{aligned}
\tag{25}
$$

**Lemma D.1** (Concentration of Initial Sampling). *It holds that*

$$\Pr[\mathcal{J}] \geq 1 - \delta'.$$

*Proof.* Note that $\pi^1, \ldots, \pi^{K_0}$ and $K_0$ are fixed before sampling, $\mathbb{E}[\sum_{i=1}^{m} \phi^{\tau_i^k}(\phi^{\tau_i^k})^\top] = \mathbb{E}_{\tau_i \sim \pi^k}\left[ \sum_{i=1}^{m} \phi(\tau_i)\phi(\tau_i)^\top \right]$, and $\| \sum_{i=1}^{m} \phi^{\tau_i^k}(\phi^{\tau_i^k})^\top \| \leq \frac{H^2}{m}$. Then, using the matrix Bernstein inequality (Theorem 6.1.1 in (Tropp et al., 2015)), we can obtain this lemma. $\qquad\square$

**Lemma D.2** (E-optimal Design). *Assume that event $\mathcal{J}$ holds. Then, we have*

$$\left\| \left( \sum_{k=1}^{K_0} \sum_{i=1}^{m} \phi^{\tau_i^k}(\phi^{\tau_i^k})^\top \right)^{-1} \right\| \leq \frac{1}{H^2}.$$

*Proof.* Using the guarantee of the rounding procedure ROUND (Theorem 1.1 in (Allen-Zhu et al., 2021)) and the fact that $K_0 \geq \frac{|\mathcal{S}||\mathcal{A}|}{\gamma^2}$, we have

$$
\left\| \left( \sum_{k=1}^{K_0} \mathbb{E}_{\tau_i \sim \pi^k} \left[ \sum_{i=1}^{m} \phi(\tau_i)\phi(\tau_i)^\top \right] \right)^{-1} \right\|
$$

$$
\leq (1+\gamma) \left\| \left( K_0 \sum_{\pi \in \Pi} w^*(\pi) \cdot \mathbb{E}_{\tau_i \sim \pi^k} \left[ \sum_{i=1}^{m} \phi(\tau_i)\phi(\tau_i)^\top \right] \right)^{-1} \right\|
$$

$$
\leq \frac{(1+\gamma)z^*}{K_0}.
$$

Let $\sigma_{\min}(\cdot)$ denote the minimum eigenvalue. Then, we have

$$
\sigma_{\min} \left( \sum_{k=1}^{K_0} \sum_{i=1}^{m} \phi^{\tau_i^k}(\phi^{\tau_i^k})^\top \right)
$$

$$
= \sigma_{\min} \left( \sum_{k=1}^{K_0} \mathbb{E}_{\tau_i \sim \pi^k} \left[ \sum_{i=1}^{m} \phi(\tau_i)\phi(\tau_i)^\top \right] + \sum_{k=1}^{K_0} \sum_{i=1}^{m} \phi^{\tau_i^k}(\phi^{\tau_i^k})^\top - \sum_{k=1}^{K_0} \mathbb{E}_{\tau_i \sim \pi^k} \left[ \sum_{i=1}^{m} \phi(\tau_i)\phi(\tau_i)^\top \right] \right)
$$

$$
\geq \sigma_{\min} \left( \sum_{k=1}^{K_0} \mathbb{E}_{\tau_i \sim \pi^k} \left[ \sum_{i=1}^{m} \phi(\tau_i)\phi(\tau_i)^\top \right] \right) - \left\| \sum_{k=1}^{K_0} \sum_{i=1}^{m} \phi^{\tau_i^k}(\phi^{\tau_i^k})^\top - \sum_{k=1}^{K_0} \mathbb{E}_{\tau_i \sim \pi^k} \left[ \sum_{i=1}^{m} \phi(\tau_i)\phi(\tau_i)^\top \right] \right\|
$$

$$
\geq \frac{K_0}{(1+\gamma)z^*} - \frac{4H^2}{m} \sqrt{\log\left(\frac{2|\mathcal{S}||\mathcal{A}|}{\delta'}\right)} \cdot \sqrt{K_0} - \frac{4H^2}{m} \log\left(\frac{2|\mathcal{S}||\mathcal{A}|}{\delta'}\right). \tag{26}
$$

Let $x = \sqrt{K_0}$ and

$$
f(x) = \frac{1}{(1+\gamma)z^*} \cdot x^2 - \frac{4H^2}{m} \sqrt{\log\left(\frac{2|\mathcal{S}||\mathcal{A}|}{\delta'}\right)} \cdot x - \frac{4H^2}{m} \log\left(\frac{2|\mathcal{S}||\mathcal{A}|}{\delta'}\right) - H^2.
$$

According to the property of quadratic functions, when

$$
x \geq \frac{\frac{4H^2}{m}\sqrt{\log\left(\frac{2|\mathcal{S}||\mathcal{A}|}{\delta'}\right)} + \sqrt{\left(\frac{4H^2}{m}\sqrt{\log\left(\frac{2|\mathcal{S}||\mathcal{A}|}{\delta'}\right)}\right)^2 + 4 \cdot \frac{1}{(1+\gamma)z^*}\left(\frac{4H^2}{m}\log\left(\frac{2|\mathcal{S}||\mathcal{A}|}{\delta'}\right) + H^2\right)}}{2 \cdot \frac{1}{(1+\gamma)z^*}}, \tag{27}
$$

we have $f(x) \geq 0$.

To make Eq. (27) hold, it suffices to set

$$
K_0 \geq \frac{(1+\gamma)^2(z^*)^2}{4} \cdot \left( 2 \cdot \left(\frac{4H^2}{m}\sqrt{\log\left(\frac{2|\mathcal{S}||\mathcal{A}|}{\delta'}\right)}\right)^2 + 2 \cdot \left(\frac{4H^2}{m}\sqrt{\log\left(\frac{2|\mathcal{S}||\mathcal{A}|}{\delta'}\right)}\right)^2 \right.
$$

$$
\left. + \frac{8}{(1+\gamma)z^*} \cdot 5H^2 \log\left(\frac{2|\mathcal{S}||\mathcal{A}|}{\delta'}\right) \right)
$$

$$
= \left( \frac{16H^4(1+\gamma)^2(z^*)^2}{m^2} + 10H^2(1+\gamma)z^* \right) \log\left(\frac{2|\mathcal{S}||\mathcal{A}|}{\delta'}\right).
$$

Furthermore, since $\left\| \sum_{\pi \in \Pi} w^*(\pi) \mathbb{E}_{\tau_i \sim \pi^k} \left[ \sum_{i=1}^{m} \phi(\tau_i)\phi(\tau_i)^\top \right] \right\| \leq H^2$ and then $z^* \geq \frac{1}{H^2}$, to make the right-hand-side in Eq. (26) no smaller than $H^2$, it suffices to set

$$
K_0 \geq 26H^4(1+\gamma)^2(z^*)^2 \log\left(\frac{2|\mathcal{S}||\mathcal{A}|}{\delta'}\right).
$$

Therefore, combining the definition of $K_0$ and Eq. (26), we have

$$\sigma_{\min}\left(\sum_{k=1}^{K_0}\phi(\tau^k)\phi(\tau^k)^\top\right) \geq H^2,$$

which completes the proof. $\square$

**Lemma D.3.** *For any $k > 0$,*

$$\sum_{k'=1}^{k}\log\left(1+\sum_{i=1}^{m}\left\|\phi^{\tau_i^{k'}}\right\|_{(\Sigma_{k'-1})^{-1}}^2\right) = \log\left(\frac{\det(\Sigma_k)}{\det(\lambda I)}\right) \leq |\mathcal{S}||\mathcal{A}|\log\left(1+\frac{kH^2}{\lambda|\mathcal{S}||\mathcal{A}|m}\right).$$

*Proof.* For any $k > 0$, it holds that

$$
\begin{aligned}
\det(\Sigma_k) &= \det\left(\Sigma_{k-1}+\sum_{i=1}^{m}\phi^{\tau_i^k}(\phi^{\tau_i^k})^\top\right)\\
&= \det(\Sigma_{k-1})\det\left(I+\sum_{i=1}^{m}(\Sigma_{k-1})^{-\frac{1}{2}}\phi^{\tau_i^k}(\phi^{\tau_i^k})^\top(\Sigma_{k-1})^{-\frac{1}{2}}\right)\\
&= \det(\Sigma_{k-1})\left(1+\sum_{i=1}^{m}\left\|\phi^{\tau_i^k}\right\|_{(\Sigma_{k-1})^{-1}}^2\right)\\
&= \det(\lambda I)\prod_{k'=1}^{k}\left(1+\sum_{i=1}^{m}\left\|\phi^{\tau_i^{k'}}\right\|_{(\Sigma_{k'-1})^{-1}}^2\right).
\end{aligned}
$$

Taking the logarithm on both sides, we have

$$\log\det(\Sigma_k) = \log\det(\lambda I) + \sum_{k'=1}^{k}\log\left(1+\sum_{i=1}^{m}\left\|\phi^{\tau_i^{k'}}\right\|_{(\Sigma_{k'-1})^{-1}}^2\right).$$

Then,

$$
\begin{aligned}
\sum_{k'=1}^{k}\log\left(1+\sum_{i=1}^{m}\left\|\phi^{\tau_i^{k'}}\right\|_{(\Sigma_{k'-1})^{-1}}^2\right) &= \log\left(\frac{\det(\Sigma_k)}{\det(\lambda I)}\right)\\
&\overset{(a)}{\leq} \log\left(\frac{\left(\frac{\operatorname{tr}(\Sigma_k)}{|\mathcal{S}||\mathcal{A}|}\right)^{|\mathcal{S}||\mathcal{A}|}}{\lambda^{|\mathcal{S}||\mathcal{A}|}}\right)\\
&= |\mathcal{S}||\mathcal{A}|\log\left(\frac{\operatorname{tr}(\Sigma_k)}{\lambda|\mathcal{S}||\mathcal{A}|}\right)\\
&\leq |\mathcal{S}||\mathcal{A}|\log\left(\frac{\lambda|\mathcal{S}||\mathcal{A}|+km\cdot\frac{H^2}{m^2}}{\lambda|\mathcal{S}||\mathcal{A}|}\right)\\
&= |\mathcal{S}||\mathcal{A}|\log\left(1+\frac{kH^2}{\lambda|\mathcal{S}||\mathcal{A}|m}\right),
\end{aligned}
$$

where (a) uses the arithmetic mean-geometric mean inequality. $\square$

**Lemma D.4** (Elliptical Potential with Optimized Initialization). *Assume that event $\mathcal{J}$ holds. Then, for any $k \geq K_0 + 1$,*

$$\sum_{i=1}^{m}\left\|\phi^{\tau_i^k}\right\|_{(\Sigma_{k-1})^{-1}}^2 \leq 1.$$

*Furthermore, for any $K \geq K_0 + 1$,*

$$\sum_{k=K_0+1}^{K} \sum_{i=1}^{m} \left\| \phi^{\tau_i^k} \right\|_{(\Sigma_{k-1})^{-1}} \leq \sqrt{2Km|\mathcal{S}||\mathcal{A}| \log\left(1 + \frac{KH^2}{\lambda|\mathcal{S}||\mathcal{A}|m}\right)}.$$

*Proof.* Using Lemma D.2, for any $k \geq K_0 + 1$, we have

$$\sum_{i=1}^{m} \left\| \phi^{\tau_i^k} \right\|_{(\Sigma_{k-1})^{-1}}^2 = \sum_{i=1}^{m} \left\| \phi^{\tau_i^k} \right\|_{\left(\lambda I + \sum_{k'=1}^{K_0} \phi^{\tau^{k'}}(\phi^{\tau^{k'}})^\top + \sum_{k'=K_0+1}^{k-1} \phi^{\tau^{k'}}(\phi^{\tau^{k'}})^\top\right)^{-1}}^2$$

$$\leq \sum_{i=1}^{m} \left\| \phi^{\tau_i^i} \right\|_{\left(\sum_{k'=1}^{K_0} \phi^{\tau^{k'}}(\phi^{\tau^{k'}})^\top\right)^{-1}}^2$$

$$\leq m \cdot \frac{H^2}{m^2} \cdot \frac{1}{H^2}$$

$$\leq 1.$$

Then, we have

$$\sum_{k=K_0+1}^{K} \sum_{i=1}^{m} \left\| \phi^{\tau_i^k} \right\|_{(\Sigma_{k-1})^{-1}} \leq \sqrt{Km \sum_{k=K_0+1}^{K} \sum_{i=1}^{m} \left\| \phi^{\tau_i^k} \right\|_{(\Sigma_{k-1})^{-1}}^2}$$

$$\overset{(a)}{\leq} \sqrt{Km \cdot 2 \sum_{k=K_0+1}^{K} \log\left(1 + \sum_{i=1}^{m} \left\| \phi^{\tau_i^k} \right\|_{(\Sigma_{k-1})^{-1}}^2\right)}$$

$$\leq \sqrt{Km \cdot 2 \sum_{k=1}^{K} \log\left(1 + \sum_{i=1}^{m} \left\| \phi^{\tau_i^k} \right\|_{(\Sigma_{k-1})^{-1}}^2\right)}$$

$$\overset{(b)}{\leq} \sqrt{2Km|\mathcal{S}||\mathcal{A}| \log\left(1 + \frac{KH^2}{\lambda|\mathcal{S}||\mathcal{A}|m}\right)},$$

where (a) uses the fact that $x \leq 2\log(1+x)$ for any $x \in [0,1]$, and (b) follows from Lemma D.3. $\qquad\square$

Define event

$$\mathcal{K} := \left\{ \left\| \hat{\theta}_k - \theta^* \right\|_{\Sigma_k} \leq \sqrt{\frac{H|\mathcal{S}||\mathcal{A}|}{m} \log\left(1 + \frac{kH^2}{\lambda|\mathcal{S}||\mathcal{A}|m}\right) + 2\log\left(\frac{1}{\delta'}\right)} + r_{\max}\sqrt{\lambda|\mathcal{S}||\mathcal{A}|}, \ \forall k > 0 \right\}. \tag{28}$$

**Lemma D.5** (Concentration of $\hat{\theta}_k$ under Sum Feedback). *It holds that*

$$\Pr\left[\mathcal{K}\right] \geq 1 - \delta'.$$

*Proof.* Since the sum feedback on each segment is $\frac{H}{m}$-sub-Gaussian given the observation of transition and $\|\theta^*\| \leq r_{\max}\sqrt{|\mathcal{S}||\mathcal{A}|}$, using Lemma 2 in (Abbasi-Yadkori et al., 2011), we can obtain this lemma. $\qquad\square$

Define event

$$\mathcal{F}_{\text{opt}}^{\text{S}} := \left\{ \left| \sum_{k'=K_0+1}^{k} \left( \mathbb{E}_{\tau \sim \pi^{k'}} \left[ \|\phi^\tau\|_{(\Sigma_{k'-1})^{-1}} \mid F_{k'-1} \right] - \|\phi^\tau\|_{(\Sigma_{k'-1})^{-1}} \right) \right| \right.$$

$$\left. \leq 4\sqrt{k \log\left(\frac{4k}{\delta'}\right)}, \ \forall k \geq K_0 + 1 \right\}. \tag{29}$$

**Lemma D.6** (Concentration of Visitation Indicators). *It holds that*

$$\Pr\left[\mathcal{F}_{\mathrm{opt}}^{\mathrm{S}}\right] \geq 1 - \delta'.$$

*Proof.* According to Lemma D.4, we have that for any $k' \geq K_0 + 1$, $\|\phi^\tau\|_{(\Sigma_{k'-1})^{-1}} \leq 1$, and then $|\mathbb{E}_{\tau \sim \pi^{k'}}[\|\phi^\tau\|_{(\Sigma_{k'-1})^{-1}} | F_{k'-1}] - \|\phi^\tau\|_{(\Sigma_{k'-1})^{-1}}| \leq 2$.

Using the Azuma-Hoeffding inequality, we have that for any fixed $k \geq K_0 + 1$, with probability at least $1 - \frac{\delta'}{2k^2}$,

$$\left| \sum_{k'=K_0+1}^{k} \left( \mathbb{E}_{\tau \sim \pi^{k'}}\left[\|\phi^\tau\|_{(\Sigma_{k'-1})^{-1}} | F_{k'-1}\right] - \|\phi^\tau\|_{(\Sigma_{k'-1})^{-1}} \right) \right| \leq \sqrt{2 \cdot 4(k - K_0 - 1) \log\left(\frac{4k^2}{\delta'}\right)}.$$

Since $\sum_{k=K_0+1}^{\infty} \frac{\delta'}{2k^2} \leq \delta'$, by a union bound over $k$, we have that with probability at least $\delta'$, for any $k \geq K_0 + 1$,

$$\left| \sum_{k'=K_0+1}^{k} \left( \mathbb{E}_{\tau \sim \pi^{k'}}\left[\|\phi^\tau\|_{(\Sigma_{k'-1})^{-1}} | F_{k'-1}\right] - \|\phi^\tau\|_{(\Sigma_{k'-1})^{-1}} \right) \right| \leq \sqrt{2 \cdot 4(k - K_0 - 1) \log\left(\frac{4k^2}{\delta'}\right)}$$

$$\leq 4\sqrt{k \log\left(\frac{4k}{\delta'}\right)}.$$

$\square$

*Proof of Theorem 4.1.* Let $\delta' = \frac{\delta}{3}$. We have $\Pr[\mathcal{J} \cap \mathcal{K} \cap \mathcal{F}_{\mathrm{opt}}^{\mathrm{S}}] \geq 1 - \delta$. To prove this theorem, it suffices to prove the regret bound when event $\mathcal{J} \cap \mathcal{K} \cap \mathcal{F}_{\mathrm{opt}}^{\mathrm{S}}$ holds.

Assume that event $\mathcal{J} \cap \mathcal{K} \cap \mathcal{F}_{\mathrm{opt}}^{\mathrm{S}}$ holds. Then, we have

$$\mathcal{R}(K) = \sum_{k=1}^{K} \left( (\phi^{\pi^*})^\top \theta - (\phi^{\pi^k})^\top \theta \right)$$

$$\overset{(a)}{\leq} \sum_{k=K_0+1}^{K} \left( (\phi^{\pi^*})^\top \hat{\theta}_{k-1} + \beta(k-1) \cdot \|\phi^{\pi^*}\|_{(\Sigma_{k-1})^{-1}} - (\phi^{\pi^k})^\top \theta \right) + K_0 H$$

$$\overset{(b)}{\leq} \sum_{k=K_0+1}^{K} \left( (\phi^{\pi^k})^\top \hat{\theta}_{k-1} + \beta(k-1) \cdot \|\phi^{\pi^k}\|_{(\Sigma_{k-1})^{-1}} - (\phi^{\pi^k})^\top \theta \right) + K_0 H$$

$$\leq \sum_{k=K_0+1}^{K} 2\beta(k-1) \cdot \|\phi^{\pi^k}\|_{(\Sigma_{k-1})^{-1}} + K_0 H$$

$$= 2\beta(K) \sum_{k=K_0+1}^{K} \|\mathbb{E}_{\tau \sim \pi^k}[\phi^\tau | F_{k-1}]\|_{(\Sigma_{k-1})^{-1}} + K_0 H$$

$$\overset{(c)}{\leq} 2\beta(K) \sum_{k=K_0+1}^{K} \mathbb{E}_{\tau \sim \pi^k}\left[\|\phi^\tau\|_{(\Sigma_{k-1})^{-1}} | F_{k-1}\right] + K_0 H$$

$$= 2\beta(K) \sum_{k=K_0+1}^{K} \left( \mathbb{E}_{\tau \sim \pi^k}\left[\|\phi^\tau\|_{(\Sigma_{k-1})^{-1}} | F_{k-1}\right] - \|\phi^\tau\|_{(\Sigma_{k-1})^{-1}} + \|\phi^\tau\|_{(\Sigma_{k-1})^{-1}} \right) + K_0 H$$

$$\leq 2\beta(K) \sum_{k=K_0+1}^{K} \left( \mathbb{E}_{\tau \sim \pi^k}\left[\|\phi^\tau\|_{(\Sigma_{k-1})^{-1}} | F_{k-1}\right] - \|\phi^\tau\|_{(\Sigma_{k-1})^{-1}} + \sum_{i=1}^{m} \left\|\phi^{\tau_i^k}\right\|_{(\Sigma_{k-1})^{-1}} \right)$$

$$+ K_0 H, \tag{30}$$

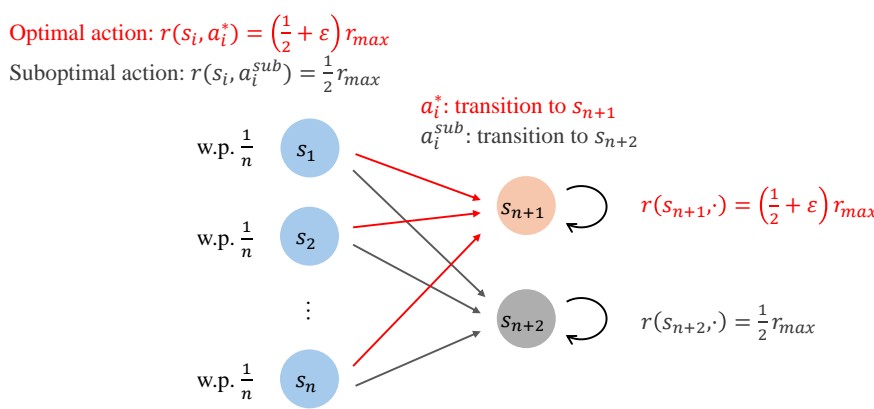

*Figure 6.* Instance for the lower bound under sum segment feedback and known transition.

where inequality (a) follows from Eq. (28), inequality (b) is due to the definition of $\pi^k$, and inequality (c) uses the Jensen inequality.

Plugging Eq. (29) and Lemma D.4 into Eq. (30) and using the fact that $\lambda := \frac{H}{r_{\max}^2 m}$, we have

$$
\mathcal{R}(K) \leq 2 \left( \sqrt{\frac{H|\mathcal{S}||\mathcal{A}|}{m} \log\left(1 + \frac{KH^2}{\lambda|\mathcal{S}||\mathcal{A}|m}\right) + 2\log\left(\frac{1}{\delta'}\right)} + r_{\max}\sqrt{\lambda|\mathcal{S}||\mathcal{A}|} \right) \cdot
$$
$$
\left( 4\sqrt{K \log\left(\frac{4K}{\delta}\right)} + \sqrt{2Km|\mathcal{S}||\mathcal{A}| \log\left(1 + \frac{KH^2}{\lambda|\mathcal{S}||\mathcal{A}|m}\right)} \right)
$$
$$
+ H\left[ \max\left\{ 26H^4(1+\gamma)^2(z^*)^2 \log\left(\frac{2|\mathcal{S}||\mathcal{A}|}{\delta'}\right), \frac{|\mathcal{S}||\mathcal{A}|}{\gamma^2} \right\} \right]
$$
$$
= O\left( |\mathcal{S}||\mathcal{A}|\sqrt{HK} \log\left( \left(1 + \frac{KHr_{\max}^2}{|\mathcal{S}||\mathcal{A}|m}\right)\frac{1}{\delta}\right) + (z^*)^2 H^5 \log\left(\frac{|\mathcal{S}||\mathcal{A}|}{\delta}\right) + |\mathcal{S}||\mathcal{A}|H \right).
$$

$\square$

### D.2. Proof for the Regret Lower Bound with Known Transition

Now we prove the regret lower bound (Theorem 4.2) for RL with sum segment feedback and known transition.

*Proof of Theorem 4.2.* We construct a random instance $\mathcal{I}$ as follows. As shown in Figure 6, there are $n$ bandit states $s_1, \ldots, s_n$ (i.e., there is an optimal action and multiple suboptimal actions), a good absorbing state $s_{n+1}$ and a bad absorbing state $s_{n+2}$. The agent starts from $s_1, \ldots, s_n$ with equal probability $\frac{1}{n}$. For any $i \in [n]$, in state $s_i$, one action $a_J$ is uniformly chosen from $\mathcal{A}$ as the optimal action. In state $s_i$, under the optimal action $a_J$, the agent transitions to $s_{n+1}$ deterministically, and $r(s_i, a_J) = (\frac{1}{2} + \varepsilon)r_{\max}$, where $\varepsilon \in (0, \frac{1}{2}]$ is a parameter specified later; Under any suboptimal action $a \in \mathcal{A} \setminus \{s_J\}$, the agent transitions to $s_{n+2}$ deterministically, and $r(s_i, a) = \frac{1}{2}r_{\max}$. For all actions $a \in \mathcal{A}$, $r(s_{n+1}, a) = (\frac{1}{2} + \varepsilon)r_{\max}$ and $r(s_{n+2}, a) = \frac{1}{2}r_{\max}$. For any $(s, a) \in \mathcal{S} \times \mathcal{A}$, the reward distribution of $(s, a)$ is Gaussian distribution $\mathcal{N}(r(s, a), 1)$.

In this proof, we will also use an alternative uniform instance $\mathcal{I}_{\text{unif}}$. The only difference between $\mathcal{I}_{\text{unif}}$ and $\mathcal{I}$ is that for any $i \in [n]$, in state $s_i$, under all actions $a \in \mathcal{A}$, the agent transitions to $s_{n+2}$ deterministically, and $r(s_i, a) = \frac{1}{2}r_{\max}$.

Fix an algorithm $\mathbb{A}$. Let $\mathbb{E}_{\text{unif}}[\cdot]$ denote the expectation with respect to $\mathcal{I}_{\text{unif}}$. Let $\mathbb{E}_*[\cdot]$ denote the expectation with respect to $\mathcal{I}$. For any $i \in [n]$ and $j \in [|\mathcal{A}|]$, let $\mathbb{E}_{i,j}[\cdot]$ denote the expectation with respect to the case where $a_j$ is the optimal action in state $s_i$, and $N_{i,j}$ denote the number of episodes where algorithm $\mathbb{A}$ chooses $a_j$ in state $s_i$, i.e., $N_{i,j} = \sum_{k=1}^K \mathbb{1}\{\pi_1^k(s_i) = a_j\}$.

The KL divergence of the reward observations if taking $a_J$ in $s_i$ ($i \in [n]$) between $\mathcal{I}_{\text{unif}}$ and $\mathcal{I}$ is

$$
\sum_{i=1}^m \text{KL}\left( \mathcal{N}\left(\frac{1}{2}r_{\max} \cdot \frac{H}{m}, \frac{H}{m}\right) \,\middle\|\, \mathcal{N}\left(\left(\frac{1}{2} + \varepsilon\right)r_{\max} \cdot \frac{H}{m}, \frac{H}{m}\right) \right)
$$

---

**Algorithm 4** `LinUCB-Tran`

---

1: **Input:** $\delta, \delta' := \frac{\delta}{4}, \lambda := \frac{H}{m}, L := \log(\frac{3|\mathcal{S}||\mathcal{A}|H}{\delta'}) + S \log(8e(1 + KH))$. For any $k \geq 1$, $\beta(k) :=$ $\sqrt{\frac{H|\mathcal{S}||\mathcal{A}|}{m} \log(1 + \frac{kH^2}{\lambda|\mathcal{S}||\mathcal{A}|m}) + 2\log(\frac{1}{\delta'})} + r_{\max}\sqrt{\lambda|\mathcal{S}||\mathcal{A}|}$.

2: **for** $k = 1, \ldots, K$ **do**

3:     $\hat{\theta}_{k-1} \leftarrow (\lambda I + \sum_{k'=1}^{k-1} \sum_{i=1}^{m} \phi^{\tau_i^{k'}}(\phi^{\tau_i^{k'}})^\top)^{-1} \sum_{k'=1}^{k-1} \sum_{i=1}^{m} \phi^{\tau_i^{k'}} R_i^{k'}$

4:     $\Sigma_{k-1} \leftarrow \lambda I + \sum_{k'=1}^{k-1} \sum_{i=1}^{m} \phi^{\tau_i^{k'}}(\phi^{\tau_i^{k'}})^\top$

5:     $\pi^k \leftarrow \operatorname{argmax}_{\pi \in \Pi}((\hat{\phi}_{k-1}^\pi)^\top \hat{\theta}_{k-1} + \beta(k-1) \cdot \|\hat{\phi}_{k-1}^\pi\|_{(\Sigma_{k-1})^{-1}} + \sum_{s',a'} \mathbb{E}_{s_1 \sim \rho}[B_1^{\pi;s',a';k}(s_1)])$, where $B_1^{\pi;s',a';k}(s_1)$ is defined in Eq. (32)

6:     Play episode $k$ with policy $\pi^k$. Observe $\tau^k$ and sum segment feedback $\{R_i^k\}_{i=1}^m$

7: **end for**

---

$$= m \cdot \frac{\left(\frac{H}{m} \cdot r_{\max}\varepsilon\right)^2}{\frac{H}{m}}$$
$$= Hr_{\max}^2\varepsilon^2.$$

In addition, the agent has probability only $\frac{1}{n}$ to arrive at (observe) state $s_i$.

Hence, using Lemma A.1 in (Auer et al., 2002), we have that for any $i \in [n]$, in state $s_i$,

$$\mathbb{E}_{i,j}[N_{i,j}] \leq \mathbb{E}_{\text{unif}}[N_{i,j}] + \frac{K}{2}\sqrt{\frac{1}{n} \cdot \mathbb{E}_{\text{unif}}[N_{i,j}] \cdot Hr_{\max}^2\varepsilon^2}.$$

Summing over $j \in [|\mathcal{A}|]$, using the Cauchy-Schwarz inequality and the fact that $\sum_{j=1}^{|\mathcal{A}|} \mathbb{E}_{\text{unif}}[N_{i,j}] = K$, we have

$$\sum_{j=1}^{|\mathcal{A}|} \mathbb{E}_{i,j}[N_{i,j}] \leq K + \frac{K}{2}\sqrt{\frac{|\mathcal{A}|}{n} \cdot K \cdot Hr_{\max}^2\varepsilon^2}$$
$$= K + \frac{Kr_{\max}\varepsilon}{2}\sqrt{\frac{|\mathcal{A}|HK}{n}}.$$

Then, we have

$$\mathcal{R}(K) = \sum_{k=1}^{K} \mathbb{E}_* \left[V^* - V^{\pi^k}\right]$$
$$= \left(\frac{1}{2} + \varepsilon\right) r_{\max}HK - \frac{1}{n}\sum_{i=1}^{n}\left(\frac{1}{2}r_{\max}HK + \varepsilon r_{\max}H \cdot \frac{1}{|\mathcal{A}|}\sum_{j=1}^{|\mathcal{A}|} \mathbb{E}_{i,j}[N_{i,j}]\right)$$
$$= \varepsilon r_{\max}H\left(K - \frac{K}{|\mathcal{A}|} - \frac{Kr_{\max}\varepsilon}{2}\sqrt{\frac{HK}{n|\mathcal{A}|}}\right).$$

Recall that $n = |\mathcal{S}| - 2$. Let $|\mathcal{S}| \geq 3$, $|\mathcal{A}| \geq 2$, $K \geq \frac{n|\mathcal{A}|}{r_{\max}^2 H}$ and $\varepsilon = \frac{1}{2r_{\max}}\sqrt{\frac{n|\mathcal{A}|}{HK}}$. Then, we have

$$\mathcal{R}(K) = \Omega\left(\sqrt{|\mathcal{S}||\mathcal{A}|HK}\right).$$

□

## D.3. Pseudo-code and Detailed Description of Algorithm `LinUCB-Tran`

Algorithm 4 presents the pseudo-code of `LinUCB-Tran`. In each episode $k$, similar to algorithm `E-LinUCB`, `LinUCB-Tran` first computes the least squares estimate of the reward parameter $\hat{\theta}_{k-1}$ and covariance matrix $\Sigma_{k-1}$ with past observations (Lines 3-4).

Then, we introduce the transition estimation in `LinUCB-Tran`. We first define some notation which also appears in algorithm `SegBiTS-Tran`. For any $k > 0$ and $(s, a) \in \mathcal{S} \times \mathcal{A}$, let $\hat{p}_k(\cdot|s, a)$ denote the empirical estimate of $p(\cdot|s, a)$, and $n_k(s, a)$ denote the number of times $(s, a)$ was visited up to the end of episode $k$. In addition, for any policy $\pi$, let $\hat{\phi}_k^\pi(s, a)$ denote the expected number of times $(s, a)$ is visited in an episode under policy $\pi$ on empirical MDP $\hat{p}_{k-1}$ (see Eq. (18) for the formal definition).

Below we establish a bound for the deviation between $\hat{\phi}_{k-1}^\pi$ and $\phi^\pi$. For ease of analysis, we first connect $\phi^\pi$ with a newly-defined visitation value function $G_h^{\pi; s', a'}(s; p)$. For any transition model $p'$, policy $\pi$ and $(s', a') \in \mathcal{S} \times \mathcal{A}$, if regarding hitting $(s', a')$ as an instantaneous reward one, then we can define a visitation value function:

$$\begin{cases} G_h^{\pi; s', a'}(s; p') = & \mathbb{1}\{s = s', \pi_h(s) = a'\} + p(\cdot|s, \pi_h(s))^\top G_{h+1}^{\pi; s', a'}(\cdot), & \forall s \in \mathcal{S}, \forall h \in [H], \\ G_{H+1}^{\pi; s', a'}(s; p') = & 0, & \forall s \in \mathcal{S}. \end{cases} \tag{31}$$

$G_h^{\pi; s', a'}(s; p')$ denotes the expected cumulative number of times $(s', a')$ was hit starting from $s$ at step $h$ under policy $\pi$ on MDP $p'$, till the end of this episode. It holds that $\phi^\pi(s', a') = \mathbb{E}_{s_1 \sim \rho}[G_1^{\pi; s', a'}(s_1|p)]$ and $\hat{\phi}_{k-1}^\pi(s', a') = \mathbb{E}_{s_1 \sim \rho}[G_1^{\pi; s', a'}(s_1|\hat{p}_{k-1})]$ for any $(s', a') \in \mathcal{S} \times \mathcal{A}$.

With the definition of $G_h^{\pi; s', a'}$, bounding the deviation between $\hat{\phi}_{k-1}^\pi$ and $\phi^\pi$ is similar to bounding the gap between the estimated and true value functions. Then, we can build a Bernstern-type uncertainty bound between $\hat{\phi}_{k-1}^\pi$ and $\phi^\pi$ using the variance of $G_h^{\pi; s', a'}$. For any policy $\pi$, $(s', a') \in \mathcal{S} \times \mathcal{A}$ and $k > 0$, define

$$\begin{cases} B_h^{\pi; s', a'; k}(s) = & \min\left\{ \left( 4\sqrt{\frac{\mathrm{Var}_{\hat{p}_{k-1}(\cdot|s, \pi_h(s))}(G_{h+1}^{\pi; s', a'}(\cdot|\hat{p}_{k-1})) \cdot L}{n_{k-1}(s, \pi_h(s))}} + \frac{13 H^2 L}{n_{k-1}(s, \pi_h(s))} \right. \right. \\ & \left. \left. + \left( 1 + \frac{2}{H} \right) \hat{p}_{k-1}(\cdot|s, \pi_h(s))^\top B_{h+1}^{\pi; s', a'; k}(\cdot) \right), H \right\}, & \forall s \in \mathcal{S}, \forall h \in [H], \\ B_{H+1}^{\pi; s', a'; k}(s) = & 0, & \forall s \in \mathcal{S}. \end{cases} \tag{32}$$

The construction of $B_h^{\pi; s', a'; k}(s)$ satisfies (see Lemma D.10 for more details)

$$|\hat{\phi}_{k-1}^\pi(s', a') - \phi^\pi(s', a')| \leq \mathbb{E}_{s_1 \sim \rho}\left[ B_1^{\pi; s', a'; k}(s_1) \right], \quad \forall (s', a') \in \mathcal{S} \times \mathcal{A},$$

$$\|\hat{\phi}_{k-1}^\pi - \phi^\pi\|_1 \leq \sum_{(s', a')} \mathbb{E}_{s_1 \sim \rho}\left[ B_1^{\pi; s', a'; k}(s_1) \right].$$

Incorporating this transition uncertainty $\mathbb{E}_{s_1 \sim \rho}[B_1^{\pi; s', a'; k}(s_1)]$ and reward uncertainty $\|\hat{\phi}_{k-1}^\pi\|_{(\Sigma_{k-1})^{-1}}$ into exploration bonuses, `LinUCB-Tran` computes the optimal policy $\pi^k$ under optimistic estimation (Line 5). After that, `LinUCB-Tran` plays episode $k$ with $\pi^k$, and collects trajectory $\tau^k$ and reward observation on each segment $\{R_i^k\}_{i=1}^m$ (Line 6).

### D.4. Proof for the Regret Upper Bound with Unknown Transition

In the following, we prove the regret upper bound (Theorem 4.3) of algorithm `LinUCB-Tran` for unknown transition.

Recall the definition of events $\mathcal{G}_{\mathrm{KL}}$ and $\mathcal{H}$ in Eqs. (19) and (20), respectively.

For any $k > 0$, define the set of state-action pairs

$$D_k := \left\{ (s, a) \in \mathcal{S} \times \mathcal{A} : \frac{1}{4} \sum_{k'=1}^k w_{k'}(s, a) \geq H \log\left( \frac{|\mathcal{S}||\mathcal{A}|H}{\delta'} \right) + H \right\}. \tag{33}$$

$D_k$ stands for the set of state-action pairs which have sufficient visitations in expectation.

**Lemma D.7.** *Assume that event $\mathcal{H}$ holds. Then, if $(s, a) \in D_k$,*

$$n_{k-1}(s, a) \geq \frac{1}{4} \sum_{k'=1}^k w_{k'}(s, a).$$

*Proof.* We have

$$
\begin{aligned}
n_{k-1}(s,a) &\geq \frac{1}{2}\sum_{k'=1}^{k-1} w_{k'}(s,a) - H\log\left(\frac{|\mathcal{S}||\mathcal{A}|H}{\delta'}\right) \\
&= \frac{1}{4}\sum_{k'=1}^{k-1} w_{k'}(s,a) + \frac{1}{4}\sum_{k'=1}^{k-1} w_{k'}(s,a) - H\log\left(\frac{|\mathcal{S}||\mathcal{A}|H}{\delta'}\right) \\
&= \frac{1}{4}\sum_{k'=1}^{k} w_{k'}(s,a) + \frac{1}{4}\sum_{k'=1}^{k} w_{k'}(s,a) - H\log\left(\frac{|\mathcal{S}||\mathcal{A}|H}{\delta'}\right) - \frac{1}{2}w_k(s,a) \\
&\overset{(a)}{\geq} \frac{1}{4}\sum_{k'=1}^{k} w_{k'}(s,a) + H - \frac{1}{2}w_k(s,a) \\
&\geq \frac{1}{4}\sum_{k'=1}^{k} w_{k'}(s,a),
\end{aligned}
$$

where (a) is due to the definition of $D_k$ (Eq. (33)). $\qquad\square$

**Lemma D.8.** *It holds that*

$$
\sum_{k=1}^{K}\sum_{h=1}^{H}\sum_{(s,a)\notin D_k} w_{k,h}(s,a) \leq 8|\mathcal{S}||\mathcal{A}|H\log\left(\frac{|\mathcal{S}||\mathcal{A}|H}{\delta'}\right).
$$

*Proof.* If $(s,a) \notin D_k$, then

$$
\frac{1}{4}\sum_{k'=1}^{k} w_{k'}(s,a) < H\log\left(\frac{|\mathcal{S}||\mathcal{A}|H}{\delta'}\right) + H.
$$

Thus, we have

$$
\begin{aligned}
\sum_{k=1}^{K}\sum_{h=1}^{H}\sum_{(s,a)\notin D_k} w_{k,h}(s,a) &= \sum_{(s,a)}\sum_{k=1}^{K}\sum_{h=1}^{H} \mathbb{1}\{(s,a)\notin D_k\}\cdot w_{k,h}(s,a) \\
&= \sum_{(s,a)}\sum_{k=1}^{K} \mathbb{1}\{(s,a)\notin D_k\}\cdot w_k(s,a) \\
&\leq 4|\mathcal{S}||\mathcal{A}|H\log\left(\frac{|\mathcal{S}||\mathcal{A}|H}{\delta'}\right) + 4|\mathcal{S}||\mathcal{A}|H \\
&\leq 8|\mathcal{S}||\mathcal{A}|H\log\left(\frac{|\mathcal{S}||\mathcal{A}|H}{\delta'}\right).
\end{aligned}
$$

$\qquad\square$

**Lemma D.9.** *Assume that event $\mathcal{H}$ holds. Then, we have*

$$
\sum_{k=1}^{K}\sum_{h=1}^{H}\sum_{(s,a)\in D_k} \frac{w_{k,h}(s,a)}{n_{k-1}(s,a)} \leq 4|\mathcal{S}||\mathcal{A}|\log(2KH).
$$

*Proof.* It holds that

$$
\sum_{k=1}^{K}\sum_{h=1}^{H}\sum_{(s,a)\in D_k} \frac{w_{k,h}(s,a)}{n_{k-1}(s,a)} = \sum_{k=1}^{K}\sum_{(s,a)\in D_k} \frac{w_k(s,a)}{n_{k-1}(s,a)}
$$

$$= \sum_{k=1}^{K} \sum_{(s,a)} \frac{w_k(s,a)}{n_{k-1}(s,a)} \cdot \mathbb{1}\{(s,a) \in D_k\}$$

$$\overset{(a)}{\leq} 4 \sum_{k=1}^{K} \sum_{(s,a)} \frac{w_k(s,a)}{\sum_{k'=1}^{k} w_k(s,a)} \cdot \mathbb{1}\{(s,a) \in D_k\}$$

$$= 4 \sum_{(s,a)} \sum_{k=1}^{K} \frac{w_k(s,a)}{\sum_{k'=1}^{k} w_k(s,a)} \cdot \mathbb{1}\{(s,a) \in D_k\}$$

$$\overset{(b)}{\leq} 4|\mathcal{S}||\mathcal{A}| \log(2KH),$$

where (a) uses Lemma D.7, and (b) follows from the analysis of Lemma 13 in (Zanette & Brunskill, 2019). $\qquad \square$

**Lemma D.10** (Error in Visitation Vectors). *Assume that event $\mathcal{G}_{\mathrm{KL}}$ holds. Then, for any $k > 0$ and policy $\pi$,*

$$\|\hat{\phi}_{k-1}(\pi) - \phi(\pi)\|_1 \leq \sum_{s',a'} \mathbb{E}_{s_1 \sim \rho}\left[ B_1^{\pi;s',a';k}(s_1) \right].$$

*Proof.* Since $\phi^\pi(s',a') = \mathbb{E}_{s_1 \sim \rho}[G_1^{\pi;s',a'}(s_1|p)]$ and $\hat{\phi}_{k-1}^\pi(s',a') = \mathbb{E}_{s_1 \sim \rho}[G_1^{\pi;s',a'}(s_1|\hat{p}_{k-1})]$, in this proof, we investigate the error in $G_h^{\pi;s',a'}$ due to the estimation of the transition model.

In the following, we prove by induction that for any $h \in [H]$ and $s \in \mathcal{S}$, $|G_h^{\pi;s',a'}(s|\hat{p}_{k-1}) - G_h^{\pi;s',a'}(s|p)| \leq B_h^{\pi;s',a';k}(s)$.

When $h = H + 1$, by definition, we have $G_{H+1}^{\pi;s',a'}(s|\hat{p}_{k-1}) = G_{H+1}^{\pi;s',a'}(s|p) = B_{H+1}^{\pi;s',a';k}(s) = 0$ for any $s \in \mathcal{S}$, and then the above statement trivially holds.

When $1 \leq h \leq H$, if $|G_{h+1}^{\pi;s',a'}(\cdot|\hat{p}_{k-1}) - G_{h+1}^{\pi;s',a'}(\cdot|p)| \leq B_{h+1}^{\pi;s',a';k}(\cdot)$ element-wise, then for any $s \in \mathcal{S}$, we have

$$|G_h^{\pi;s',a'}(s|\hat{p}_{k-1}) - G_h^{\pi;s',a'}(s|p)|$$

$$= \left| \hat{p}_{k-1}(\cdot|s,\pi_h(s))^\top G_{h+1}^{\pi;s',a'}(\cdot|\hat{p}_{k-1}) - p(\cdot|s,\pi_h(s))^\top G_{h+1}^{\pi;s',a'}(\cdot|p) \right|$$

$$= \hat{p}_{k-1}(\cdot|s,\pi_h(s))^\top \left| G_{h+1}^{\pi;s',a'}(\cdot|\hat{p}_{k-1}) - G_{h+1}^{\pi;s',a'}(\cdot|p) \right|$$

$$\quad + \left| (\hat{p}_{k-1}(\cdot|s,\pi_h(s)) - p(\cdot|s,\pi_h(s)))^\top G_{h+1}^{\pi;s',a'}(\cdot|p) \right|$$

$$\overset{(a)}{\leq} \hat{p}_{k-1}(\cdot|s,\pi_h(s))^\top \left| G_{h+1}^{\pi;s',a'}(\cdot|\hat{p}_{k-1}) - G_{h+1}^{\pi;s',a'}(\cdot|p) \right| + 2\sqrt{\frac{\mathrm{Var}_{p(\cdot|s,\pi_h(s))}(G_{h+1}^{\pi;s',a'}(\cdot|p)) \cdot L}{n_{k-1}(s,\pi_h(s))}}$$

$$\quad + \frac{HL}{n_{k-1}(s,\pi_h(s))}, \tag{34}$$

where (a) is due to Lemma E.4.

Here, we have

$$\mathrm{Var}_{p(\cdot|s,\pi_h(s))}(G_{h+1}^{\pi;s',a'}(\cdot|p))$$

$$\overset{(a)}{\leq} 2\mathrm{Var}_{\hat{p}_{k-1}(\cdot|s,\pi_h(s))}(G_{h+1}^{\pi;s',a'}(\cdot|p)) + \frac{4H^2 L}{n_{k-1}(s,\pi_h(s))}$$

$$\overset{(b)}{\leq} 4\mathrm{Var}_{\hat{p}_{k-1}(\cdot|s,\pi_h(s))}(G_{h+1}^{\pi;s',a'}(\cdot|\hat{p}_{k-1})) + 4H\hat{p}_{k-1}(\cdot|s,\pi_h(s))^\top |G_{h+1}^{\pi;s',a'}(\cdot|\hat{p}_{k-1}) - G_{h+1}^{\pi;s',a'}(\cdot|p)|$$

$$\quad + \frac{4H^2 L}{n_{k-1}(s,\pi_h(s))},$$

where (a) uses Lemma E.5 and (b) comes from Lemma E.6.

Then,

$$
\sqrt{\frac{\mathrm{Var}_{p(\cdot|s,\pi_h(s))}(G_{h+1}^{\pi;s',a'}(\cdot|p)) \cdot L}{n_{k-1}(s,\pi_h(s))}}
\tag{35}
$$

$$
\leq \sqrt{\frac{4\mathrm{Var}_{\hat{p}_{k-1}(\cdot|s,\pi_h(s))}(G_{h+1}^{\pi;s',a'}(\cdot|\hat{p}_{k-1})) \cdot L}{n_{k-1}(s,\pi_h(s))}}
$$

$$
+ \sqrt{\frac{1}{H}\hat{p}_{k-1}(\cdot|s,\pi_h(s))^\top |G_{h+1}^{\pi;s',a'}(\cdot|\hat{p}_{k-1}) - G_{h+1}^{\pi;s',a'}(\cdot|p)| \cdot \frac{4H^2L}{n_{k-1}(s,\pi_h(s))}} + \frac{2HL}{n_{k-1}(s,\pi_h(s))}
$$

$$
\overset{(a)}{\leq} 2\sqrt{\frac{\mathrm{Var}_{\hat{p}_{k-1}(\cdot|s,\pi_h(s))}(G_{h+1}^{\pi;s',a'}(\cdot|\hat{p}_{k-1})) \cdot L}{n_{k-1}(s,\pi_h(s))}}
$$

$$
+ \frac{1}{H}\hat{p}_{k-1}(\cdot|s,\pi_h(s))^\top |G_{h+1}^{\pi;s',a'}(\cdot|\hat{p}_{k-1}) - G_{h+1}^{\pi;s',a'}(\cdot|p)| + \frac{6H^2L}{n_{k-1}(s,\pi_h(s))},
\tag{36}
$$

where (a) is due to the fact that $\sqrt{xy} \leq x + y$.

Hence, plugging Eq. (36) into Eq. (34) and using the fact that $|G_h^{\pi;s',a'}(s)| \in [0, H]$, we have

$$
|G_h^{\pi;s',a'}(s|\hat{p}_{k-1}) - G_h^{\pi;s',a'}(s|p)|
$$

$$
\leq \left( 4\sqrt{\frac{\mathrm{Var}_{\hat{p}_{k-1}(\cdot|s,\pi_h(s))}(G_{h+1}^{\pi;s',a'}(\cdot|\hat{p}_{k-1})) \cdot L}{n_{k-1}(s,\pi_h(s))}} + \frac{13H^2L}{n_{k-1}(s,\pi_h(s))} \right.
$$

$$
\left. + \left(1 + \frac{2}{H}\right)\hat{p}_{k-1}(\cdot|s,\pi_h(s))^\top \left|G_{h+1}^{\pi;s',a'}(\cdot|\hat{p}_{k-1}) - G_{h+1}^{\pi;s',a'}(\cdot|p)\right| \right) \wedge H.
$$

$$
\leq \left( 4\sqrt{\frac{\mathrm{Var}_{\hat{p}_{k-1}(\cdot|s,\pi_h(s))}(G_{h+1}^{\pi;s',a'}(\cdot|\hat{p}_{k-1})) \cdot L}{n_{k-1}(s,\pi_h(s))}} + \frac{13H^2L}{n_{k-1}(s,\pi_h(s))} \right.
$$

$$
\left. + \left(1 + \frac{2}{H}\right)\hat{p}_{k-1}(\cdot|s,\pi_h(s))^\top B_{h+1}^{\pi;s',a';k}(\cdot) \right) \wedge H
$$

$$
= B_h^{\pi;s',a';k}(s),
$$

which completes the induction proof.

Therefore,

$$
\left|\hat{\phi}_{k-1}^{\pi}(s',a') - \phi^{\pi}(s',a')\right| = \left|\mathbb{E}_{s_1 \sim \rho}\left[G_1^{\pi;s',a'}(s_1|\hat{p}_{k-1})\right] - \mathbb{E}_{s_1 \sim \rho}\left[G_1^{\pi;s',a'}(s_1|p)\right]\right|
$$

$$
\leq \mathbb{E}_{s_1 \sim \rho}\left[\left|G_1^{\pi;s',a'}(s_1|\hat{p}_{k-1}) - G_1^{\pi;s',a'}(s_1|p)\right|\right]
$$

$$
\leq \mathbb{E}_{s_1 \sim \rho}\left[B_1^{\pi;s',a';k}(s_1)\right].
$$

Summing over $(s',a') \in \mathcal{S} \times \mathcal{A}$, we obtain this lemma. $\qquad\square$

**Lemma D.11.** *Assume that event $\mathcal{G}_{\mathrm{KL}} \cap \mathcal{H}$ holds. Then, for any $k > 0$ and policy $\pi$,*

$$
\mathbb{E}_{s_1 \sim \rho}\left[B_1^{\pi;s',a';k}(s_1)\right]
$$

$$
\leq e^{12}\sum_{h=1}^{H}\sum_{s,a} w_h^{\pi}(s,a)\left(8\sqrt{\frac{\mathrm{Var}_{p(\cdot|s,a)}(G_{h+1}^{\pi;s',a'}(\cdot|p)) \cdot L}{n_{k-1}(s,a)}} + \frac{46H^2L}{n_{k-1}(s,a)}\right) \wedge H,
$$

*and*

$$
\sum_{s',a'}\sum_{k=1}^{K}\mathbb{E}_{s_1 \sim \rho}\left[B_1^{\pi^k;s',a';k}(s_1)\right]
$$

$$\leq 16e^{12}|\mathcal{S}|^{\frac{3}{2}}|\mathcal{A}|^{\frac{3}{2}}H\sqrt{KL\log(2KH)} + 192e^{12}|\mathcal{S}|^2|\mathcal{A}|^2H^2L\log(2KH).$$

*Proof.* First, we prove the first statement.

For any policy $\pi$, $k > 0$, $(s', a') \in \mathcal{S} \times \mathcal{A}$, $h \in [H]$ and $s \in \mathcal{S}$, we have

$$
\begin{aligned}
B_h^{\pi;s',a';k}(s) &\leq 4\sqrt{\frac{\mathrm{Var}_{\hat{p}_{k-1}(\cdot|s,\pi_h(s))}(G_{h+1}^{\pi;s',a'}(\cdot|\hat{p}_{k-1})) \cdot L}{n_{k-1}(s,\pi_h(s))}} + \frac{13H^2L}{n_{k-1}(s,\pi_h(s))} \\
&\quad + \left(1 + \frac{2}{H}\right)\hat{p}_{k-1}(\cdot|s,\pi_h(s))^\top B_{h+1}^{\pi;s',a';k}(\cdot) \\
&= 4\sqrt{\frac{\mathrm{Var}_{\hat{p}_{k-1}(\cdot|s,\pi_h(s))}(G_{h+1}^{\pi;s',a'}(\cdot|\hat{p}_{k-1})) \cdot L}{n_{k-1}(s,\pi_h(s))}} + \frac{13H^2L}{n_{k-1}(s,\pi_h(s))} \\
&\quad + \left(1 + \frac{2}{H}\right)p(\cdot|s,\pi_h(s))^\top B_{h+1}^{\pi;s',a';k}(\cdot) \\
&\quad + \left(1 + \frac{2}{H}\right)(\hat{p}_{k-1}(\cdot|s,\pi_h(s)) - p(\cdot|s,\pi_h(s)))^\top B_{h+1}^{\pi;s',a';k}(\cdot) \\
&\overset{(a)}{\leq} 4\sqrt{\frac{\mathrm{Var}_{\hat{p}_{k-1}(\cdot|s,\pi_h(s))}(G_{h+1}^{\pi;s',a'}(\cdot|\hat{p}_{k-1})) \cdot L}{n_{k-1}(s,\pi_h(s))}} + \frac{13H^2L}{n_{k-1}(s,\pi_h(s))} \\
&\quad + \left(1 + \frac{2}{H}\right)p(\cdot|s,\pi_h(s))^\top B_{h+1}^{\pi;s',a';k}(\cdot) \\
&\quad + \left(1 + \frac{2}{H}\right) \cdot \left(2\sqrt{\frac{\mathrm{Var}_{p(\cdot|s,\pi_h(s))}(B_{h+1}^{\pi;s',a';k}(\cdot)) \cdot L}{n_{k-1}(s,\pi_h(s))}} + \frac{HL}{n_{k-1}(s,\pi_h(s))}\right) \\
&\leq 4\sqrt{\frac{\mathrm{Var}_{\hat{p}_{k-1}(\cdot|s,\pi_h(s))}(G_{h+1}^{\pi;s',a'}(\cdot|\hat{p}_{k-1})) \cdot L}{n_{k-1}(s,\pi_h(s))}} + \frac{13H^2L}{n_{k-1}(s,\pi_h(s))} \\
&\quad + \left(1 + \frac{2}{H}\right)p(\cdot|s,\pi_h(s))^\top B_{h+1}^{\pi;s',a';k}(\cdot) \\
&\quad + \left(1 + \frac{2}{H}\right)\left(2\sqrt{\frac{1}{H}p(\cdot|s,\pi_h(s))^\top B_{h+1}^{\pi;s',a';k}(\cdot)\frac{H^2L}{n_{k-1}(s,\pi_h(s))}} + \frac{HL}{n_{k-1}(s,\pi_h(s))}\right) \\
&\overset{(b)}{\leq} 4\sqrt{\frac{\mathrm{Var}_{\hat{p}_{k-1}(\cdot|s,\pi_h(s))}(G_{h+1}^{\pi;s',a'}(\cdot|\hat{p}_{k-1})) \cdot L}{n_{k-1}(s,\pi_h(s))}} + \frac{22H^2L}{n_{k-1}(s,\pi_h(s))} \\
&\quad + \left(1 + \frac{8}{H}\right)p(\cdot|s,\pi_h(s))^\top B_{h+1}^{\pi;s',a';k}(\cdot),
\end{aligned}
\tag{37}
$$

where (a) uses Lemma E.4, and (b) follows from the fact that $\sqrt{xy} \leq x + y$.

In addition, we have

$$
\begin{aligned}
&\mathrm{Var}_{\hat{p}_{k-1}(\cdot|s,\pi_h(s))}(G_{h+1}^{\pi;s',a'}(\cdot|\hat{p}_{k-1})) \\
&\overset{(a)}{\leq} 2\mathrm{Var}_{p(\cdot|s,\pi_h(s))}(G_{h+1}^{\pi;s',a'}(\cdot|\hat{p}_{k-1})) + \frac{4H^2L}{n_{k-1}(s,a)} \\
&\overset{(b)}{\leq} 4\mathrm{Var}_{p(\cdot|s,\pi_h(s))}(G_{h+1}^{\pi;s',a'}(\cdot|p)) + 4Hp(\cdot|s,\pi_h(s))^\top \left|G_{h+1}^{\pi;s',a'}(\cdot|\hat{p}_{k-1}) - G_{h+1}^{\pi;s',a'}(\cdot|p)\right| \\
&\quad + \frac{4H^2L}{n_{k-1}(s,a)} \\
&\leq 4\mathrm{Var}_{p(\cdot|s,\pi_h(s))}(G_{h+1}^{\pi;s',a'}(\cdot|p)) + 4Hp(\cdot|s,\pi_h(s))^\top B_{h+1}^{\pi;s',a'}(\cdot|\hat{p}_{k-1}) + \frac{4H^2L}{n_{k-1}(s,a)},
\end{aligned}
$$

where (a) uses Lemma E.5, and (b) comes from Lemma E.6.

Then,

$$
\sqrt{\frac{\mathrm{Var}_{\hat{p}_{k-1}(\cdot|s,\pi_h(s))}(G_{h+1}^{\pi;s',a'}(\cdot|\hat{p}_{k-1}))\cdot L}{n_{k-1}(s,\pi_h(s))}}
$$

$$
\leq \sqrt{\frac{4\mathrm{Var}_{p(\cdot|s,\pi_h(s))}(G_{h+1}^{\pi;s',a'}(\cdot|p))\cdot L}{n_{k-1}(s,\pi_h(s))}} + \sqrt{\frac{1}{H}p(\cdot|s,\pi_h(s))^\top B_{h+1}^{\pi;s',a'}(\cdot|\hat{p}_{k-1})\cdot\frac{4H^2L}{n_{k-1}(s,\pi_h(s))}}
$$

$$
+ \frac{2HL}{n_{k-1}(s,a)}
$$

$$
\leq 2\sqrt{\frac{\mathrm{Var}_{p(\cdot|s,\pi_h(s))}(G_{h+1}^{\pi;s',a'}(\cdot|p))\cdot L}{n_{k-1}(s,\pi_h(s))}} + \frac{1}{H}p(\cdot|s,\pi_h(s))^\top B_{h+1}^{\pi;s',a'}(\cdot|\hat{p}_{k-1})
$$

$$
+ \frac{6H^2L}{n_{k-1}(s,\pi_h(s))}
\tag{38}
$$

Plugging Eq. (38) into Eq. (37) and using the clipping definition of $B_h^{\pi;s',a';k}(s)$, we have

$$
B_h^{\pi;s',a';k}(s) \leq \left(8\sqrt{\frac{\mathrm{Var}_{p(\cdot|s,\pi_h(s))}(G_{h+1}^{\pi;s',a'}(\cdot|p))\cdot L}{n_{k-1}(s,\pi_h(s))}} + \frac{46H^2L}{n_{k-1}(s,\pi_h(s))}\right) \wedge H
$$

$$
+ \left(1 + \frac{12}{H}\right)p(\cdot|s,\pi_h(s))^\top B_{h+1}^{\pi;s',a';k}(\cdot)
$$

Using the above inequality, taking $s_1 \sim \rho$, and unfolding $B_1^{\pi;s',a';k}(s_1)$ over $h$, we have

$$
\mathbb{E}_{s_1\sim\rho}\left[B_1^{\pi;s',a';k}(s_1)\right]
$$

$$
\leq e^{12}\sum_{h=1}^{H}\sum_{s,a}w_h^\pi(s,a)\left(8\sqrt{\frac{\mathrm{Var}_{p(\cdot|s,a)}(G_{h+1}^{\pi;s',a'}(\cdot|p))\cdot L}{n_{k-1}(s,a)}} + \frac{46H^2L}{n_{k-1}(s,a)}\right) \wedge H.
\tag{39}
$$

Next, we prove the second statement.

It holds that

$$
\sum_{s',a'}\sum_{k=1}^{K}\mathbb{E}_{s_1\sim\rho}\left[B_1^{\pi^k;s',a';k}(s_1)\right]
$$

$$
\leq e^{12}\sum_{s',a'}\sum_{k=1}^{K}\sum_{h=1}^{H}\sum_{(s,a)\in D_k}w_{k,h}(s,a)\left(8\sqrt{\frac{\mathrm{Var}_{p(\cdot|s,a)}(G_{h+1}^{\pi^k;s',a'}(\cdot|p))\cdot L}{n_{k-1}(s,a)}} + \frac{46H^2L}{n_{k-1}(s,a)}\right)
$$

$$
+ e^{12}H|\mathcal{S}||\mathcal{A}|\sum_{k=1}^{K}\sum_{h=1}^{H}\sum_{(s,a)\notin D_k}w_{k,h}(s,a)
$$

$$
\stackrel{(a)}{\leq} 8e^{12}\sqrt{L}\sum_{s',a'}\sqrt{\sum_{k=1}^{K}\sum_{h=1}^{H}\sum_{(s,a)\in D_k}w_{k,h}(s,a)\mathrm{Var}_{p(\cdot|s,a)}(G_{h+1}^{\pi^k;s',a'}(\cdot|p))}\cdot
$$

$$
\sqrt{\sum_{k=1}^{K}\sum_{h=1}^{H}\sum_{(s,a)\in D_k}\frac{w_{k,h}(s,a)}{n_{k-1}(s,a)}} + e^{12}|\mathcal{S}||\mathcal{A}|\cdot 46H^2L\sum_{k=1}^{K}\sum_{h=1}^{H}\sum_{(s,a)\in D_k}\frac{w_{k,h}(s,a)}{n_{k-1}(s,a)}
$$

$$
+ 8e^{12}|\mathcal{S}|^2|\mathcal{A}|^2H^2\log\left(\frac{|\mathcal{S}||\mathcal{A}|H}{\delta'}\right)
$$

$$\overset{(b)}{\leq} 8e^{12}|\mathcal{S}||\mathcal{A}|\sqrt{L}\sqrt{KH^2} \cdot \sqrt{4|\mathcal{S}||\mathcal{A}|\log(2KH)} + 184e^{12}|\mathcal{S}|^2|\mathcal{A}|^2 H^2 L \log(2KH)$$
$$+ 8e^{12}|\mathcal{S}|^2|\mathcal{A}|^2 H^2 \log\left(\frac{|\mathcal{S}||\mathcal{A}|H}{\delta'}\right)$$
$$\leq 16e^{12}|\mathcal{S}|^{\frac{3}{2}}|\mathcal{A}|^{\frac{3}{2}} H \sqrt{KL\log(2KH)} + 192e^{12}|\mathcal{S}|^2|\mathcal{A}|^2 H^2 L \log(2KH),$$

where (a) is due to Lemma D.8, and (b) follows from Lemmas E.3 and D.9. $\qquad\square$

**Lemma D.12** (Optimism under Sum Feedback and Unknown Transition). *Assume that event $\mathcal{G}_{\mathrm{KL}}$ holds. Then, for any $k > 0$ and fixed policy $\pi$,*

$$V_1^\pi(s_1) \leq \hat{\phi}_{k-1}(\pi)^\top \hat{\theta}_{k-1} + \beta(k-1) \cdot \|\hat{\phi}_{k-1}(\pi)\|_{(\Sigma_{k-1})^{-1}} + r_{\max} \sum_{s',a'} \mathbb{E}_{s_1 \sim \rho}\left[B_1^{\pi;s',a';k}(s_1)\right].$$

*Proof.* It holds that

$$V_1^\pi(s_1) = \phi(\pi)^\top \theta$$
$$= \hat{\phi}_{k-1}(\pi)^\top \hat{\theta}_{k-1} + \phi(\pi)^\top \theta - \hat{\phi}_{k-1}(\pi)^\top \theta + \hat{\phi}_{k-1}(\pi)^\top \theta - \hat{\phi}_{k-1}(\pi)^\top \hat{\theta}_{k-1}$$
$$\leq \hat{\phi}_{k-1}(\pi)^\top \hat{\theta}_{k-1} + \|\phi(\pi) - \hat{\phi}_{k-1}(\pi)\|_1 \cdot \|\theta\|_\infty + \beta(k-1) \cdot \|\hat{\phi}_{k-1}(\pi)\|_{(\Sigma_{k-1})^{-1}}$$
$$\overset{(a)}{\leq} \hat{\phi}_{k-1}(\pi)^\top \hat{\theta}_{k-1} + \beta(k-1) \cdot \|\hat{\phi}_{k-1}(\pi)\|_{(\Sigma_{k-1})^{-1}} + r_{\max} \sum_{s',a'} \mathbb{E}_{s_1 \sim \rho}\left[B_1^{\pi;s',a';k}(s_1)\right],$$

where (a) uses Lemma D.10. $\qquad\square$

**Lemma D.13.** *For any $K \geq 1$, we have*

$$\sum_{k=1}^{K} \sum_{i=1}^{m} \left\|\phi^{\tau_i^k}\right\|_{(\Sigma_{k-1})^{-1}} \leq H\sqrt{\frac{2K|\mathcal{S}||\mathcal{A}|}{\lambda} \log\left(1 + \frac{KH^2}{\lambda|\mathcal{S}||\mathcal{A}|m}\right)}.$$

*Proof.* We have

$$\sum_{k=1}^{K} \sum_{i=1}^{m} \left\|\phi^{\tau_i^k}\right\|_{(\Sigma_{k-1})^{-1}} \leq \sqrt{Km \sum_{k=1}^{K} \sum_{i=1}^{m} \left\|\phi^{\tau_i^k}\right\|_{(\Sigma_{k-1})^{-1}}^2}$$
$$= \sqrt{Km \sum_{k=1}^{K} \min\left\{\sum_{i=1}^{m} \left\|\phi^{\tau_i^k}\right\|_{(\Sigma_{k-1})^{-1}}^2, \frac{H^2}{m\lambda}\right\}}$$
$$= \sqrt{\frac{H^2 K}{\lambda} \sum_{k=1}^{K} \min\left\{\frac{m\lambda}{H^2} \sum_{i=1}^{m} \left\|\phi^{\tau_i^k}\right\|_{(\Sigma_{k-1})^{-1}}^2, 1\right\}}$$
$$\overset{(a)}{\leq} \sqrt{\frac{2H^2 K}{\lambda} \sum_{k=1}^{K} \log\left(1 + \min\left\{\frac{m\lambda}{H^2} \sum_{i=1}^{m} \left\|\phi^{\tau_i^k}\right\|_{(\Sigma_{k-1})^{-1}}^2, 1\right\}\right)}$$
$$\overset{(b)}{\leq} \sqrt{\frac{2H^2 K}{\lambda} \sum_{k=1}^{K} \log\left(1 + \sum_{i=1}^{m} \left\|\phi^{\tau_i^k}\right\|_{(\Sigma_{k-1})^{-1}}^2\right)}$$
$$\overset{(c)}{\leq} \sqrt{\frac{2KH^2|\mathcal{S}||\mathcal{A}|}{\lambda} \log\left(1 + \frac{KH^2}{\lambda|\mathcal{S}||\mathcal{A}|m}\right)},$$

where inequality (a) uses the fact that $x \leq 2\log(1+x)$ for any $0 \leq x \leq 1$, inequality (b) is due to the fact that $\lambda \leq \frac{H^2}{m}$, and inequality (c) follows from Lemma D.3. $\qquad\square$

Define event

$$\mathcal{F}_{\text{reg}}^{\text{S}} := \left\{ \left| \sum_{k'=1}^{k} \left( \mathbb{E}_{\tau \sim \pi^{k'}} \left[ \|\phi^\tau\|_{(\Sigma_{k'-1})^{-1}} \,|F_{k'-1} \right] - \|\phi^\tau\|_{(\Sigma_{k'-1})^{-1}} \right) \right| \le 4H \sqrt{\frac{k}{\lambda} \log\left(\frac{4k}{\delta'}\right)}, \right.$$

$$\left. \forall k > 0 \right\}. \tag{40}$$

Event $\mathcal{F}_{\text{reg}}^{\text{S}}$ is similar to $\mathcal{F}_{\text{opt}}^{\text{S}}$, except that here the universal upper bound of $\|\phi^\tau\|_{(\Sigma_{k'-1})^{-1}}$ is $\frac{H}{\sqrt{\lambda}}$ rather than 1.

**Lemma D.14.** *It holds that*

$$\Pr\left[\mathcal{F}_{\text{reg}}^{\text{S}}\right] \ge 1 - \delta'.$$

*Proof.* For any $k' \ge 1$, we have that $\|\phi^\tau\|_{(\Sigma_{k'-1})^{-1}} \le \frac{H}{\sqrt{\lambda}}$, and then $|\mathbb{E}_{\tau \sim \pi^{k'}}[\|\phi^\tau\|_{(\Sigma_{k'-1})^{-1}}\,|F_{k'-1}] - \|\phi^\tau\|_{(\Sigma_{k'-1})^{-1}}| \le \frac{2H}{\sqrt{\lambda}}$.

Using the Azuma-Hoeffding inequality, we have that for any fixed $k > 0$, with probability at least $1 - \frac{\delta'}{2k^2}$,

$$\left| \sum_{k'=1}^{k} \left( \mathbb{E}_{\tau \sim \pi^{k'}} \left[ \|\phi^\tau\|_{(\Sigma_{k'-1})^{-1}} \,|F_{k'-1} \right] - \|\phi^\tau\|_{(\Sigma_{k'-1})^{-1}} \right) \right| \le \sqrt{2 \cdot \frac{4H^2}{\lambda} \cdot k \log\left(\frac{4k^2}{\delta'}\right)}.$$

Since $\sum_{k=1}^{\infty} \frac{\delta'}{2k^2} \le \delta'$, by a union bound over $k$, we have that with probability at least $\delta'$, for any $k \ge 1$,

$$\left| \sum_{k'=1}^{k} \left( \mathbb{E}_{\tau \sim \pi^{k'}} \left[ \|\phi^\tau\|_{(\Sigma_{k'-1})^{-1}} \,|F_{k'-1} \right] - \|\phi^\tau\|_{(\Sigma_{k'-1})^{-1}} \right) \right| \le \sqrt{2 \cdot \frac{4H^2}{\lambda} \cdot k \log\left(\frac{4k^2}{\delta'}\right)}$$

$$\le 4H \sqrt{\frac{k}{\lambda} \log\left(\frac{4k}{\delta'}\right)}.$$

$\square$

*Proof of Theorem 4.3.* Let $\delta' = \frac{\delta}{4}$. Then, we have $\Pr[\mathcal{K} \cap \mathcal{F}_{\text{reg}}^{\text{S}} \cap \mathcal{G}_{\text{KL}} \cap \mathcal{H}] \ge 1 - \delta$. Thus, it suffices to prove the regret upper bound when event $\mathcal{K} \cap \mathcal{F}_{\text{reg}}^{\text{S}} \cap \mathcal{G}_{\text{KL}} \cap \mathcal{H}$ holds.

Assume that event $\mathcal{K} \cap \mathcal{F}_{\text{reg}}^{\text{S}} \cap \mathcal{G}_{\text{KL}} \cap \mathcal{H}$ holds. For any $k > 0$, we have

$$\sum_{k=1}^{K} \left( V^*(s_1) - V^{\pi^k}(s_1) \right)$$

$$\overset{(a)}{\le} \sum_{k=1}^{K} \left( \hat{\phi}_{k-1}(\pi^*)^\top \hat{\theta}_{k-1} + \beta(k-1) \cdot \|\hat{\phi}_{k-1}(\pi^*)\|_{(\Sigma_{k-1})^{-1}} + r_{\max} \sum_{s',a'} \mathbb{E}_{s_1 \sim \rho} \left[ B_1^{\pi^*;s',a';k}(s_1) \right] \right.$$

$$\left. - V^{\pi^k} \right)$$

$$\overset{(b)}{\le} \sum_{k=1}^{K} \left( \hat{\phi}_{k-1}(\pi^k)^\top \hat{\theta}_{k-1} + \beta(k-1) \cdot \|\hat{\phi}_{k-1}(\pi^k)\|_{(\Sigma_{k-1})^{-1}} + r_{\max} \sum_{s',a'} \mathbb{E}_{s_1 \sim \rho} \left[ B_1^{\pi^k;s',a';k}(s_1) \right] \right.$$

$$\left. - V^{\pi^k} \right)$$

$$\le \sum_{k=1}^{K} \left( \hat{\phi}_{k-1}(\pi^k)^\top \hat{\theta}_{k-1} - \hat{\phi}_{k-1}(\pi^k)^\top \theta + \hat{\phi}_{k-1}(\pi^k)^\top \theta - (\phi^{\pi^k})^\top \theta \right.$$

$$+ \beta(k-1) \cdot \|\hat{\phi}_{k-1}(\pi^k)\|_{(\Sigma_{k-1})^{-1}} + r_{\max} \sum_{s',a'} \mathbb{E}_{s_1 \sim \rho} \left[ B_1^{\pi^k; s', a'; k}(s_1) \right] \Big)$$

$$\overset{(c)}{\leq} \sum_{k=1}^{K} \left( 2\beta(k-1) \cdot \|\hat{\phi}_{k-1}(\pi^k)\|_{(\Sigma_{k-1})^{-1}} + 2r_{\max} \sum_{s',a'} \mathbb{E}_{s_1 \sim \rho} \left[ B_1^{\pi^k; s', a'; k}(s_1) \right] \right)$$

$$\leq 2\beta(K) \sum_{k=1}^{K} \|\hat{\phi}_{k-1}(\pi^k)\|_{(\Sigma_{k-1})^{-1}} + 2r_{\max} \sum_{k=1}^{K} \sum_{s',a'} \mathbb{E}_{s_1 \sim \rho} \left[ B_1^{\pi^k; s', a'; k}(s_1) \right], \tag{41}$$

where (a) uses Lemma D.12, (b) is due to the definition of $\pi^k$, and (c) follows from Lemma D.10 and the definition of event $\mathcal{K}$.

Next, we first bound $\sum_{k=1}^{K} \|\hat{\phi}_{k-1}(\pi^k)\|_{(\Sigma_{k-1})^{-1}}$.

We have

$$\sum_{k=1}^{K} \|\hat{\phi}_{k-1}(\pi^k)\|_{(\Sigma_{k-1})^{-1}} \leq \sum_{k=1}^{K} \left( \|\phi^{\pi^k}\|_{(\Sigma_{k-1})^{-1}} + \|\hat{\phi}_{k-1}(\pi^k) - \phi^{\pi^k}\|_{(\Sigma_{k-1})^{-1}} \right)$$

$$\leq \sum_{k=1}^{K} \left( \|\phi^{\pi^k}\|_{(\Sigma_{k-1})^{-1}} + \frac{1}{\sqrt{\lambda}} \cdot \|\hat{\phi}_{k-1}(\pi^k) - \phi^{\pi^k}\|_2 \right)$$

$$\leq \sum_{k=1}^{K} \left( \|\phi^{\pi^k}\|_{(\Sigma_{k-1})^{-1}} + \frac{1}{\sqrt{\lambda}} \cdot \|\hat{\phi}_{k-1}(\pi^k) - \phi^{\pi^k}\|_1 \right). \tag{42}$$

Here we have

$$\sum_{k=1}^{K} \|\phi^{\pi^k}\|_{(\Sigma_{k-1})^{-1}}$$

$$= \sum_{k=1}^{K} \|\mathbb{E}_{\tau \sim \pi^k} [\phi^\tau | F_{k-1}]\|_{(\Sigma_{k-1})^{-1}}$$

$$\overset{(a)}{\leq} \sum_{k=1}^{K} \mathbb{E}_{\tau \sim \pi^k} \left[ \|\phi^\tau\|_{(\Sigma_{k-1})^{-1}} | F_{k-1} \right]$$

$$= \sum_{k=1}^{K} \left( \mathbb{E}_{\tau \sim \pi^k} \left[ \|\phi^\tau\|_{(\Sigma_{k-1})^{-1}} | F_{k-1} \right] - \|\phi(\tau^k)\|_{(\Sigma_{k-1})^{-1}} + \|\phi(\tau^k)\|_{(\Sigma_{k-1})^{-1}} \right)$$

$$\leq \sum_{k=1}^{K} \left( \mathbb{E}_{\tau \sim \pi^k} \left[ \|\phi^\tau\|_{(\Sigma_{k-1})^{-1}} | F_{k-1} \right] - \|\phi(\tau^k)\|_{(\Sigma_{k-1})^{-1}} + \sum_{i=1}^{m} \|\phi^{\tau_i^k}\|_{(\Sigma_{k-1})^{-1}} \right)$$

$$\overset{(b)}{\leq} 4H \sqrt{\frac{K}{\lambda} \log\left(\frac{4K}{\delta'}\right)} + H \sqrt{\frac{2K|\mathcal{S}||\mathcal{A}|}{\lambda} \log\left(1 + \frac{KH^2}{\lambda |\mathcal{S}||\mathcal{A}|m}\right)}, \tag{43}$$

where (a) uses the Jensen inequality, and (b) comes from the definition of $\mathcal{F}_{\text{reg}}^S$ and Lemma D.13.

Hence, plugging Eq. (43) into Eq. (42) and using Lemma D.10, we have

$$\sum_{k=1}^{K} \|\hat{\phi}_{k-1}(\pi^k)\|_{(\Sigma_{k-1})^{-1}} \leq 4H \sqrt{\frac{K}{\lambda} \log\left(\frac{4K}{\delta'}\right)} + H \sqrt{\frac{2K|\mathcal{S}||\mathcal{A}|}{\lambda} \log\left(1 + \frac{KH^2}{\lambda |\mathcal{S}||\mathcal{A}|}\right)}$$

$$+ \frac{1}{\sqrt{\lambda}} \sum_{k=1}^{K} \sum_{s',a'} \mathbb{E}_{s_1 \sim \rho} \left[ B_1^{\pi; s', a'; k}(s_1) \right]. \tag{44}$$

On the other hand, according to Eq. (39), we have

$$\mathbb{E}_{s_1 \sim \rho}\left[B_1^{\pi;s',a';k}(s_1)\right] \le e^{12} \sum_{h=1}^{H} \sum_{s,a} w_h^{\pi}(s,a) \left(8\sqrt{\frac{\mathrm{Var}_{p(\cdot|s,a)}(G_{h+1}^{\pi;s',a'}(\cdot|p)) \cdot L}{n_{k-1}(s,a)}} + \frac{46 H^2 L}{n_{k-1}(s,a)}\right) \wedge H.$$

Therefore, plugging Eqs. (44) and (39) into Eq. (41), we have

$$\sum_{k=1}^{K}\left(V^* - V^{\pi^k}\right)$$

$$\le 2\beta(K)\left(4H\sqrt{\frac{K}{\lambda}\log\left(\frac{4K}{\delta'}\right)} + H\sqrt{\frac{2K|\mathcal{S}||\mathcal{A}|}{\lambda}\log\left(1 + \frac{KH^2}{\lambda|\mathcal{S}||\mathcal{A}|m}\right)}\right)$$

$$+ 2\left(\frac{\beta(K)}{\sqrt{\lambda}} + r_{\max}\right)\sum_{s',a'}\sum_{k=1}^{K}\mathbb{E}_{s_1 \sim \rho}\left[B_1^{\pi^k;s',a';k}(s_1)\right]$$

$$\overset{(a)}{\le} 2\beta(K)\left(4H\sqrt{\frac{K}{\lambda}\log\left(\frac{4K}{\delta'}\right)} + H\sqrt{\frac{2K|\mathcal{S}||\mathcal{A}|}{\lambda}\log\left(1 + \frac{KH^2}{\lambda|\mathcal{S}||\mathcal{A}|m}\right)}\right)$$

$$+ \frac{4\beta(K)}{\sqrt{\lambda}}\left(16e^{12}|\mathcal{S}|^{\frac{3}{2}}|\mathcal{A}|^{\frac{3}{2}}H\sqrt{KL\log(2KH)} + 192e^{12}|\mathcal{S}|^2|\mathcal{A}|^2H^2L\log(2KH)\right)$$

$$= O\left(\left(\sqrt{\frac{H|\mathcal{S}||\mathcal{A}|}{m}\log\left(\left(1 + \frac{KH^2}{\lambda|\mathcal{S}||\mathcal{A}|m}\right)\frac{1}{\delta}\right)} + r_{\max}\sqrt{\lambda|\mathcal{S}||\mathcal{A}|}\right)\cdot\right.$$

$$\left.\left(H\sqrt{\frac{K|\mathcal{S}||\mathcal{A}|}{\lambda}\log\left(\left(1 + \frac{KH^2}{\lambda|\mathcal{S}||\mathcal{A}|m}\right)\frac{1}{\delta}\right)} + |\mathcal{S}|^{\frac{3}{2}}|\mathcal{A}|^{\frac{3}{2}}H\sqrt{\frac{KL}{\lambda}\log(KH)} + \frac{|\mathcal{S}|^2|\mathcal{A}|^2H^2L}{\sqrt{\lambda}}\log(KH)\right)\right)$$

$$\overset{(b)}{=} O\left((1 + r_{\max})|\mathcal{S}|^2|\mathcal{A}|^2H\sqrt{K}\left(\log\left(\left(1 + \frac{KH}{|\mathcal{S}||\mathcal{A}|}\right)\frac{1}{\delta}\right) + \sqrt{L\log(KH)}\sqrt{\log\left(\left(1 + \frac{KH}{|\mathcal{S}||\mathcal{A}|}\right)\frac{1}{\delta}\right)}\right)\right.$$

$$\left. + (1 + r_{\max})|\mathcal{S}|^{\frac{5}{2}}|\mathcal{A}|^{\frac{5}{2}}H^2L\log(KH)\sqrt{\log\left(\left(1 + \frac{KH}{|\mathcal{S}||\mathcal{A}|}\right)\frac{1}{\delta}\right)}\right)$$

$$= \tilde{O}\left((1 + r_{\max})|\mathcal{S}|^{\frac{5}{2}}|\mathcal{A}|^2H\sqrt{K} + (1 + r_{\max})|\mathcal{S}|^{\frac{7}{2}}|\mathcal{A}|^{\frac{5}{2}}H^2\right),$$

where inequality (a) comes from Lemma D.11, and equality (b) uses the fact that $\lambda := \frac{H}{m}$. $\qquad\square$

### D.5. A Lower Bound for Unknown Transition and its Proof

Below we provide a lower bound for RL with sum segment feedback and unknown transition with the proof.

**Theorem D.15.** *Consider the problem of RL with sum segment feedback and unknown transition. There exists a distribution of instances where the regret of any algorithm must be*

$$\Omega\left(r_{\max}H\sqrt{|\mathcal{S}||\mathcal{A}|K}\right).$$

*Proof of Theorem D.15.* We construct a random instance $\mathcal{I}$ as follows. As shown in Figure 7, there are $n$ bandit states $s_1, \ldots, s_n$ (i.e., there are an optimal action and multiple suboptimal actions), a good absorbing state $s_{n+1}$ and a bad absorbing state $s_{n+2}$. The agent starts from $s_1, \ldots, s_n$ with equal probability $\frac{1}{n}$. For any $i \in [n]$, in state $s_i$, one action $a_J$ is uniformly chosen from $\mathcal{A}$ as the optimal action. In state $s_i$, under the optimal action $a_J$, the agent transitions to $s_{n+1}$ and $s_{n+2}$ with probabilities $\frac{1}{2} + \varepsilon$ and $\frac{1}{2} - \varepsilon$, respectively, where $\varepsilon \in (0, \frac{1}{4})$ is a parameter specified later; Under any suboptimal action $a \in \mathcal{A} \setminus \{s_J\}$, the agent transitions to $s_{n+1}$ and $s_{n+2}$ with equal probability $\frac{1}{2}$.

The rewards are deterministic for all state-action pairs. For any $a \in \mathcal{A}$, $r(s_{n+1}, a) = r_{\max}$. For any $i \in \{1, ..., n, n+2\}$ and $a \in \mathcal{A}$, $r(s_i, a) = 0$.

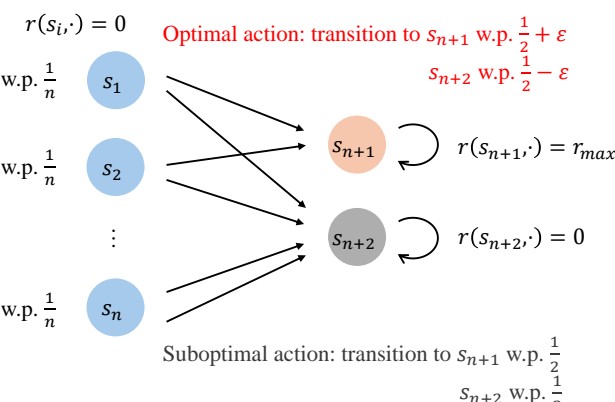

*Figure 7.* Instance for the lower bound under sum segment feedback and unknown transition.

In this proof, we will also use an alternative uniform instance $\mathcal{I}_{\text{unif}}$. The only difference between $\mathcal{I}_{\text{unif}}$ and $\mathcal{I}$ is that for any $i \in [n]$, in state $s_i$, under all actions $a \in \mathcal{A}$, the agent transitions to $s_{n+1}$ and $s_{n+2}$ with equal probability $\frac{1}{2}$.

Fix an algorithm $\mathbb{A}$. Let $\mathbb{E}_{\text{unif}}[\cdot]$ denote the expectation with respect to $\mathcal{I}_{\text{unif}}$. Let $\mathbb{E}_*[\cdot]$ denote the expectation with respect to $\mathcal{I}$. For any $i \in [n]$ and $j \in [|\mathcal{A}|]$, let $\mathbb{E}_{i,j}[\cdot]$ denote the expectation with respect to the case where $a_j$ is the optimal action in state $s_i$, and $N_{i,j}$ denote the number of episodes where algorithm $\mathbb{A}$ chooses $a_j$ in state $s_i$, i.e., $N_{i,j} = \sum_{k=1}^{K} \mathbb{1}\{\pi_1^k(s_i) = a_j\}$.

The KL divergence of transition distribution on $(s_i, a_J)$ $(i \in [n])$ between $\mathcal{I}_{\text{unif}}$ and $\mathcal{I}$ is

$$
\text{KL}\left( \mathcal{B}\left(\frac{1}{2}\right) \| \mathcal{B}\left(\frac{1}{2} + \varepsilon\right) \right) = \frac{1}{2} \ln\left( \frac{\frac{1}{2}}{\frac{1}{2} - \varepsilon} \right) + \frac{1}{2} \ln\left( \frac{\frac{1}{2}}{\frac{1}{2} + \varepsilon} \right)
$$

$$
= \frac{1}{2} \ln\left( \frac{\frac{1}{4}}{\frac{1}{4} - \varepsilon^2} \right)
$$

$$
= -\frac{1}{2} \ln\left( 1 - 4\varepsilon^2 \right)
$$

$$
\overset{(a)}{\leq} 4\varepsilon^2,
$$

where (a) uses the fact that $-\ln(1 - x) \leq 2x$ when $x \in (0, \frac{1}{4})$.

In addition, the agent has probability only $\frac{1}{n}$ to arrive at (observe) state $s_i$.

Thus, using Lemma A.1 in (Auer et al., 2002), we have that for any $i \in [n]$, in state $s_i$,

$$
\mathbb{E}_{i,j}[N_{i,j}] \leq \mathbb{E}_{\text{unif}}[N_{i,j}] + \frac{K}{2}\sqrt{ \frac{1}{n} \cdot \mathbb{E}_{\text{unif}}[N_{i,j}] \cdot \text{KL}\left( \mathcal{B}\left(\frac{1}{2}\right) \| \mathcal{B}\left(\frac{1}{2} + \varepsilon\right) \right)}
$$

$$
\leq \mathbb{E}_{\text{unif}}[N_{i,j}] + \frac{K}{2}\sqrt{ \frac{1}{n} \cdot \mathbb{E}_{\text{unif}}[N_{i,j}] \cdot 4\varepsilon^2}
$$

$$
= \mathbb{E}_{\text{unif}}[N_{i,j}] + K\varepsilon\sqrt{ \frac{1}{n} \cdot \mathbb{E}_{\text{unif}}[N_{i,j}]}.
$$

Summing over $j \in [|\mathcal{A}|]$, using the Cauchy-Schwarz inequality and the fact that $\sum_{j=1}^{|\mathcal{A}|} \mathbb{E}_{\text{unif}}[N_{i,j}] = K$, we have

$$
\sum_{j=1}^{|\mathcal{A}|} \mathbb{E}_{i,j}[N_{i,j}] \leq K + K\varepsilon\sqrt{ \frac{|\mathcal{A}|}{n} \cdot K}.
$$

Then, we have

$$
\mathcal{R}(K) = \sum_{k=1}^{K} \mathbb{E}_*\left[ V^* - V^{\pi^k} \right]
$$

$$= \left(\frac{1}{2} + \varepsilon\right)(H-1)r_{\max}K$$

$$- \frac{1}{n}\sum_{i=1}^{n}\left(\frac{1}{2}(H-1)r_{\max}K + \varepsilon(H-1)r_{\max}\cdot\frac{1}{|\mathcal{A}|}\sum_{j=1}^{|\mathcal{A}|}\mathbb{E}_{i,j}[N_{i,j}]\right)$$

$$\geq \varepsilon(H-1)r_{\max}\left(K - \frac{K}{|\mathcal{A}|} - K\varepsilon\sqrt{\frac{K}{|\mathcal{A}|n}}\right).$$

Recall that $n = |\mathcal{S}| - 2$. Let $|\mathcal{S}| \geq 3$, $|\mathcal{A}| \geq 2$, $H \geq 2$, $K > |\mathcal{A}|n$ and $\varepsilon = \frac{1}{4}\sqrt{\frac{|\mathcal{A}|n}{K}}$. Then, we have

$$\mathcal{R}(K) = \Omega\left(r_{\max}H\sqrt{|\mathcal{S}||\mathcal{A}|K}\right).$$

$\square$

# E. Technical Tools

In this section, we introduce several technical tools.

**Lemma E.1** (Self-concordance, Lemma 9 in (Faury et al., 2020)). *For any $x_1, x_2 \in \mathbb{R}$, we have*

$$\mu'(x_1)\frac{1 - \exp(-|x_1 - x_2|)}{|x_1 - x_2|} \leq \int_{z=0}^{1}\mu'((1-z)x_1 + zx_2)dz \leq \mu'(x_1)\frac{\exp(|x_1 - x_2|) - 1}{|x_1 - x_2|}.$$

*Furthermore, we have*

$$\int_{z=0}^{1}\mu'((1-z)x_1 + zx_2)dz \geq \frac{\mu'(x_1)}{1 + |x_1 - x_2|}.$$

**Lemma E.2** (Value Difference Lemma, Lemma E.15 in (Dann et al., 2017)). *For any two MDPs $M'$ and $M''$ with rewards $r'$ and $r''$ and transition distributions $p'$ and $p''$, we have that for any $h \in [H]$ and $s \in \mathcal{S}$,*

$$V_h'(s) - V_h''(s) = \mathbb{E}_{p''}\left[\sum_{t=h}^{H}\left(r'(s_t, a_t) - r''(s_t, a_t) + (p'(\cdot|s_t, a_t) - p''(\cdot|s_t, a_t))^\top V_{h+1}'(\cdot)\right)|s_t = s\right].$$

**Lemma E.3** (Law of Total Variance, Lemma 15 in (Zanette & Brunskill, 2019)). *For an MDP $p$ and a fixed policy $\pi$, we have*

$$\mathbb{E}_{\pi,p}\left[\left(\sum_{h=1}^{H}r(s_h, \pi_h(s)) - V_1^\pi(s_1)\right)\Bigg|s_1\right] = \mathbb{E}_{\pi,p}\left[\sum_{h=1}^{H}\text{Var}_{s_{h+1}\sim p(\cdot|s_h,\pi_h(s_h))}\left(V_{h+1}^\pi(s_{h+1})\right)\Bigg|s_1\right].$$

The idea of Lemma E.3 was also used in earlier works, e.g., (Munos & Moore, 1999; Lattimore & Hutter, 2012; Gheshlaghi Azar et al., 2013).

**Lemma E.4** (Lemma 10 in (Ménard et al., 2021)). *For distributions $p, q \in \triangle_{\mathcal{S}}$ and function $f : \mathcal{S} \to [0, b]$, if $\text{KL}(p, q) \leq \alpha$, then*

$$|(p(\cdot) - q(\cdot))^\top f(\cdot)| \leq \sqrt{2\text{Var}_q(f)\alpha} + \frac{2}{3}b\alpha.$$

**Lemma E.5** (Lemma 11 in (Ménard et al., 2021)). *For distributions $p, q \in \triangle_{\mathcal{S}}$ and function $f : \mathcal{S} \to [0, b]$, if $\text{KL}(p, q) \leq \alpha$, then*

$$\text{Var}_q(f) \leq 2\text{Var}_p(f) + 4b^2\alpha.$$

**Lemma E.6** (Lemma 12 in (Ménard et al., 2021)). *For distribution $p \in \triangle_{\mathcal{S}}$ and functions $f, g : \mathcal{S} \to [0, b]$, we have*

$$\text{Var}_p(f) \leq 2\text{Var}_p(g) + 2bp(\cdot)^\top|f(\cdot) - g(\cdot)|.$$

