# OpenReview forum: "Reinforcement Learning with Segment Feedback"
_ICML.cc/2025/Conference — ICML 2025 poster_

### Official Review · Reviewer_FY6P · 2025-03-11

**Overall Recommendation:** 3

**Summary:**

This paper studies RL with segment feedback, where each episode in an MDP is divided into $m$ equal-length segments, and the agent receives reward feedback only at the end of each segment instead of every state-action pair. The authors investigate two feedback settings: binary feedback, where the agent receives a binary outcome per segment, and sum feedback, where the agent observes the cumulative reward within a segment. They design TS-style algorithms for binary feedback and UCB-style algorithms for sum feedback, establishing regret upper and lower bounds for both known and unknown transitions. Their theoretical and empirical results show that increasing $m$ exponentially decreases regret under binary feedback, but has little effect under sum feedback.

**Claims And Evidence:**

The paper's claims are generally well-supported by theoretical analysis and empirical results. However, two strong assumptions should be further discussed:
1. The paper assumes that the type of segment feedback (sum or binary) is known in advance. Since the conclusions differ significantly depending on the feedback type, understanding the feedback structure seems crucial before applying these methods. However, in real-world scenarios, it is often difficult to determine how segment feedback is derived from step-level rewards.
2. The assumption that each episode is divided into $m$ equal-length segments is somewhat strong and may not always hold in real-world scenarios where feedback aggregation is irregular. A discussion on the potential impact of non-uniform segment lengths on the proposed methods, along with possible extensions to handle variable-length segments, would improve the robustness of the claims.

**Essential References Not Discussed:**

The paper does not discuss "Harnessing Causality in Reinforcement Learning with Bagged Decision Times" (AISTATS 2025), which presents a similar problem setting.

**Experimental Designs Or Analyses:**

Since trajectory feedback algorithms can be naturally extended to the segment feedback setting, it is important to clarify the advantages of the proposed approach over these existing methods. An experimental comparison with trajectory feedback algorithms (e.g., [1, 2]) should also be included.

[1] On the Theory of Reinforcement Learning with Once-per-Episode Feedback.
[2] Reinforcement Learning with Trajectory Feedback.

**Methods And Evaluation Criteria:**

The authors design TS-style algorithms for binary feedback and UCB-style algorithms for sum feedback, which are appropriate choices given the problem structure.

**Other Comments Or Suggestions:**

1. (Claims And Evidence) Discuss potential extensions to non-uniform segment lengths to enhance practical applicability.
2. (Claims And Evidence) In practical applications, how do we determine whether segment feedback follows the sum or binary structure, or if it follows a different aggregation rule altogether? Since the proposed methods rely on knowing the feedback type, addressing this question would improve their real-world usability.
3. Can the proposed methods be understood as making structural assumptions about the reward aggregation process? If so, does this imply that any form of segment feedback (inclued binary segment feedback) could, in principle, be analyzed based on sum segment feedback as a general case?
4. (Experimental Designs or Analyses) The paper does not include a comparison with trajectory feedback methods. Since trajectory feedback algorithms can be extended to the segment feedback setting, an experimental comparison would strengthen the empirical evaluation and clarify the advantages of the proposed approach.

I will look forward to your response.

**Other Strengths And Weaknesses:**

**Strengths**

1. The paper is well written, and the theoretical results appear correct, with detailed analysis and proofs supporting the claims.
2. The counterintuitive finding that increasing $m$ does not significantly reduce regret under sum feedback is well-supported by both theoretical analysis and experimental results.

**Weaknesses**

See comments.

**Questions For Authors:**

See comments.

**Relation To Broader Scientific Literature:**

This paper discusses segment feedback, a topic previously studied from different perspectives in [1,2]. Building on [3], the authors design TS-style algorithms for binary feedback, and based on [4], they develop UCB-style algorithms for sum feedback.

[1] Reinforcement Learning from Bagged Reward.
[2] Harnessing Causality in Reinforcement Learning with Bagged Decision Times.
[3] On the Theory of Reinforcement Learning with Once-per-Episode Feedback.
[4] Reinforcement Learning with Trajectory Feedback.

**Theoretical Claims:**

The theoretical claims and proofs, particularly those on regret bounds, appear correct and well-structured, with no major issues observed.

---

> ### Author Rebuttal · Authors · 2025-03-31
>
> Thank you for your time in reviewing our paper! We will revise our paper according to your comments.
>
> **1. Determine the Feedback Type**
>
> We assume an MDP setting, so the sum of step-level rewards being the trajectory reward is natural. Given this model, we can ask a human to give a score to a trajectory, which is an approximation to the sum feedback and has been considered in prior works cited in our paper. Or we can ask for thumbs up/down feedback which is the binary feedback model. So we believe that in our context, it is reasonable to assume that the type of feedback is known ahead of time since the type of feedback is designed by us.
>
> **2. Assumption on Equal-Length Segments**
>
> We agree with the reviewer that in practice, the segment length may be unequal. However, studying segments of equal length is an important starting point which has uncovered non-trivial challenges, and we reveal a new and surprising result: segments help under binary feedback, but do not help much under sum feedback.
>
> We believe that our algorithms and analysis can be extended to segments of variable length. For example, a natural extension is to consider that the arrival of reward feedback follows a point process. In this case, the variable segment length will affect the noise variance of reward feedback, and the sum analysis of segment visitation indicators. One idea to handle this case is to obtain high probability bounds of the segment length and the number of segments within a trajectory, and use these bounds in our analysis.
>
> **3. Experimental Comparison**
>
> We ran preliminary experiments to compare our algorithms with the adaptations of trajectory feedback algorithms [1, 2] (reviewer's reference numbers) to the segment setting.
>
> In our experiments, we set $|\mathcal{A}|=5$, $r_{\max}=0.5$, $\delta=0.005$ and $m=2$. In the binary feedback setting, we set $|\mathcal{S}|=5$, $H=6$ and $K=1000$. In the sum feedback setting, we set $|\mathcal{S}|=3$, $H=10$ and $K=5000$. For each algorithm, we performed $15$ independent runs and presented the average regret with standard deviation, and average running time across runs.
>
>
> | Binary Feedback |  Regret (Std.)    |  Running Time (s)    |
> | -------- | -------- | -------- |
> | SegBiTS    | 234.26 (35.12)    | 36.98     |
> | Adaptation of "UCBVI with trajectory labels" [1]     | 199.94 (29.54)    | 7796.90     |
>
> | Sum Feedback |  Regret (Std.)    |  Running Time (s)    |
> | -------- | -------- | -------- |
> | E-LinUCB | 3954.66 (7.68)    | 940.15     |
> | Adaptation of OFUL [2]  | 4180.95 (12.74)    | 925.60     |
>
>
> As shown in the tables, under binary feedback, while our algorithm SegBiTS has a slightly higher regret than the adaptation of "UCBVI with trajectory labels" [1], SegBiTS achieves a significantly faster running time, since SegBiTS is computationally efficient while "UCBVI with trajectory labels" is not. Under sum feedback, our algorithm E-LinUCB attains a lower regret than the adaptation of OFUL [2], since E-LinUCB employs the experimental design technique to sharpen the regret.
>
> The experimental results for $m\in\\{1,H\\}$ are similar. Here we only present the results for $m=2$ due to space limit. We will include more experiments in our revision.
>
> **4. Reviewer's Reference [2] ("Harnessing ...")**
>
> The problem and focus in [2] are very different from ours. [2] considers RL with bagged decision times, where the state transitions are non-Markovian within a bag, and a reward is observed at the end of the bag. This problem can be regarded as a periodic MDP with non-stationary state transitions and reward functions in a period, and [2] adapts an RL algorithm for standard stationary MDPs to this periodic MDP to solve this problem. The focus of [2] is to handle the non-Markovian state transitions within a bag using a causal directed acyclic graph, instead of investigating bagged rewards. [2] does not consider how to infer the reward function of state-action pairs from bagged rewards, or study how the frequency of reward feedback impacts learning performance like us.
>
> We will cite [2] and include this discussion in our revision.
>
> **5. Structural Assumptions**
>
> We assume an MDP model, which is widely used in reinforcement learning. In the MDP model, the reward of any segment of any trajectory is the sum of the rewards of each individual step in the segment. So the type of reward aggregation is fixed by our MDP model. And for binary feedback, we assume that it is generated according to a sigmoid function, which is also widely accepted and used. Beyond these two assumptions, we didn't make any other structural assumptions. To the best of our knowledge, the fact that one gets extremely coarse information about the sum reward in the binary feedback case makes it impossible to have a common analytical technique for both feedback models.

---

### Official Review · Reviewer_spk8 · 2025-03-12

**Overall Recommendation:** 4

**Summary:**

This paper studies the problem of RL with segment feedback, which offers a general paradigm filling the gap between per-state-action feedback and trajectory feedback. The authors study two settings with binary and sum feedback, developing computationally efficient algorithms achieving nearly optimal regret upper bounds. They show that the number of segment m significantly matters in the binary feedback setting, while not that important in the sum feedback setting. They also provide regret lower bounds and experimental results to validate their claims.

**Claims And Evidence:**

Yes.

**Essential References Not Discussed:**

No.

**Ethical Review Flag:**

Flag this paper for an ethics review.

**Experimental Designs Or Analyses:**

No.

**Methods And Evaluation Criteria:**

Yes.

**Other Comments Or Suggestions:**

None.

**Other Strengths And Weaknesses:**

Strengths:

1. This is the first work to theoretically study the important RL with segment feedback setting. It is well motivated.
2. The paper is well-written and easy-to-follow.
3. The authors conduct a complete study in this topic, providing algorithms with nearly optimal regret upper bounds, and proving novel lower bounds. They also provide experimental results to support the theoretical findings.

Weaknesses:

The authors consider a different setting for the binary feedback from the previous work (Chatterji et al.,2021). It would be better to explain why the setting the authors considered is better.

**Questions For Authors:**

Could you please explain more about why the setting for the binary feedback considered in this work is better than that of (Chatterji et al.,2021)?

**Relation To Broader Scientific Literature:**

There have been several works that consider RL with trajectory feedback (Efronietal.,2021; Chatterjietal.,2021).  And a recent work, Tang et al.(2024), studies the segment feedback, yet only from the empirical aspect. This is the first work to theoretically study RL with segment feedback.

**Theoretical Claims:**

No.

---

> ### Author Rebuttal · Authors · 2025-03-31
>
> Thank you for your time and effort in reviewing our paper! We will certainly incorporate your comments in our revision.
>
> **1. Comparison with [Chatterji et al., 2021] in Formulation**
>
> The objectives in [Chatterji et al., 2021] and our paper are different. In [Chatterji et al., 2021], the goal is to choose a policy which maximizes the probability of a trajectory getting a rating of $1$, i.e., getting positive feedback. Whereas, in our model, our goal is to maximize the expected sum of rewards over trajectories. In our opinion, both models have value and are applicable in different contexts. In particular, our model is better suited to situations where we want to solve an MDP but only get binary segment feedback.
>
> ---
>
> Reference:
>
> Chatterji et al. On the theory of reinforcement learning with once-per-episode feedback. NeurIPS, 2021.

---

### Official Review · Reviewer_JY4c · 2025-03-14

**Overall Recommendation:** 3

**Summary:**

This paper introduces Reinforcement Learning with Segment Feedback, a framework where episodic MDPs are divided into
$m$ segments, and feedback is provided at the end of each segment. The authors study binary feedback (stochastic binary outcomes) and sum feedback (cumulative rewards with noise), analyzing how the number of segments $m$ impacts learning efficiency.

**Claims And Evidence:**

**Binary Feedback: Exponential Regret Reduction with More Segments**

Theoretical Bounds: Theorems 4.1 and 4.3 establish regret upper bounds scaling as $\tilde{O}(\exp(H/m)/\sqrt{K})$, while Theorem 4.2 provides a matching lower bound. The exponential dependency arises from the sigmoid function’s sensitivity: smaller segments reduce the effective reward scale, improving distinguishability of actions.

Figure 2a shows regret decreases rapidly as $m$ increases.

**Sum Feedback: Regret Insensitive**
Theorems 5.1 and 5.3 show regret $\tilde{O}(\sqrt{HK})$ independent of $m$, with a matching lower bound (Theorem 5.2). The intuition is that segment-level noise and covariance matrix shrinkage counteract the benefit of more observations.

Figure 2b shows  minimal regret variation across $m$ validating the theory.

**Essential References Not Discussed:**

This paper properly discusses the references I know.

**Experimental Designs Or Analyses:**

The authors use a very simple experiment to show verify the theory results. See **Claims And Evidence**.

I understand that this is mainly a theory paper, but is it possible to experiment on more complex environments than what you have used, such as Atari.

**Methods And Evaluation Criteria:**

**SegBiTS** (binary feedback): Uses Thompson sampling with Gaussian perturbations and MLE for reward inference.

**E-LinUCB** (sum feedback): Incorporates E-optimal design to refine the covariance matrix, improves Efroni et al. (2021) by a $\sqrt{H}$ factor.

In the experiments, the authors plot the cumulative regret to verify the theory.

**Other Comments Or Suggestions:**

Please see previous reviews.

**Other Strengths And Weaknesses:**

Please see previous reviews.

**Questions For Authors:**

I think this paper deals with tabular RL. Do you expect similar results will happen in non-tabular settings?

**Relation To Broader Scientific Literature:**

**Bridging Classic RL and Trajectory Feedback**: The paper introduces a new RL model with segment feedback that acts as a middle ground between per-state-action feedback in classic RL (Sutton & Barto, 2018) and trajectory feedback (Efroni et al., 2021; Chatterji et al., 2021)​. This novel paradigm allows feedback at the segment level, which is more practical in real-world scenarios like autonomous driving and robotics. The authors highlight that previous works on trajectory feedback rely on the martingale property of entire trajectories (Efroni et al., 2021), which does not hold for segment feedback due to the dependency among segments within a trajectory.

**Theoretical Claims:**

This is a theory paper. See **Claims And Evidence**.

---

> ### Author Rebuttal · Authors · 2025-03-31
>
> Thank you for your time and effort in reviewing our paper! We will certainly incorporate your suggestions in our revision.
>
> **1. More Experiments**
>
> This is an excellent suggestion. However, it is difficult to perform experiments on the Atari environments before the rebuttal deadline since it would take some time to adapt our discrete state-space setting to Atari's continuous state space setting of images. However, following the reviewer's suggestion, we ran preliminary experiments on a more complex Grid World environment called Cliff Walking, which is used in the popular RL textbook [Sutton \& Barto, 2018].
>
>
> | (0) | (0) | (0) | (0) | (0) | (0) |
> | -------- | -------- | -------- | -------- | -------- | -------- |
> |   $\rightarrow$ (r_max)     |  $\rightarrow$ (r_max)    |  $\rightarrow$ (r_max)     |  $\rightarrow$ (r_max)     |  $\rightarrow$ (r_max)     |  $\downarrow$  (r_max)     |
> | $\uparrow$ start (0)     | $\otimes$ (-r_max)    |  $\otimes$ (-r_max)     |  $\otimes$ (-r_max)     |  $\otimes$ (-r_max)     |  end (r_max)     |
>
>
> The above table depicts the Cliff Walking environment considered in our experiments. Each table cell denotes a state. An agent starts from the left bottom state (marked as "start") and aims to arrive at the right bottom state (marked as "end"). The reward of each state is marked in the bracket $(\cdot)$ in the state. The states with $\otimes$ denote a cliff and generate negative rewards $-r_{\max}$. The optimal path is the path close to the cliff, and the path close to the upper wall is suboptimal.
> In our experiments, $|\mathcal{S}|=18$, $|\mathcal{A}|=4$, $H=8$, $m\in\\{1,4,8\\}$, $K=50000$, $r_{\max}=2$ and $\delta=0.005$. We performed our algorithm SegBiTS for $3$ independent runs under binary segment feedback, and reported the average regret with standard deviation to see how the number of segments $m$ impacts learning performance.
>
>
> | Setting | Regret of SegBiTS (Std.) |
> | -------- | -------- |
> | $H=8$, $m=1$     |  $1.114 \times 10^6$ ($1.112 \times 10^3$)   |
> | $H=8$, $m=4$     |  $6.365  \times 10^5$ ($1.167 \times 10^3$)   |
> | $H=8$, $m=8$    |  $4.544 \times 10^5$ ($1.186 \times 10^2$)   |
>
> As shown in the  table, under binary feedback, as the number of segments $m$ increases, the regret significantly decreases, which matches our theoretical finding.
>
> Due to time limit, here we only present the results for this instance and binary segment feedback. We will include more experimental results on larger instances in our revision.
>
> ---
>
> Reference:
>
> Richard S. Sutton and Andrew G. Barto. Reinforcement learning: An introduction. 2018.
>
> **2. Non-tabular Settings**
>
> Yes, we expect that similar results will hold in non-tabular settings, e.g., linear function approximation.
> In the linear function approximation setting, it is assumed that each state-action pair is associated with a feature vector $\phi(s,a) \in \mathbb{R}^d$, there exists a global reward parameter $\theta^* \in \mathbb{R}^d$, and segment reward feedback is generated according to $(\sum_{h \in \mathcal{H}} \phi(s_h,a_h))^\top \theta^*$, where $\mathcal{H}$ denotes the set of step indices in a segment. Since our analytical procedure is mainly based on visitation indicators $\phi(s,a)$ and the fact that segment reward feedback is generated linearly with respect to $\phi(s,a)$, we believe that generalizing $\phi(s,a)$ from a visitation indicator in the tabular setting to a feature vector in the linear function approximation setting should be possible.
> We will further investigate non-tabular settings in our future work.

---

### Decision · Program_Chairs · 2025-05-01

**Decision:**

Accept (poster)

**Comment:**

This paper provides a theoretical analysis of Reinforcement Learning with segment feedback. They study two settings with binary and sum feedback, developing computationally efficient algorithms achieving nearly optimal regret bounds for each one of these. During the discussion phase with reviewers, it was agreed this work provides a meaningful contribution to the study or RL with novel forms of feedback. That is why I am recommending acceptance.